# Posterior Meta-Replay for Continual Learning

**Christian Henning\*, Maria R. Cervera\*,**
**Francesco D'Angelo, Johannes von Oswald, Regina Traber,**
**Benjamin Ehret, Seijin Kobayashi, Benjamin F. Grewe, João Sacramento**

\*Equal contribution

Institute of Neuroinformatics
University of Zürich and ETH Zürich
Zürich, Switzerland
`{henningc,mariacer}@ethz.ch`

## Abstract

Learning a sequence of tasks without access to i.i.d. observations is a widely studied form of continual learning (CL) that remains challenging. In principle, Bayesian learning directly applies to this setting, since recursive and one-off Bayesian updates yield the same result. In practice, however, recursive updating often leads to poor trade-off solutions across tasks because approximate inference is necessary for most models of interest. Here, we describe an alternative Bayesian approach where task-conditioned parameter distributions are continually inferred from data. We offer a practical deep learning implementation of our framework based on probabilistic task-conditioned hypernetworks, an approach we term *posterior meta-replay*. Experiments on standard benchmarks show that our probabilistic hypernetworks compress sequences of posterior parameter distributions with virtually no forgetting. We obtain considerable performance gains compared to existing Bayesian CL methods, and identify task inference as our major limiting factor. This limitation has several causes that are independent of the considered sequential setting, opening up new avenues for progress in CL.

## 1 Introduction

In recent years, a variety of continual learning (CL) algorithms have been developed to overcome the need to train neural networks with an independent and identically distributed (i.i.d.) sample. Most CL literature focuses on the particular scenario of continually learning a sequence of $T$ tasks with datasets $\mathcal{D}^{(1)}, \ldots, \mathcal{D}^{(T)}$. Because only access to the current task is granted, successful training of a discriminative model that captures $p(\mathbf{Y} \mid \mathbf{X})$ has to occur without an i.i.d. training sample from the overall joint $\mathcal{D}^{(1:T)} \overset{i.i.d.}{\sim} p(\mathbf{X})p(\mathbf{Y} \mid \mathbf{X})$.

The advantages of a Bayesian approach for solving this problem are numerous and include the ability to drop all i.i.d. assumptions across and within tasks in a mathematically sound way, the ability to revisit tasks whenever new data becomes available, and access to principled uncertainty estimates capturing both data and parameter uncertainty. Up until now, Bayesian approaches to CL essentially focused on finding a combined posterior distribution via a recursive Bayesian update $p(\mathbf{W} \mid \mathcal{D}^{(1:T)}) \propto p(\mathbf{W} \mid \mathcal{D}^{(1:T-1)})p(\mathcal{D}^{(T)} \mid \mathbf{W})$. Because the posterior of the previous task is used as prior for the next task, these approaches are also known as **prior-focused** [17]. In theory, the

35th Conference on Neural Information Processing Systems (NeurIPS 2021).

above recursive update can always recover the posterior $p(\mathbf{W} \mid \mathcal{D}^{(1:T)})$, independently of how the data is presented. However, because proper Bayesian inference is intractable, approximations are needed in practice, which lead to errors that are recursively amplified. As a result, whether solutions that are easily found in the i.i.d. setting can be obtained via the approximate recursive update strongly depends on factors such as task ordering, task similarity and the considered family of distributions. These factors limit the effectiveness of the recursive update and have a detrimental effect on the performance of prior-focused methods, especially in task-agnostic CL settings.

To overcome these limitations, we propose an alternative Bayesian approach to CL that does not rely on the recursive update to learn distinct tasks and instead aims to learn task-specific posteriors (Fig. 1, refer to SM F.1 for a detailed discussion of the graphical model). In this view, finding trade-off solutions across tasks is not required, and knowledge transfer can be explicitly controlled for each task via the prior, which is no longer prescribed by the recursive update and can thus be set freely. By introducing probabilistic extensions of task-conditioned hypernetworks [91], we show how task-specific posteriors can be learned with a single shared meta-model, an approach we term **posterior meta-replay**.

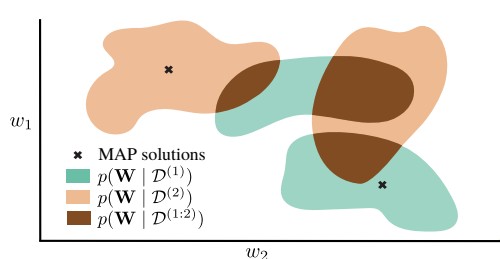

Figure 1: The proposed *posterior meta-replay* framework learns task-specific posteriors $p(\mathbf{W} \mid \mathcal{D}^{(t)})$ via a single shared meta-model, with task-specific point estimates (e.g., MAP) being a limit case. In this view, the modelled solution space is not limited to admissible solutions that lie in the overlap of all task-specific posteriors. By contrast, *prior-focused* methods learn a single posterior $p(\mathbf{W} \mid \mathcal{D}^{(1:T)})$ recursively and thus require the existence of trade-off solutions between learned and future tasks in the currently modelled solution space. Shaded areas indicate high density regions.

This approach introduces two challenges: forgetting at the level of the hypernetwork, and the need to know task identity to correctly condition the hypernetwork. We empirically show that forgetting at the meta-level can be prevented by using a simple regularizer that replays parameters of previous posteriors. In task-agnostic inference settings, often referred to as *class-incremental learning* in the context of classification benchmarks [88], the main hurdle therefore becomes task inference at test time. Here we focus on this task-agnostic setting, arguably the most challenging but also the most natural CL scenario, since the obtained models can be deployed just like those obtained via i.i.d. training (e.g., irrespective of the sequential training, the final model will be a classifier across all classes). In order to explicitly infer task identity from unseen inputs without resorting to generative models, we thoroughly study the use of principled uncertainty that naturally arises in Bayesian models. We show that results obtained in this task-agnostic setting with our approach constitute a leap in performance compared to prior-focused methods. Furthermore we show that limitations in task inference via predictive uncertainty are not related to our CL solution, but depend instead on the combination of approximate inference method, architecture, uncertainty measure and prior. Finally, we investigate how task inference can be further improved through several extensions.

We summarize our main contributions below:

- We describe a Bayesian CL framework where task-conditioned posterior parameter distributions are continually learned and compressed in a hypernetwork.

- In a series of synthetic and real-world CL benchmarks we show that our task-conditioned hypernetworks exhibit essentially no forgetting, both for explicitly parameterized and implicit posterior distributions, despite using the parameter budget of a single model.

- Compared to prior-focused methods, our approach leads to a leap in performance in task-agnostic inference while maintaining the theoretical benefits of a Bayesian approach.

- Our approach scales to modern architectures such as ResNets, and remaining performance limitations are linked to uncertainty-based out-of-distribution detection but not to our CL solution.

- Finally, we show how prominent existing Bayesian CL methods such as elastic weight consolidation can be dramatically improved in task-agnostic settings by introducing a small set of task-specific parameters and explicitly inferring the task.

## 2 Related Work

**Continual learning.** CL algorithms attempt to mitigate catastrophic interference while facilitating transfer of skills whenever possible. They can be coarsely categorized as (1) *regularization-methods* that put constraints on weight updates, (2) *replay-methods* that mimic pseudo-i.i.d. training by rehearsing stored or generated data and (3) *dynamic architectures* which can grow to allocate capacity for new knowledge [71]. Most related to our work is the study from von Oswald et al. [91] that introduces task-conditioned hypernetworks for CL, and already considers task inference via predictive uncertainty in the deterministic case. Our framework can be seen as a probabilistic extension of their work, which provides task-specific point estimates via a shared meta-model (cf. Sec. 3). Follow-up work also achieves task inference via predictive uncertainty, e.g., Wortsman et al. [94] use it to select a learned binary mask per task that modulates a random base network. Here we complement these studies by thoroughly exploring task inference via several uncertainty measures, disclosing the factors that limit task inference and highlighting the importance of parameter uncertainty.

A variety of methods tackling CL have been derived from a Bayesian perspective. A prominent example are *prior-focused* methods [17], which incorporate knowledge from past data via the prior and, in contrast to our work, aim to find a shared posterior for all data. Examples include (Online) EWC [38, 78] and VCL [65, 54]. Other methods like CN-DPM [46] use Bayes' rule for task inference on the joint $p(\mathbf{X}, C)$, where $C$ is a discrete condition such as task identity. An evident downside of CN-DPM is the need for a separate generative and discriminative model per condition. More generally, such an approach requires meaningful density estimation in the input space, a requirement that is challenging for modern ML problems [64].

Other Bayesian CL approaches consider instead task-specific posterior parameter distributions. Lee et al. [47] learn separate task-specific Gaussian posterior approximations which are merged into a single posterior after all tasks have been seen. CBLN [49] also learns a separate Gaussian posterior approximation per task but later tries to merge similar posteriors in the induced Gaussian mixture model. Task inference is thus required and achieved via predictive uncertainty, although for a more reliable estimation all experiments consider batches of 200 samples that are assumed to belong to the same task. Tuor et al. [85] also learn a separate approximate posterior per task and use predictive uncertainty for task-boundary detection and task inference. In contrast to these approaches, we learn all task-specific posteriors via a single shared meta-model and remain agnostic to the approximate inference method being used. A conceptually related approach is MERLIN [33], which learns task-specific weight distributions by training an ensemble of models per task that is used as training set for a task-conditioned variational autoencoder. Importantly, MERLIN requires a fine-tuning stage at inference, such that every drawn model is fine-tuned on stored coresets, i.e., a small set of samples withheld throughout training. By contrast, our approach learns the parameters of an approximate Bayesian posterior $p(\mathbf{W} \mid \mathcal{D}^{(t)})$ per task $t$, and no fine-tuning of drawn models is required.

**Bayesian neural networks.** Because neural networks are expressive enough to fit almost any data [98] and are often deployed in an overparametrized regime, it is implausible to expect that any single solution obtained from limited data generalizes to the ground truth $p(\mathbf{Y} \mid \mathbf{w}, \mathbf{X}) \approx p(\mathbf{Y} \mid \mathbf{X})$ almost everywhere on $p(x)$. By contrast, Bayesian statistics considers a distribution over models, explicitly handling uncertainty to acknowledge data insufficiencies. This distribution is called the *posterior parameter distribution* $p(\mathbf{W} \mid \mathcal{D}) \propto p(\mathcal{D} \mid \mathbf{W}) p(\mathbf{W})$, which weights models based on their ability to fit the data (via the likelihood $p(\mathcal{D} \mid \mathbf{W})$), while considering only plausible models according to the prior $p(\mathbf{W})$. Predictions are made by marginalizing over models (for an introduction see MacKay [57]). Bayesian neural networks (BNN) apply this formalism to network parameters $\mathbf{w}$, whereas for practical reasons hyperparameters like architecture are chosen deterministically [56].

While a deterministic discriminative model can only capture *aleatoric uncertainty* (i.e., uncertainty intrinsic to the data $p(\mathbf{Y} \mid \mathbf{X})$), a Bayesian treatment allows to also capture *epistemic uncertainty* by being uncertain about the model's parameters (*parameter uncertainty*). This proper treatment of uncertainty is of utmost importance for safety-critical applications, where intelligent systems are expected to *know what they don't know*. However, due to the complexity of modelling high-dimensional distributions at the scale of modern deep learning, BNNs still face severe scalability issues [82]. Here, we employ several approximations to the posterior based on variational inference [5] from prior work, ranging from simple and scalable methods with a mean-field variational family like Bayes-by-Backprop (BbB, [6]) to methods with complex but rich variational families like the spectral Stein gradient estimator [79]. For more details see Sec. 3 and SM C.

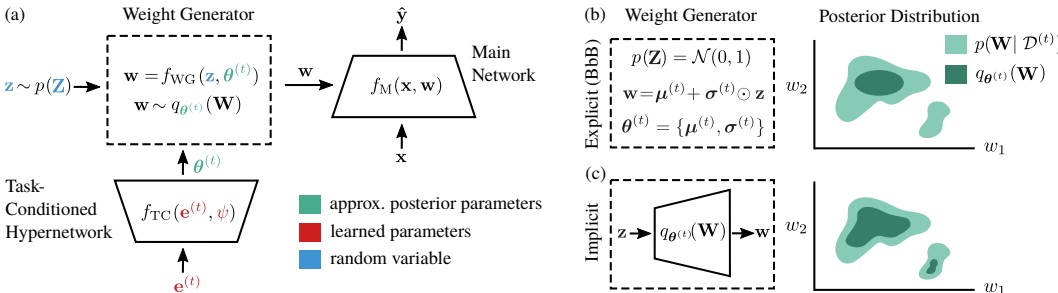

Figure 2: *Posterior meta-replay* for CL. **(a)** The architecture consists of a main network M that processes inputs $\mathbf{x}$ and generates predictions $\hat{\mathbf{y}}$ according to a set of weights $\mathbf{w}$ generated by a weight generator (WG). The WG is a deterministic function $f_{\text{WG}}(\mathbf{z}, \boldsymbol{\theta}^{(t)})$ that transforms a base distribution $p(\mathbf{Z})$ into a distribution over main network weights, where $\boldsymbol{\theta}^{(t)}$ are the parameters of the approximate posterior $q_{\boldsymbol{\theta}^{(t)}}(\mathbf{W})$. Crucially, $\boldsymbol{\theta}^{(t)}$ are task-specific, and generated by a task-conditioned (TC) hypernetwork, which receives task embeddings $\mathbf{e}^{(t)}$ as input. The embeddings and the parameters $\boldsymbol{\psi}$ of the TC are learned continually via a simple *meta-replay* regularizer (Eq. 1). **(b)** We refer to the approximate posteriors as *explicit* if $f_{\text{WG}}$ is predefined. In Bayes-by-Backprop (BbB), for example, the reparametrization trick transforms Gaussian noise into weight samples. **(c)** More complex, *implicit* posterior approximations are parametrized by an auxiliary hypernetwork, which receives its task-conditioned parameters from the TC, which now plays the role of a hyper-hypernetwork. The obtained posterior approximations are more flexible and can, for example, capture multi-modality.

## 3 Methods

In this section we describe our *posterior meta-replay* framework (Fig. 2). We start by introducing task-conditioned hypernetworks as a tool to continually learn parameters of task-specific posteriors, each of which is learned using variational inference (SM C.1). We then explain how the framework can be instantiated for both simple, explicit posterior approximations, and complex ones parametrized by an auxiliary network, and describe how forgetting can be mitigated through the use of a meta-regularizer. We next explain how predictive uncertainty, naturally arising from a probabilistic view of learning, can be used to infer task identity for both **PosteriorReplay** methods, and **PriorFocused** methods that use a multihead output. Finally, we outline ways to boost task inference.

**Task-conditioned hypernetworks.** Traditionally, hypernetworks are seen as neural networks that generate the weights $\mathbf{w}$ of a main network $\mathbf{M}$ processing inputs as $\hat{\mathbf{y}} = f_{\text{M}}(\mathbf{x}, \mathbf{w})$ [22, 77]. Here, we consider instead hypernetworks that learn to generate $\boldsymbol{\theta}$, the *parameters of a distribution* $q_{\boldsymbol{\theta}}(\mathbf{W})$ over main network weights. By taking low-dimensional task embeddings $\mathbf{e}^{(t)}$ as inputs and computing $\boldsymbol{\theta}^{(t)} = f_{\text{TC}}(\mathbf{e}^{(t)}, \boldsymbol{\psi})$, task-conditional (**TC**) computation is possible. Sampling is realized by transforming a base distribution $p(\mathbf{Z})$ via a weight generator (**WG**) $f_{\text{WG}}(\mathbf{z}, \boldsymbol{\theta}^{(t)})$, whose choice determines the family of distributions considered for the approximation (i.e., the *variational family*). In our framework, weights $\mathbf{w} \sim q_{\boldsymbol{\theta}^{(t)}}(\mathbf{W})$ are directly used for inference without requiring any fine-tuning.

Importantly, all learnable parameters are comprised in the TC system, which can be designed to have less parameters than the main network, i.e., $\dim(\boldsymbol{\psi}) + \sum_t \dim(\mathbf{e}^{(t)}) < \dim(\mathbf{w})$. Such constraint is vital to ensure fairness when comparing different CL methods, and is enforced in all our computer vision experiments. Additional details can be found in SM C.2.

**Posterior-replay with explicit distributions.** Different families of distributions can be realized within our framework. In the special case of a point estimate $q_{\boldsymbol{\theta}^{(t)}}(\mathbf{W}) = \delta(\mathbf{W} - \boldsymbol{\theta}^{(t)})$, the WG system can be omitted altogether as it corresponds to the identity $\boldsymbol{\theta}^{(t)} = f_{\text{WG}}(\mathbf{z}, \boldsymbol{\theta}^{(t)})$. This reduces our solution to the deterministic CL method introduced by von Oswald et al. [91], which we refer to as **PosteriorReplay-Dirac**. However, capturing parameter uncertainty is a key ingredient of Bayesian statistics that is necessary for more robust task inference (cf. Sec. 4.2). We thus turn as a first step to *explicit* distributions $q_{\boldsymbol{\theta}^{(t)}}(\mathbf{W})$, for which the WG system samples according to a predefined function. We refer as **PosteriorReplay-Exp** to finding a mean-field Gaussian approximation via the BbB algorithm (SM C.3.1, [6]). In this case, $\boldsymbol{\theta}^{(t)}$ corresponds to the set of means and variances that define a Gaussian for each weight, which is directly generated by the TC. In the SM, we also report results for another instance of explicit distribution (cf. SM C.3.2).

**Posterior-replay with implicit distributions.** Since the expressivity of *explicit* distributions is limited, we also explore the more diverse variational family of *implicit* distributions [19, 31]. These are parametrized by a WG that now takes the form of an auxiliary neural network, making the parameters $\boldsymbol{\theta}^{(t)}$ of the approximate posterior dependent on the chosen WG architecture. This setting, referred to as **PosteriorReplay-Imp**, results in a hierarchy of three networks: a TC network generates task-specific parameters $\boldsymbol{\theta}^{(t)}$ for the approximate posterior, which is defined through an arbitrary base distribution $p(\mathbf{Z})$ and the WG hypernetwork, which in turn generates weights $\mathbf{w}$ for a main network M that processes the actual inputs of the dataset $\mathcal{D}^{(t)}$. Interestingly, the TC now plays the role of a *hyper-hypernetwork* as it generates the weights of another hypernetwork (Fig. 2a and Fig. 2c).

Variational inference commonly resorts to optimizing an objective consisting of a data-dependent term and a *prior-matching term* $\mathrm{KL}(q_{\boldsymbol{\theta}}(\mathbf{W}) \,\|\, p(\mathbf{W}))$. Estimating the *prior-matching term* when using implicit distributions is not straightforward since we do not have analytic access to the density nor the entropy of $q_{\boldsymbol{\theta}^{(t)}}(\mathbf{W})$. To overcome this challenge, we resort to the spectral Stein gradient estimator (SSGE, SM C.4.2, [79]). This method is based on the insight that direct access to the log-density is not required, but only to its gradient with respect to $\mathbf{W}$. Noticing that this quantity appears in Stein's identity, the authors consider a spectral decomposition of the term and use the Nyström method to approximate the eigenfunctions. We test an alternative method for dealing with implicit distributions in the SM that is based on estimating the log-density ratio (SM C.4.1).

As an additional challenge introduced by the use of implicit distributions, the support of $q_{\boldsymbol{\theta}^{(t)}}(\mathbf{W})$ is limited to a low-dimensional manifold when using an inflating architecture for WG, causing the *prior-matching term* to be ill-defined. To overcome this, we investigate the use of small noise perturbations in WG outputs (SM C.4.3). Normalizing flows [70] can also be utilized as WG architectures to gain analytic access to $q_{\boldsymbol{\theta}^{(t)}}(\mathbf{W})$, albeit at the cost of architectural constraints such as invertibility.

**Overcoming forgetting via meta-replay.** Since all learnable parameters are part of the TC system, forgetting only needs to be addressed at this meta-level. With $\mathcal{L}_{\text{task}}$ the task-specific loss (SM Eq. 3) and $\mathrm{D}(\cdot\|\cdot)$ a divergence measure between distributions, the loss for task $t$ becomes:

$$\mathcal{L}^{(t)}(\boldsymbol{\psi}, \mathcal{E}, \mathcal{D}^{(t)}) = \mathcal{L}_{\text{task}}(\boldsymbol{\psi}, \mathbf{e}^{(t)}, \mathcal{D}^{(t)}) + \beta \sum_{t' < t} \mathrm{D}\big(q_{\boldsymbol{\theta}^{(t',*)}}(\mathbf{W}) \| q_{\boldsymbol{\theta}^{(t')}}(\mathbf{W})\big) \tag{1}$$

where $\mathcal{E} = \{\mathbf{e}^{(t')}\}_{t'=1}^{t}$ is the set of task embeddings up to the current task, $\beta$ is the strength of the regularizer and $\boldsymbol{\theta}^{(t',*)} \equiv f_{\text{TC}}(\mathbf{e}^{(t',*)}, \boldsymbol{\psi}^*)$ are the parameters of the posterior approximation obtained from a checkpoint of the TC parameters before learning task $t$: $\{\boldsymbol{\psi}^*\} \cup \{\mathbf{e}^{(t',*)}\}_{t'<t}$. The checkpointed meta-model allows replaying posterior distributions from the past and retaining them via divergence minimization as detailed below, hence the name *posterior meta-replay*. Importantly, the loss on task $t$ only depends on the corresponding dataset $\mathcal{D}^{(t)}$, but knowledge transfer across tasks is possible because task-specific models are learned through a shared meta-model.[1] Since the computation required to compute this regularizer linearly scales with the number of tasks, we also explore stochastically regularizing on a subset of randomly selected tasks in each update (cf. SM D.3), and show that this does not impair performance. Notably, our *PosteriorReplay* method does not incur in a significant increase of runtime or memory usage (SM F.2).

The evaluation of Eq. 1 requires estimates of a divergence measure to prevent changes in learned posterior approximations of past tasks. Because these are required at every loss evaluation and need to be cheap to compute, we do not consider sample-based estimates but only estimates that directly utilize posterior parameters. This goal is easy to achieve for families of posterior approximations that possess an analytic expression for a divergence measure (e.g., Gaussian distributions). More specifically, for *PosteriorReplay-Exp*, we consider the forward KL, backward KL and the 2-Wasserstein distance but did not observe that the specific choice of divergence measure is crucial in practice (cf. SM C.3.1 and Table S14). In all other cases, approximations are required. Specifically, we resort in our experiments to the use of an L2 regularizer at the output of the TC network:

$$\beta \sum_{t' < t} \| f_{\text{TC}}(\mathbf{e}^{(t',*)}, \boldsymbol{\psi}^*) - f_{\text{TC}}(\mathbf{e}^{(t')}, \boldsymbol{\psi}) \|_2^2 \tag{2}$$

Perhaps surprisingly, we observe that this crude regularization, reminiscent to the one used in von Oswald et al. [91] for point estimates, does not harm performance and leads to models that exhibit

---

[1]While this type of transfer is rather implicit, explicit knowledge transfer can be realized via task-specific priors (cf. SM F.7).

virtually no forgetting. However, we discuss in SM F.3 how this isotropic regularization could be improved given that the KL is locally approximated by a norm $\|\cdot\|_F$ induced by the Fisher information matrix $F$ on $q_{\boldsymbol{\theta}}(\mathbf{W})$.

**Task inference.** A system with task-specific solutions requires access to task identity when processing unseen samples. In our framework, this amounts to selecting the correct task embedding to condition the TC. Although auxiliary systems can be used to infer task identity [24, 91], here we exploit predictive uncertainty, assuming task identity can be inferred from the input alone. For a task inference approach based on predictive uncertainty to work, the properties of the input data distribution $p^{(t)}(\boldsymbol{x})$ need to be reflected in the uncertainty measure, i.e., uncertainty needs to be low for in-distribution data and high for out-of-distribution (OOD) data [11]. Uncertainty-based task inference with deterministic discriminative models only captures aleatoric uncertainty, making its overall validity debatable. Indeed, aleatoric uncertainty is only calibrated in-distribution and its OOD behavior is hard to foresee. Instead, we argue that epistemic uncertainty arising naturally in a Bayesian setting is crucial for robust uncertainty-based task inference. In our case, epistemic uncertainty stems from the fact that the posterior parameter distribution $p(\mathbf{W}|\mathcal{D}^{(t)})$ in conjunction with the network architecture induces a distribution over functions. If this distribution captures a rich set of functions, a diversity of predictions on OOD data can be expected even if those functions agree in-distribution (cf. Sec. 4.2). Note, however, that inducing such diverse distribution over functions is not straightforward with neural networks, and that more research is required to justify uncertainty-based OOD detection [11].

We explore two different ways to quantify uncertainty for task inference. In **Ent** the task $t$ leading to the lowest entropy on the predictive posterior $p(\mathbf{y} \mid \mathcal{D}^{(t)}; \tilde{\mathbf{x}})$ is selected, where $p(\mathbf{y} \mid \mathcal{D}^{(t)}; \tilde{\mathbf{x}}) = \int_{\mathbf{W}} p(\mathbf{y} \mid \mathbf{W}; \tilde{\mathbf{x}}) p(\mathbf{W}|\mathcal{D}^{(t)}) \, d\mathbf{W}$ is approximated via Monte-Carlo with samples from $q_{\boldsymbol{\theta}^{(t)}}(\mathbf{W})$. This approach captures both aleatoric and epistemic uncertainty when used in a probabilistic setting. In **Agree** the task leading to the highest agreement in predictions across models drawn from $q_{\boldsymbol{\theta}^{(t)}}(\mathbf{W})$ is selected. This approach exclusively measures epistemic uncertainty and can therefore only be estimated in a probabilistic setting. Although we generally consider task inference for individual samples, we also explore batch-wise (**BW**) task inference for batches of 100 samples that are assumed to belong to the same task. Such approach drastically boosts task inference simply due to a statistical accumulation effect when having above chance level task inference for single inputs. Intuitively, *BW* corresponds to accumulating evidence to decrease uncertainty (e.g., an agent looking at an object from multiple perspectives). Further details can be found in SM C.6, and using uncertainty for task-boundary detection when training without explicit access to task identity is explored in SM D.8.

**Facilitating task inference through coresets.** A key advantage of Bayesian statistics is the ability to update models as new evidence arrives. When continually learning a sequence of tasks, posteriors may for example undergo a post-hoc fine-tuning on stored coresets to mitigate catastrophic forgetting of earlier tasks. Specifically, given a dataset split $\mathcal{D}^{(t)} \setminus \mathcal{C}^{(t)} \cup \mathcal{C}^{(t)}$, one can perform a final update $p(\mathbf{W}|\mathcal{D}^{(t)}) \propto p(\mathbf{W} \mid \mathcal{D}^{(t)} \setminus \mathcal{C}^{(t)}) p(\mathcal{C}^{(t)} \mid \mathbf{W})$ using a stored coreset $\mathcal{C}^{(t)}$ in conjunction with an already learned posterior approximation for $p(\mathbf{W} \mid \mathcal{D}^{(t)} \setminus \mathcal{C}^{(t)})$ that now acts as a prior. Interestingly, access to coresets after training on all tasks can also be exploited to facilitate task inference via predictive uncertainty. Here, we explore this idea by encouraging task-specific models to produce uncertain predictions for OOD samples (i.e., coresets from other tasks), an approach that we denote **CS** (SM C.7), and in which we store 100 inputs per task. Intriguingly, training on OOD inputs makes these become in-distribution, and therefore renders model agreement (*Agree*) inapplicable for task inference, which we empirically observe.

**Improving prior-focused CL.** We investigate ways to improve *PriorFocused* methods within our framework. First, we endow them with *implicit* posterior approximations $q_{\boldsymbol{\theta}}(\mathbf{W})$ parametrized by a WG hypernetwork, an approach we refer to as **PriorFocused-Imp**. Because the posterior is shared across tasks, no TC system is required and the parameters $\boldsymbol{\theta}$ can be directly optimized via SSGE. Second, we enrich *PriorFocused* methods with a small set of task-specific parameters that enable uncertainty-based task inference for prior-focused methods too. Specifically, the learned parameters $\mathbf{w}$ consist of a set of shared weights $\phi$ and a set of task-specific output heads with weights $\{\xi^{(t)}\}_{t=1}^{T}$. This approach is in contrast with how *PriorFocused* methods like *Online EWC* are commonly deployed in task-agnostic inference settings, where the softmax output grows as new tasks arrive (e.g., [88]). The use of a growing softmax causes the model class parametrized by $\mathbf{w}$ to change over time, and therefore violates the Bayesian assumption that the approximate posterior is obtained from a model class containing the ground-truth model. We show that this leads to limitations that

can be overcome by a multihead approach. For *Online EWC*, we refer to the growing softmax and multihead scenarios as **EWC-growing** and **EWC-multihead**, respectively (cf. SM C.5.2). We also explore the prior-focused instantiation of BbB, known as **VCL** (cf. SM C.5.1).

# 4 Experiments

In this section, we start by illustrating the conceptual advantage of the *PosteriorReplay* approach compared to *PriorFocused* methods, as well as the importance of parameter uncertainty for robust task inference. We then explore scalability to more challenging computer vision CL benchmarks.

To assess forgetting, we provide **During** scores, measured directly after training each task, and **Final** scores, evaluated after training on all tasks. We consider two different testing scenarios: (1) either task-identity is explicitly given (**TGiven**) or (2) task-identity has to be inferred (**TInfer**), e.g. via predictive uncertainty. Unless explicitly mentioned, task inference is performed for each sample in isolation and is obtained using the *Ent* criterion (*TInfer-Final*). Whenever the wrong task is inferred, the sample is directly considered as incorrectly classified. Supplementary results and controls are provided in SM D, and all experimental details can be found in SM E.[2]

## 4.1 Simple 1D regression illustrates the pitfalls of prior-focused learning

To illustrate conceptual differences between *PosteriorReplay* and *PriorFocused* methods we study 1D regression, for which the predictive posterior can be visualized. Each task-specific posterior obtained with *PosteriorReplay-Exp* fits the training data well (Fig. 3a) and exhibits increasing uncertainty when leaving the in-distribution domain, as desired for successful task inference. Interestingly, when studying low-dimensional problems, we generally found it easier to find viable hyperparameter configurations for *PosteriorReplay* with *implicit* methods than with *explicit* ones. As we do not consider a multihead for this problem, *PriorFocused* methods have to fit a single posterior to all polynomials in a sequential manner and struggle to find a good fit (Fig. 3b), independent of the type of

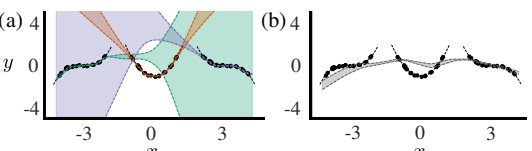

Figure 3: *PriorFocused* methods struggle to learn three 1D polynomial regression tasks. **(a)** Different colors represent the task-specific posterior approximations within the final *PosteriorReplay-Exp* model. For unseen inputs $x$ the posterior with the lowest predictive uncertainty is chosen to make predictions. **(b)** Predictive posterior using the final approximation of $p(\boldsymbol{W} \mid \mathcal{D}^{(1:3)})$, obtained via *PriorFocused-Imp*. Shaded areas represent standard deviation, and black dots training samples.

posterior approximation used. Results for other methods and an analysis of the correlations and multi-modality that can be captured by implicit methods in weight space can be found in SM D.1.

Because we use a mean squared error (MSE) loss, the likelihood is a Gaussian with fixed variance (SM C.3.1) and all $x$-dependent uncertainty originates from parameter uncertainty. We next consider classification problems where both epistemic and aleatoric uncertainty can be explicitly modelled.

## 4.2 Maintaining parameter uncertainty is crucial for robust task inference

To investigate the importance of parameter uncertainty, we consider a 2D classification problem for which uncertainty can be visualized in- and out-of-distribution. Classification tasks are of special interest as it is possible to model arbitrary input-dependent discrete distributions, e.g. via a softmax at the outputs. This surprisingly often results in meaningful OOD performance in high-dimensional benchmarks without any treatment of parameter uncertainty [82].

Table 1: 2D mode classification accuracies (Mean $\pm$ standard error of the mean (SEM) in %, $n = 10$). Task identity is inferred via predictive uncertainty using an entropy (*Ent*) or model agreement (*Agree*) criterion. *PR* denotes *PosteriorReplay*.

| Final Acc | TGiven | TInfer (Ent) | TInfer (Agree) |
|---|---|---|---|
| PR-Dirac | $99.78 \pm 0.21$ | $44.90 \pm 5.74$ | N/A |
| PR-Exp | $100.0 \pm 0.00$ | $81.07 \pm 6.78$ | $90.02 \pm 3.57$ |
| PR-Imp | $100.0 \pm 0.00$ | $100.0 \pm 0.00$ | $100.0 \pm 0.00$ |

---

[2]Source code for all experiments (including all baselines) is available at: `https://github.com/chrhenning/posterior_replay_cl`.

Here, we consider a Gaussian mixture of two modes per task, each mode being a different class (Fig. 4a). *TInfer-Final (Agree)*, which is indicative of the importance of epistemic uncertainty for OOD detection, is the most robust measure for task inference in this experiment (Table 1). *PosteriorReplay-Dirac*, which does not incorporate epistemic uncertainty, performs poorly. Finally, *implicit* methods maintain an advantage over *explicit* ones, presumably due to the increased flexibility in modelling the posterior.

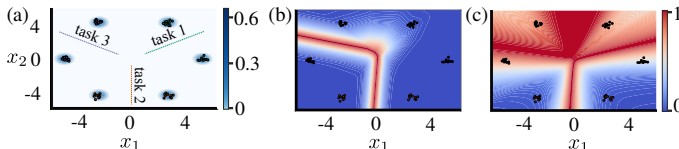

Figure 4: Parameter uncertainty is crucial for robust task inference. **(a)** Density of input distribution $p(\mathbf{x})$ across tasks. Dots represent training points, colors task-affiliation and lines decision boundaries for each of the three consecutively learned 2D binary classification tasks. **(b)** Entropy of predictive distribution induced by the approximate posterior of task 2 learned via *PosteriorReplay-Dirac*. **(c)** Same as (b) for *PosteriorReplay-Imp*.

To better understand why differences between methods arise we provide uncertainty maps over the input space (Fig. 4). Consistent with the observed low task inference accuracy, *PosteriorReplay-Dirac* displays arbitrary uncertainty away from the training data (Fig. 4b). By contrast, *PosteriorReplay-Imp* (Fig. 4c) approaches the desired behavior of displaying high uncertainty away from the training data of the corresponding task. We provide detailed analysis in SM D.2.

### 4.3 Multiple factors affect uncertainty-based task inference accuracy

To investigate the factors that affect uncertainty-based task inference, we next consider SplitMNIST [96], a popular variant of the MNIST dataset, adapted to CL by splitting the ten digit classes into five binary classification tasks. The results can be found in Table 2.

While all methods successfully prevent forgetting (i.e., *Final* scores are maxed-out and close to the *During* accuracies, SM D.3) and achieve acceptable *Final* accuracies when task identity is provided, large differences can be observed when the task needs to be inferred. Methods with task-specific solutions outperform by a large margin *PriorFocused* approaches such as *Online EWC*, whose performance substantially improves when using uncertainty-based task inference through a multihead. Despite superior performance of *PosteriorReplay* approaches, a gap in performance between task-inferred and task-given scenarios remains. However, training separate posteriors that are not embedded in a hypernetwork (**SeparatePosteriors**) leads to similar results, showing that task inference limitations are not linked to our solution. These limitations can be surmounted by inferring the task on batches rather than single samples (BW) or by using coresets to encourage high uncertainty for OOD data (CS), which leads to performances comparable to generative-replay methods which explicitly capture $p(\mathbf{X})$ (i.e., HNET+R and DGR).

Table 2: Accuracies of SplitMNIST experiments (Mean $\pm$ SEM in %, $n = 10$) after learning all tasks when task identity is provided (*TGiven-Final*) and when it needs to be inferred (*TInfer-Final*, based on the *Ent* criterion if explicit task-inference is required). Results are shown for an MLP with two hidden layers of 400 neurons (MLP-400,400), an MLP-100,100[1] or a Lenet[2]. *PR* denotes *PosteriorReplay* and *SP SeparatePosteriors*.

|  | TGiven-Final | TInfer-Final |
|---|---|---|
| EWC-growing [87] | N/A | $19.96 \pm 0.07$ |
| EWC-multihead | $96.40 \pm 0.62$ | $47.67 \pm 1.52$ |
| VCL-multihead | $96.45 \pm 0.13$ | $58.84 \pm 0.64$ |
| PR-Dirac | $99.65 \pm 0.01$ | $70.88 \pm 0.61$ |
| PR-Exp | $99.72 \pm 0.02$ | $71.73 \pm 0.87$ |
| PR-Imp | $99.77 \pm 0.01$ | $71.91 \pm 0.79$ |
| SP-Dirac | $99.77 \pm 0.01$ | $70.39 \pm 0.27$ |
| SP-Exp | $99.81 \pm 0.00$ | $68.40 \pm 0.23$ |
| PR-Exp-BW | $99.72 \pm 0.02$ | $99.72 \pm 0.02$ |
| PR-Exp-CS | $98.50 \pm 0.09$ | $90.83 \pm 0.24$ |
| DGR [87] | N/A | $91.79 \pm 0.32$ |
| HNET+R [91] | N/A | $95.30 \pm 0.13$ |
| PR-Dirac[1] | $99.72 \pm 0.01$ | $63.41 \pm 1.54$ |
| PR-Exp[1] | $99.75 \pm 0.01$ | $70.07 \pm 0.56$ |
| PR-Dirac[2] | $99.87 \pm 0.04$ | $72.33 \pm 2.75$ |
| PR-Exp[2] | $99.20 \pm 0.67$ | $74.09 \pm 1.38$ |

To better understand the factors that influence task inference, we consider a variety of approximate inference methods and architectures. Since epistemic uncertainty seems to play a vital role for task inference, the approximate inference method will likely affect *TInfer* performance (e.g., *Exp* vs. *Imp*; in supplementary results). In addition, because diversity in function space enables uncertainty-based OOD detection and because different architectures induce different priors in function space [93], one can expect that prior and architecture play a key role as well, which we observe

by comparing the *TInfer* performance for different architectures. Additional SplitMNIST results can be found in SM D.3, and results showing scalability to sequences of up to 100 PermutedMNIST tasks can be found in SM D.4 and D.5.

## 4.4 PosteriorReplay scales to natural image datasets

While BNNs are advocated because of their theoretical promises, practitioners are often put off by scalability issues. Here we show that our approach scales to natural images by considering SplitCIFAR-10 [39], a dataset consisting of five tasks with two classes each. Results obtained with a Resnet-32 [23] (Table 3) show performance gains in the task-agnostic setting compared to recent methods like EBM [50], and to the *PriorFocused* method VCL.

Furthermore, our results reveal considerable improvements through the incorporation of epistemic uncertainty, as shown by differences between *PosteriorReplay-Exp* and *PosteriorReplay-Dirac*.

Notably, *PosteriorReplay-Exp-BW* solves CIFAR-10 with a performance comparable to a classifier trained on all data at once, with the caveat that successive unseen samples are assumed to belong to the same task. In contrast to low-dimensional problems, the *implicit* method *PosteriorReplay-Imp* does not exhibit a competitive advantage, as it appears to suffer from scalability issues. Other baselines and results for a WRN-28-10 can be found in SM D.6, and results showing that our framework scales to the SplitCIFAR-100 benchmark can be found in SM D.7.

Table 3: Accuracies of SplitCIFAR-10 experiments (Mean $\pm$ SEM in %, $n = 10$). *TInfer-Final* is based on the *Ent* criterion if explicit task-inference is required and *PR* denotes *PosteriorReplay*.

|  | TGiven-During | TGiven-Final | TInfer-Final |
|---|---|---|---|
| VCL-multihead | $95.78 \pm 0.09$ | $61.09 \pm 0.54$ | $15.97 \pm 1.91$ |
| PR-Dirac | $94.59 \pm 0.10$ | $93.77 \pm 0.31$ | $54.83 \pm 0.79$ |
| PR-Exp | $95.59 \pm 0.08$ | $95.43 \pm 0.11$ | $61.90 \pm 0.66$ |
| PR-Imp | $94.25 \pm 0.07$ | $92.83 \pm 0.16$ | $51.95 \pm 0.53$ |
| PR-Exp-BW | $95.59 \pm 0.08$ | $95.43 \pm 0.11$ | $92.94 \pm 1.04$ |
| PR-Exp-CS | $95.15 \pm 0.11$ | $92.48 \pm 0.13$ | $64.76 \pm 0.34$ |
| EBM [50] | N/A | N/A | $38.84 \pm 1.08$ |

## 5 Discussion

In this study we propose *posterior meta-replay*, a framework for continually learning task-specific posterior approximations within a single shared meta-model. In contrast to *prior-focused* methods based on a recursive Bayesian update, our approach does not directly seek trade-off solutions across tasks. This results in more flexibility for learning new tasks but introduces the need to know task identity when processing unseen inputs.

**Task Inference.** Probabilistic inference on task identity can be achieved by additionally considering inputs and task embeddings as random variables, a strategy that would require task-conditioned generative models with tractable density [46]. However, learning generative models on high-dimensional data is a challenging problem and, even if tractable densities are accessible, these do not currently reflect the underlying data-generative process [64].[3] To sidestep these limitations, we study the use of predictive uncertainty for task inference [91] and show that an entropy-based criterion works best for both deterministic and Bayesian models. Nevertheless, we highlight that proper task inference requires epistemic uncertainty (e.g., measured in terms of model disagreement). Indeed, in-distribution samples with high aleatoric uncertainty can lead to high predictive entropy, causing them to be misclassified as OOD. This does not pose a problem in highly-curated ML datasets where samples with high aleatoric uncertainty are excluded [62], but drastically limits the applicability of entropy-based uncertainty estimation in more practical scenarios. For these reasons, we advocate for the use of Bayesian models whose epistemic uncertainty can induce diversity in function space for OOD inputs and enable more robust task inference.[4]

---

[3] Interestingly, a concurrent study by van de Ven et al. [89] successfully trains class-conditioned generative models, indicating that this approach could nevertheless be feasible to tackle task inference.

[4] Note, while also models with deterministic parameters may be well suited for OOD detection (e.g., Lakshminarayanan et al. [43], that utilizes a deterministic distance preserving input-to-hidden mapping), these solutions

**Limitations.** Compared to methods performing deterministic inference, the Bayesian model average incurs in significant computational overhead. This overhead is reinforced when performing uncertainty-based task inference, since each predictive posterior needs to be evaluated in parallel. Moreover, despite strong performance gains compared to *prior-focused* approaches, we observe general limitations of such task inference procedure. These could be overcome once a better understanding of how epistemic uncertainty influences OOD behavior in neural networks is available. In addition, our work builds on algorithms to perform variational inference, and is therefore only applicable to problems where these can be successfully deployed. Finally, all our experiments consist of a set of clearly defined tasks within which i.i.d. samples are available. Although this scenario is in line with most existing CL literature, it might be of limited relevance for practical CL problems, and a focus on established benchmarks adhering to these constraints could therefore misguide research on this area. Indeed, a more natural CL problem might arise from the need to online learn from a stream of autocorrelated samples. In this context, it is important to note that unlike non-Bayesian CL methods, our approach can utilize any type of online *prior-focused* method (such as FOO-VB [97]) to also learn *within* tasks in a non-i.i.d. manner. Therefore, as long as some coarse split into tasks is meaningful, such hierarchical approach holds great promise. However, it should be noted that any progress towards learning from non-i.i.d. data opens the door to training algorithms from raw, uncurated datasets, and could therefore counter some of the efforts that are currently done to mitigate algorithmic bias.

**Conclusion.** Taken together, our work shows that it is possible to continually learn an approximate posterior per task without an increased parameter budget, and that task-agnostic inference can be achieved via predictive uncertainty to obtain a Bayesian CL approach that is scalable to real-world data. Since forgetting is not a prevalent issue in our experiments and task inference limitations are not linked to our CL solution, progress in the field of uncertainty-based OOD detection will automatically translate into further improvements of our method.

## Acknowledgements

This work was supported by the Swiss National Science Foundation (B.F.G. CRSII5-173721 and 315230_189251), ETH project funding (B.F.G. ETH-20 19-01) and funding from the Swiss Data Science Center (B.F.G, C17-18, J. v. O. P18-03). J.S. was supported by an Ambizione grant (PZ00P3_186027) from the Swiss National Science Foundation. We are grateful for discussions with Harald Dermutz, Simone Carlo Surace and Jean-Pascal Pfister. We also would like to thank Sebastian Farquhar for discussions on Radial posteriors and for proofreading our implementation of this method.

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
