# Supplementary Material:
# Posterior Meta-Replay for Continual Learning

**Christian Henning\*, Maria R. Cervera\*, Francesco D'Angelo, Johannes von Oswald, Regina Traber, Benjamin Ehret, Seijin Kobayashi, Benjamin F. Grewe, João Sacramento**

## Table of Contents

# A  Acronyms

For clarity, we provide a list of acronyms used throughout the paper in Table S1.

Table S1: List of acronyms.

| | |
|---|---|
| ML | Machine learning |
| CL | Continual learning |
| MLP | Multilayer perceptron |
| SEM | Standard error of the mean |
| SD | Standard deviation |
| MSE | Mean-squared error |
| NLL | Negative log-likelihood |
| BNN | Bayesian neural network |
| VI | Variational inference |
| ELBO | Evidence lower-bound |
| RKL/FKL | Reverse/Forward Kullback-Leibler divergence |
| W2 | 2-Wasserstein distance |
| MC | Monte Carlo |
| AVB | Adversarial variational Bayes |
| BBB | Bayes-by-Backprop |
| EWC | Elastic weight consolidation |
| SSGE | Spectral Stein gradient estimator |
| VCL | Variational continual learning |
| PF | Prior-focused |
| PR | Posterior meta-replay |
| M | Main network |
| WG | Weight generator |
| TC | Task-conditioning network |
| BW | Task inference using batches |
| CS | Final fine-tuning using coresets |
| SP | Separate posteriors per task |
| OOD | Out of distribution |
| TGIVEN | Task identity is given |
| TINFER | Task identity is inferred |
| ENT | Task inference via minimum entropy |
| CONF | Task inference via maximum confidence |
| AGREE | Task inference via maximum model agreement |

# B  Summary of Notation

We summarize here the mathematical notation used throughout the paper and reintroduce the role of important variables. Whenever applicable, we denote random variables by capital letters, random variates by lowercase letters and sample spaces using calligraphic font, e.g., the prior density of $w \in \mathcal{W}$ is $p(W = w)$. In general, sets use calligraphic font. Vectors are assumed to be column vectors and highlighted as bold symbols. Indexing is performed using subscripts, e.g., $x_i$ is the $i$-th component of $\mathbf{x} \in \mathcal{X}$. Superscripts are used to disambiguate external factors such as task identity. Matrices also use capital letters, and it is clearly stated whenever a variable has to be interpreted as matrix, e.g., the *Fisher information matrix* $F$.

In this study, we distinguish between three different networks with clearly defined roles (plus a *discriminator* network when using AVB). The *main network* (M), $\hat{\mathbf{y}} = f_{\mathrm{M}}(\mathbf{x}, \mathbf{w})$, processes inputs $\mathbf{x}$ to generate predictions $\hat{\mathbf{y}}$ using trainable parameters $\mathbf{w}$. This notation is chosen for mathematical convenience, but it should be noted that in most cases $f_{\mathrm{M}}(\cdot)$ represents a likelihood function. For instance, in the case of a $C$-way classification with labels $y \in \mathcal{Y} = \{1, \ldots, C\}$, the output of the main network is an element of the probability simplex $\Delta(\mathcal{Y})$, and in regression $f_{\mathrm{M}}(\cdot)$ represents the mean of a normal distribution $\mathcal{N}\big(f_{\mathrm{M}}(\cdot), \sigma_{\mathrm{ll}}^2\big)$ with variance $\sigma_{\mathrm{ll}}^2$. The *task-conditioned hypernetwork* (TC), $\boldsymbol{\theta}^{(t)} = f_{\mathrm{TC}}(\mathbf{e}^{(t)}, \boldsymbol{\psi})$, has parameters $\boldsymbol{\psi}$ and processes task embeddings $\mathbf{e}^{(t)}$ to generate the

parameters $\boldsymbol{\theta}^{(t)}$ of the *weight generator* (WG). In the deterministic case (e.g., *PosteriorReplay-Dirac*), the weight generator simply reduces to $\mathbf{w}^{(t)} \equiv \boldsymbol{\theta}^{(t)} = f_{\text{WG}}(\cdot, \boldsymbol{\theta}^{(t)})$. In all other cases, it transforms noise samples $\mathbf{z} \sim p(\mathbf{Z})$ into main network parameter configurations.

A dataset is a set of input-output tuples $\mathcal{D} = \{(\mathbf{x}^{(n)}, \mathbf{y}^{(n)})\}_{n=1}^{N}$. If not noted otherwise, a dataset is considered an i.i.d. sample from some unknown data-generating process $\mathcal{D} \overset{i.i.d.}{\sim} p(\mathbf{X})p(\mathbf{Y} \mid \mathbf{X})$.

We consider a Bayesian treatment of main network parameters $\mathbf{w}$ to incorporate parameter uncertainty [56]. The prior is denoted by $p(\mathbf{W})$, the likelihood by $p(\mathcal{D} \mid \mathbf{W}) \equiv \prod_{n=1}^{N} p(\mathbf{y}^{(n)} \mid \mathbf{W}; \mathbf{x}^{(n)})$, the posterior parameter distribution by $p(\mathbf{W} \mid \mathcal{D})$ and the posterior predictive distribution for an unseen input $\tilde{\mathbf{x}}$ by $p(\mathbf{Y} \mid \mathcal{D}; \tilde{\mathbf{x}})$. We consider families of distributions parametrized by $\boldsymbol{\theta}$ to approximate the posterior parameter distribution $q_{\boldsymbol{\theta}}(\mathbf{W}) \approx p(\mathbf{W} \mid \mathcal{D})$, e.g., for the deterministic case we have $q_{\boldsymbol{\theta}}(\mathbf{W}) = \delta(\boldsymbol{\theta} - \mathbf{W})$, where $\delta(\cdot)$ denotes the Dirac delta function.

# C    Algorithms

In this section we provide details about the different algorithms used throughout the paper. We start by quickly reviewing variational inference, and explain how it can be applied to task-conditioned hypernetworks in the context of CL in order to obtain task-specific approximate posterior distributions. Then, we present several variational algorithms that we employed to obtain the approximate posteriors, either based on a predefined function (*explicit* posterior) or based on a parametrization by an auxiliary network (*implicit* posterior). We also explain how *prior-focused* CL can be achieved within our framework, and how it can be rendered more flexible by allowing the parameter posteriors to be approximated by *implicit* distributions, and by incorporating a set of task-specific parameters alongside the shared ones. Because task-specific solutions require having access to the identity of the task, we explain how task identity can be inferred via predictive uncertainty for both *prior-focused* and *posterior meta-replay* approaches. Finally, we explain how coresets can be used in our framework to perform a fine-tuning stage after training which mitigates forgetting and facilitates task inference.

## C.1    Variational inference

Whenever confronted with unseen inputs $\tilde{\mathbf{x}}$, we aspire to obtain predictions via the posterior predictive distribution: $p(\mathbf{y} \mid \mathcal{D}; \tilde{\mathbf{x}}) = \int_{\mathbf{W}} p(\mathbf{y} \mid \mathbf{W}; \tilde{\mathbf{x}})p(\mathbf{W}|\mathcal{D}) \, d\mathbf{W}$. Unfortunately, the posterior parameter distribution $p(\mathbf{W}|\mathcal{D})$ is in general intractable. Furthermore, since sampling cannot be efficiently performed and would require high storage demands, we need to approximate this distribution in order to evaluate the posterior predictive distribution. For this purpose, we apply variational inference (VI), a standard approximate probabilistic inference technique (see e.g., Blei et al. [5]) where we look for approximations $q_{\boldsymbol{\theta}}(\mathbf{W})$ within a certain family of distributions $\mathcal{Q}$, referred to as the *variational family*, with members parametrized by $\boldsymbol{\theta} \in \Theta$. In VI, this problem is solved by optimizing for nearest approximations within $\mathcal{Q}$ according to some divergence, most often the Kullback-Leibler (KL) divergence: $\text{KL}(q_{\boldsymbol{\theta}}(\mathbf{W}) \,||\, p(\mathbf{W} \mid \mathcal{D}))$. The choice of the reverse KL allows rewriting the above expression such that it does not contain the intractable posterior. The resulting expression (known as the evidence lower bound; ELBO) is therefore a suitable optimization objective that needs to be maximized:

$$\mathbb{E}_{q_{\boldsymbol{\theta}}(\mathbf{W})}\big[\log p(\mathcal{D} \mid \mathbf{W})\big] - \text{KL}(q_{\boldsymbol{\theta}}(\mathbf{W}) \,||\, p(\mathbf{W})) \equiv -\mathcal{L}_{\text{task}}(\mathcal{D}) \tag{3}$$

where the first term corresponds to minimizing the negative log-likelihood (NLL) of the data $\mathcal{D}$ (cf. SM C.3.1), and the second term can be seen as a regularization that aims to match the approximation $q_{\boldsymbol{\theta}}(\mathbf{W})$ to the prior $p(\mathbf{W})$, and is referred to as *prior-matching term*. Finding an optimal approximation within the variational family amounts to optimizing the parameters $\boldsymbol{\theta}$. Since in our framework these are task-specific and their optimization depends only on the corresponding dataset $\mathcal{D}^{(t)}$ we write from here on $q_{\boldsymbol{\theta}^{(t)}}(\mathbf{W})$.

## C.2    Task-conditioned hypernetworks

**Task-conditioned hypernetworks**, which generate task-specific weights for a main network, have recently been proposed as an effective method to continually learn several tasks [91]. The outputs of the hypernetwork are made task-specific by providing low-dimensional task embeddings $\mathbf{e}^{(t)}$ as input,

which are learned continually alongside the hypernetwork's parameters $\psi$. Catastrophic forgetting at this meta level can be prevented using a simple L2 regularizer (cf. Eq. 2 in the main text) which makes sure that the outputs of the hypernetwork for previously learned tasks do not change. When learning task $t$ the loss thus becomes:

$$\mathcal{L}^{(t)}(\psi, \mathcal{E}, \mathcal{D}^{(t)}) = \mathcal{L}_{\text{task}}(\psi, \mathbf{e}^{(t)}, \mathcal{D}^{(t)}) + \beta \sum_{t' < t} \| f_{\text{TC}}(\mathbf{e}^{(t', *)}, \psi^*) - f_{\text{TC}}(\mathbf{e}^{(t')}, \psi) \|^2 \qquad (4)$$

Notably, shifting CL to the meta-level simplifies the problem considerably, because only a single input-output mapping per task needs to be fixed.

**Chunking.** Here we consider hypernetworks that parameterize target models in a compressed form using a simple but effective technique called *chunking*: the hypernetwork is iteratively invoked using an additional input $\mathbf{c}$ at each step, which addresses a distinct subset of parameters (e.g., a distinct layer of the target neural network). To be precise, chunk embeddings $\mathbf{c}^{(l)}$, $l = 1..L$, are unconditional parameters (i.e., not task-specific). Thus, the hypernetwork parameters $\psi$ can be split into a set of chunk embeddings $\{\mathbf{c}^{(l)}\}_{l=1}^{L}$ and a set of weights $\tilde{\psi}$, which are the actual weights of the network that produces individual chunks. This introduces soft weight sharing, as the entire set of target model parameters $\mathbf{w}$ depends on $\tilde{\psi}$. Note, that modern graphics hardware allows the parallel generation of all chunks (batch-processing). For more details, please refer to von Oswald et al. [91] and Ehret et al. [15]. For all experiments other than the low-dimensional toy problems, we apply the chunking strategy to the TC and WG networks. Low-dimensional problems utilize MLP hypernetworks, where $\mathbf{w}$ is directly the output of the network. Both task and chunk embeddings are initialized according to a normal distribution with zero mean, whose variance is considered a hyperparameter.

**Probabilistic extension of task-conditioned hypernetworks.** In their original formulation, hypernetworks for CL provide a single main network weight configuration per task (*PosteriorReplay-Dirac*). Here, we extend this deterministic approach to a probabilistic setting and aim to model task-specific *distributions* over main network weights instead. Therefore, the outputs of the hypernetwork can no longer be interpreted as the weights $\mathbf{w}$ of the main network, but rather as parameters $\boldsymbol{\theta}^{(t)}$ defining the approximate distribution $q_{\boldsymbol{\theta}^{(t)}}(\mathbf{W})$, from which weights $\mathbf{w}$ for the main network can be sampled. In practice, $q_{\boldsymbol{\theta}^{(t)}}(\mathbf{W})$ is given by a function $f_{\text{WG}}(\mathbf{z}, \boldsymbol{\theta}^{(t)})$, which also depends on a base distribution $p(\mathbf{Z})$ from which inputs $\mathbf{z}$ for the hypernetwork are sampled.

**Meta-regularizing in distribution space.** Crucially, in this probabilistic setting, catastrophic forgetting is avoided via a regularizer that prevents the distributions $q_{\boldsymbol{\theta}^{(t)}}(\mathbf{W})$ of previously learned tasks to change, hence our naming *posterior meta-replay*. Whenever the utilized variational family for approximate posteriors has an analytic expression for a divergence measure, this can be achieved by turning the original hypernetwork regularizer into a divergence measure between the distributions (cf. Eq. 1) before and after any given task is being learned. However divergence measures cannot always be analytically evaluated, for example if the distributions are *implicit*, in which case other solutions are necessary. One option is to use a sample-based distance estimate between the distributions [21, 53], which would act upon the output of the WG network (i.e., the weight samples). However, due to the high dimensionality of $\mathbf{w}$, meaningful distance estimates in distribution space might be prohibitively expensive since the regularizer has to be evaluated in every training iteration. For this reason, whenever no analytic divergence measure is available, we resort to the use of a mean-squared error (MSE) regularizer at the output of the TC network (Eq. 2), as done in the deterministic case considered by von Oswald et al. [91]. Therefore, rather than ensuring that a distribution does not change, we ensure that the parameters of a distribution do not change. Note however, that this treatment forces an interpretation onto the TC network as encoding a Gaussian likelihood with isotropic variance.

While in our experiments forgetting is not a prevailing issue and we therefore were not urged to improve upon this simple regularization, we would still like to comment on potential improvements that can be considered by future work. One option would be to assign importance values to individual outputs of the TC network and thus transform the isotropic regularization of Eq. 2 into a weighted sum per previous task. Such importance values could incorporate the curvature of the loss landscape. As the task-specific loss is data-dependent, the importance values would need to be computed at the end of the training of each task, where data is still accessible. In our case, the task-specific loss is the ELBO (Eq. 3), which embodies the KL to the posterior $p(\mathcal{D} \mid \mathbf{W})$. Thus, if importance values reflect the Hessian of the ELBO with respect to the outputs of the TC network, the CL regularization

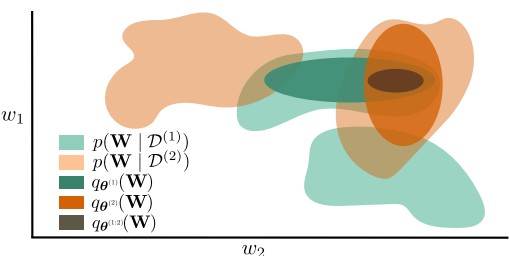

Figure S1: Differences between *prior-focused* and *posterior-replay* CL approaches when using Gaussian posterior approximations with diagonal covariance. In this case, the proposed *posterior meta-replay* framework can capture one mode of each task-specific posterior via $q_{\boldsymbol{\theta}^{(1)}}(\mathbf{W})$ and $q_{\boldsymbol{\theta}^{(2)}}(\mathbf{W})$. In contrast, a *prior-focused* method with this type of approximation (e.g., EWC [38] or VCL [65]) has to assume that the currently found mode $q_{\boldsymbol{\theta}^{(1:2)}}(\mathbf{W})$ also contains admissible solutions for upcoming tasks.

would target our posterior approximation criterion (i.e., the ELBO) directly, ensuring that parameter changes are only tolerated if the ELBO remains stable. In SM. F.3 we further elaborate on how importance values could be constructed such that this simple type of regularization can be interpreted as preserving distances in distribution space. Nonetheless, an imminent drawback of this procedure is the need to store the importance values of each previous task.[5] However, the memory implications of this shortcoming can be alleviated by using compression schemes for importance values or by distilling them into an auxiliary network [28].

### C.3    Posterior-Replay with explicit parametric distributions

We now present the algorithms that can be used to obtain what we refer to as *explicit* approximate posterior distributions, i.e., simple posterior approximations $q_{\boldsymbol{\theta}^{(t)}}(\mathbf{W})$ that can be sampled from according to a predefined function (cf. Eq. 5 and Eq. 12). Although in the main text only results obtained for BbB are reported (noted *PosteriorReplay-Exp*), we consider here additional *explicit* methods and therefore denote each individual method as *PosteriorReplay-\<method\>*.

### C.3.1    Bayes-by-Backprop

Blundell et al. [6] present Bayes-by-Backprop (BbB), a VI algorithm that uses a mean-field Gaussian approximation, i.e., $q_{\boldsymbol{\theta}}(\mathbf{W}) = \prod_i \mathcal{N}(W_i; \mu_i, \sigma_i^2)$, where $\boldsymbol{\theta}$ consists of vectors $\boldsymbol{\mu}$ and $\boldsymbol{\sigma}^2$ containing a mean and variance for each weight. To be able to optimize these two vectors, the authors make use of the *reparametrization trick* [36], which allows gradient computation with respect to $\boldsymbol{\mu}$ and $\boldsymbol{\sigma}$ through the stochastic sampling process of $q_{\boldsymbol{\theta}}(\mathbf{W})$. Specifically, samples $\mathbf{w}$ are obtained via:

$$\mathbf{w} = \boldsymbol{\mu} + \boldsymbol{\sigma} \odot \boldsymbol{\epsilon} \tag{5}$$

where $\boldsymbol{\epsilon} \sim \mathcal{N}(0, \mathbf{I})$.

Recall, that optimizing the ELBO (cf. Eq. 3) requires estimating the NLL $-\mathbb{E}_{q_{\boldsymbol{\theta}}(\mathbf{W})}[\log p(\mathcal{D} \mid \mathbf{W})]$, which in practice is achieved via a Monte-Carlo (MC) estimate of $K$ samples $\mathbf{w}^{(k)} \sim q_{\boldsymbol{\theta}}(\mathbf{W})$:

$$\text{NLL} \approx -\frac{1}{K} \sum_{k=1}^{K} \log p(\mathcal{D} \mid \mathbf{W} = \mathbf{w}^{(k)}) \tag{6}$$

As a full summation $\log p(\mathcal{D} \mid \mathbf{W}) = \sum_{n=1}^{N} \log p(\mathbf{y}^{(n)} \mid \mathbf{W}; \mathbf{x}^{(n)})$ over the whole dataset at every training iteration is prohibitively expensive, this term is approximated via mini-batches $\mathcal{B}$ of size $N_{\text{mb}}$, which requires a corrective scaling:

$$\text{NLL} \approx -\frac{1}{K} \sum_{k=1}^{K} \frac{N}{N_{\text{mb}}} \sum_{(\mathbf{x},\mathbf{y}) \in \mathcal{B}} \log p(\mathbf{y} \mid \mathbf{w}^{(k)}; \mathbf{x}) \tag{7}$$

---

[5]Note, that also the original EWC formulation required the storage of one Fisher matrix per task [38].

How to compute the term $\log p(\mathbf{y} \mid \mathbf{w}; \mathbf{x})$ depends on the problem at hand. In regression tasks, it is common to model the likelihood as a Gaussian distribution. For simplicity, we only consider 1D regression and a model likelihood with fixed variance $\sigma_{\text{ll}}^2$ such that:

$$\log p(y \mid \mathbf{w}; x) = \text{const} - \frac{1}{2\sigma_{\text{ll}}^2} \left( f_{\text{M}}(x, \mathbf{w}) - y \right)^2 \tag{8}$$

Dropping all constant terms that do not affect the optimization, the NLL can be written as a properly scaled ($\frac{N}{2\sigma_{\text{ll}}^2}$) MSE loss inside an MC estimate:

$$\text{NLL} \approx \frac{N}{2KN_{\text{mb}}\sigma_{\text{ll}}^2} \sum_{k=1}^{K} \sum_{(x,y)\in\mathcal{B}} \left( f_{\text{M}}(x, \mathbf{w}^{(k)}) - y \right)^2 \tag{9}$$

In $C$-way classification problems, the likelihood of class $c$ is computed as:

$$p(Y = c \mid \mathbf{w}; \mathbf{x}) = \text{sm}\big(f_{\text{M}}(\mathbf{x}, \mathbf{w})\big)_c \tag{10}$$

where $\text{sm}(\cdot)$ refers to the softmax, assuming the main network produces unnormalized logits for mathematical convenience. Under this notation, the NLL for classification problems can be estimated as follows:

$$
\begin{aligned}
\text{NLL} &\approx -\frac{N}{KN_{\text{mb}}} \sum_{k=1}^{K} \sum_{(\mathbf{x},y)\in\mathcal{B}} \log\left( \text{sm}\big(f_{\text{M}}(\mathbf{x}, \mathbf{w}^{(k)})\big)_y \right) \\
&= \frac{N}{KN_{\text{mb}}} \sum_{k=1}^{K} \sum_{(\mathbf{x},y)\in\mathcal{B}} \bigg( \\
&\quad \underbrace{-\sum_{c=1}^{C} [c=y] \log\left( \text{sm}\big(f_{\text{M}}(\mathbf{x}, \mathbf{w}^{(k)})\big)_c \right)}_{\text{cross-entropy loss with 1-hot targets}} \bigg)
\end{aligned} \tag{11}
$$

where $[\cdot]$ denotes the Iverson bracket. The second term in the ELBO, i.e., the *prior-matching term*, can be analytically evaluated if a Gaussian prior is used.

We adapt the BbB algorithm to our *posterior meta-replay* framework by having a task-conditioned (TC) network that generates task-specific $\boldsymbol{\mu}^{(t)}$ and $\boldsymbol{\sigma}^{(t)}$ (Fig. S1), an approach we denote **PosteriorReplay-BbB** (*PosteriorReplay-Exp* in the main text). Since outputs of the TC network are real-valued, variances $\boldsymbol{\sigma}^{(t)}$ are obtained through a softplus transformation of the network's outputs. The current task's loss $\mathcal{L}_{\text{task}}(\boldsymbol{\psi}, \mathbf{e}^{(t)}, \mathcal{D}^{(t)})$ corresponds to the negative ELBO and can be estimated as described above, where we use the analytic expression for the *prior-matching term*.

Since approximate posteriors in BbB correspond to Gaussian distributions, the CL regularizer can take the form of an explicit divergence measure in distribution space computed solely based on the TC network's output. We experiment with the forward (FKL) and reverse KL (RKL), and the 2-Wasserstein distance (W2). The loss when learning task $t$ therefore becomes Eq. 1, where $\text{D}(\cdot||\cdot)$ corresponds to one of the above mentioned divergence measures. In practice, we do not observe a notable difference between any of these divergence measures and the L2 regularization in Eq. 2.

As a variance reduction trick to improve training stability, we also experiment with the *local reparametrization trick* [37] whenever the main network has an MLP architecture. Whether or not this trick is used is determined by a hyperparameter and therefore selected by the hyperparameter search of each experiment conducted with BbB or VCL (cf. SM C.5.1).

### C.3.2 Radial Posteriors

As an alternative method to obtain *explicit* posterior approximations we also experimented with *radial posteriors* [18], which we briefly describe below.

Intuitively, one would expect that the probability mass of a Gaussian lies around the mean. This is however not the case in high dimensions, where the probability mass is clustered in a thin hyper-annulus far away from the mean. This happens because a sample point from an isotropic, high-dimensional Gaussian distribution can be interpreted as many sample points from the corresponding 1D Gaussian distribution, therewith representing with high probability an element of the typical set. Farquhar et al. [18] argues that training BNNs with a mean-field Gaussian approximation (as in BbB), can lead to gradient estimates with high variance and impaired training stability due to the effects that arise due to typicality in isotropic, high-dimensional Gaussian distributions. Based on this insight, Farquhar et al. [18] propose the following corrective normalization to Eq. 5, and argue it helps recover the intuitive behavior of Gaussian distributions in low dimensions:

$$\mathbf{w} = \boldsymbol{\mu} + \boldsymbol{\sigma} \odot \frac{\boldsymbol{\epsilon}}{\|\boldsymbol{\epsilon}\|} \cdot r \tag{12}$$

where $r \sim \mathcal{N}(0, 1)$ and $\boldsymbol{\epsilon} \sim \mathcal{N}(0, I)$. Here, we apply the corrective normalization layer-wise, but treating weights and biases separately, such that the dimensions of $I$ correspond to the number of weights or biases in a given layer.

When computing the *prior-matching term* between a *radial* approximation and a Gaussian prior $p(\mathbf{W})$, we use the following expression for the negative entropy of a *radial* distribution (cf. Eq. 5 in Farquhar et al. [18]):

$$\int q_{\boldsymbol{\theta}}(\mathbf{W}) \log q_{\boldsymbol{\theta}}(\mathbf{W}) d\mathbf{W} = -\sum_i \log \sigma_i + \text{const} \tag{13}$$

The cross-entropy term is approximated via an MC estimate with samples from the *radial* posterior. Note, that the log-density of the Gaussian prior can be computed analytically.

We use *radial* posteriors within our framework (**PosteriorReplay-Radial**), where task-specific means $\boldsymbol{\mu}$ and "variances" $\boldsymbol{\sigma}^2$ are generated by a TC hypernetwork, whose outputs need to be regularized to prevent forgetting (cf. SM C.3.1). Because an analytic expression for a divergence between *radial* distributions is unknown, we resort to the L2 hypernetwork regularizer (Eq. 2) when working with *radial* distributions.

## C.4 Posterior-Replay with implicit distributions

We also explore the use of *implicit* $q_{\boldsymbol{\theta}^{(t)}}(\mathbf{W})$ distributions as approximate posteriors. In our framework, these are parametrized by an auxiliary WG network, that has parameters $\boldsymbol{\theta}^{(t)}$ and receives samples from an arbitrary base distribution $p(\mathbf{Z})$ as inputs. In our experiments, we always consider a Gaussian base distribution with zero mean, and experiment with different variances.

The use of implicit posterior approximations introduces a challenge when optimizing the *prior-matching* term of the ELBO, since we do not have access to the analytic expression of the density, nor to its entropy. This problem can be avoided when the change-of-variables formula is applicable, i.e., when using an invertible architecture for the weight-generator network WG with tractable base distribution $p(\mathbf{Z})$ (cf. normalizing flows [70]). In this case, the ELBO objective can be approximated via an MC estimate and optimized via automatic differentiation.

Alternative approaches sidestep the need to have invertible networks, and aim to find other estimates that allow optimizing the ELBO objective. Multiple studies have already investigated the use of weight generators for BNNs [14, 27, 34, 35, 40], ranging from sample-based estimates of the *prior-matching term* [73] to the use of normalizing flows with shared influence on a set of weights for improved scalability [55]. Within the deep learning community, generative adversarial networks (GANs, Goodfellow et al. [19]) represent the most successful use case of implicit distributions. This approach is purely sample-based and requires an auxiliary network that engages in a minimax optimization. A wide range of loss functions can be used to approximately optimize different kinds of divergences or distances, including the KL required for the *prior-matching term*, [67] as done in AVB (cf. SM C.4.1). However, as training corresponds to playing a non-convex game and an inner-loop optimization is required, optimization difficulties arise when applying GAN-like approaches to high-dimensional problems. We experienced these difficulties and therefore also explore alternative ways to train implicit distributions, e.g., methods based on Stein's identity such as SSGE (cf. SM C.4.2). Although in the main text only results obtained using SSGE are reported (noted *PosteriorReplay-Imp*), we consider here additional *implicit* methods and therefore denote each individual method as *PosteriorReplay-<method>*.

In practice, approaches that do not require invertible networks are used with architectures that have a support of measure zero in weight space $\mathcal{W}$, which causes the KL to be ill-defined. We comment on this issue in SM C.4.3.

### C.4.1 Adversarial Variational Bayes

Adversarial variational Bayes (AVB, Mescheder et al. [61]) was introduced as a method to estimate the log-density ratio of the *prior-matching term* by using the GAN framework. We denote using AVB to find approximate *implicit* posteriors within our *posterior meta-replay* framework as **PosteriorReplay-AVB**.

Given that the *prior-matching term* (Eq. 3) can be rewritten as $\mathbb{E}_{q_{\boldsymbol{\theta}}(\mathbf{W})}\big[\log q_{\boldsymbol{\theta}}(\mathbf{W}) - \log p(\mathbf{W})\big]$, AVB introduces an auxiliary *discriminator* network (**D**) that, within each training iteration, learns to approximate $\log q_{\boldsymbol{\theta}}(\mathbf{W}) - \log p(\mathbf{W})$. Mescheder et al. [61] show that this is achieved for a discriminator that maximizes the following expression:

$$\mathbb{E}_{q_{\boldsymbol{\theta}}(\mathbf{W})}[\log \sigma(f_{\mathrm{D}}(\mathbf{W}))] + \mathbb{E}_{p(\mathbf{W})}\left[\log(1 - \sigma(f_{\mathrm{D}}(\mathbf{W})))\right] \tag{14}$$

where $\sigma(\cdot)$ denotes the logistic sigmoid function and $f_{\mathrm{D}}$ the function performed by the discriminator. Having the optimal discriminator $f_{\mathrm{D}}^*(\mathbf{w})$, the *prior-matching term* can be approximated via an MC sample:

$$\mathbb{E}_{q_{\boldsymbol{\theta}}(\mathbf{W})}\Big[\log \frac{q_{\boldsymbol{\theta}}(\mathbf{W})}{p(\mathbf{W})}\Big] = \mathbb{E}_{q_{\boldsymbol{\theta}}(\mathbf{W})}\big[f_{\mathrm{D}}^*(\mathbf{W})\big]$$

$$\approx \frac{1}{K}\sum_{k=1}^{K} f_{\mathrm{D}}^*(\mathbf{w}^{(k)}) \tag{15}$$

This means that at every training iteration of $\boldsymbol{\theta}$ (or in our case $\boldsymbol{\psi}$) the parameters of the discriminator should be trained to optimality. In practice, however, discriminator weights are only fine-tuned for a few iterations in the inner loop.

Note that training requires access to $\nabla_{\boldsymbol{\theta}} \mathbb{E}_{q_{\boldsymbol{\theta}}(\mathbf{W})}\big[f_{\mathrm{D}}^*(\mathbf{W})\big]$, and that the optimal parameter configuration of the discriminator might also depend on $\boldsymbol{\theta}$ as it is an outcome of the optimization procedure described in Eq. 14. Fortunately, as shown in Mescheder et al. [61], the term $\mathbb{E}_{q_{\boldsymbol{\theta}}(\mathbf{W})}\big[\nabla_{\boldsymbol{\theta}} f_{\mathrm{D}}^*(\mathbf{W})\big]$ vanishes and no backpropagation through the discriminator is required.

We also employ a trick suggested in Mescheder et al. [61], termed *adaptive contrast*, which can be used whenever analytic access to the prior density is guaranteed (which is not the case when using AVB in a prior-focused setting, **PriorFocused-AVB**, except for the first task). The incentive for the trick is the fact that a density ratio, especially in high-dimensions, has high variance. Therefore, an auxiliary Gaussian distribution $r_{\boldsymbol{\alpha}}(\mathbf{W})$ is introduced, whose mean and variance parameters are set to the empirical mean and variance of $q_{\boldsymbol{\theta}}(\mathbf{W})$, assuming that the ratio when involving such $r_{\boldsymbol{\alpha}}(\mathbf{W})$ is "easier" to estimate. The ELBO is then rewritten in the following way to include $r_{\boldsymbol{\alpha}}(\mathbf{W})$:

$$\begin{aligned}
&\mathbb{E}_{q_{\boldsymbol{\theta}}(\mathbf{W})}\big[\log p(\mathcal{D} \mid \mathbf{W})\big] - \mathrm{KL}(q_{\boldsymbol{\theta}}(\mathbf{W}) \,\|\, p(\mathbf{W})) \\
=&\,\mathbb{E}_{q_{\boldsymbol{\theta}}(\mathbf{W})}\big[\log p(\mathcal{D} \mid \mathbf{W}) - \log q_{\boldsymbol{\theta}}(\mathbf{W}) + \log r_{\boldsymbol{\alpha}}(\mathbf{W}) \\
&\qquad - \log r_{\boldsymbol{\alpha}}(\mathbf{W}) + \log p(\mathbf{W})\big] \\
=&\,\mathbb{E}_{q_{\boldsymbol{\theta}}(\mathbf{W})}\big[\log p(\mathcal{D} \mid \mathbf{W}) - \log r_{\boldsymbol{\alpha}}(\mathbf{W}) + \log p(\mathbf{W}) \\
&\qquad - \tilde{f}_{\mathrm{D}}^*(\mathbf{W})\big]
\end{aligned} \tag{16}$$

where now $\tilde{f}_{\mathrm{D}}^*(\mathbf{w})$ is trained to approximate the log-density ratio $\log q_{\boldsymbol{\theta}}(\mathbf{W}) - \log r_{\boldsymbol{\alpha}}(\mathbf{W})$.

Because of this minimax optimization, AVB suffers in our experiments from scalability issues,[6] and we therefore only experimented with it for low-dimensional problems, where it turns out to be in general the best method, both in terms of performance of individual runs and in ease of finding suitable hyperparameters.

An interesting question regarding AVB is the choice of the discriminator's architecture, which processes complete weight samples $\mathbf{w}$ to determine whether they originate from the prior or from the

---

[6]Note, also Pawlowski et al. [73] reports difficulties when applying AVB to BNNs.

approximate posterior distribution. For low-dimensional problems, we use MLP architectures since the dimensionality of $\mathbf{w}$ is only in the order of 100 weights. For high-dimensional problems, where using a plain MLP is infeasible, we experimented with a chunking approach. Specifically, we used one MLP to reduce the dimensionality of individual weight chunks, which are then concatenated and fed as input to a second MLP that generates the actual output of the discriminator. However, we did not succeed with this approach, and leave the scaling of AVB to large main networks as an open problem for future work.

### C.4.2 Spectral Stein Gradient Estimator

Stein's identity and the related Stein discrepancy have also been investigated to develop training methods for *implicit* distributions [30]. Li and Turner [51] and Shi et al. [79] observed that the training of *implicit* distributions (e.g., via VI) often only requires access to $\nabla_{\mathbf{w}} \log q_{\boldsymbol{\theta}}(\mathbf{w})$, a quantity that appears in Stein's identity. Notably, both approaches do no require an auxiliary network nor an inner-loop optimization.

Li and Turner [51] uses an MC estimate of Stein's identity in combination with ridge regression to obtain $\nabla_{\mathbf{w}} \log q_{\boldsymbol{\theta}}(\mathbf{w})$ estimates. Unfortunately, their method can only be used to estimate this quantity for sample points retrieved from $q_{\boldsymbol{\theta}}(\mathbf{w})$. When requiring an estimate for sample points obtained from a different distribution, as it is the case for the cross-entropy term appearing in a *prior-focused* setting, this method is not applicable.

Therefore, we consider an alternative, the spectral Stein gradient estimator (SSGE, Shi et al. [79]), referred to as **PosteriorReplay-SSGE** (*PosteriorReplay-Imp* in the main text) within our *posterior meta-replay* framework. For completeness, we sketch below the inner workings of SSGE.

Recall that our ultimate goal is to find the parameters $\boldsymbol{\theta}$ that maximize the ELBO, and we therefore need to evaluate the following expression:

$$
\begin{aligned}
\nabla_{\boldsymbol{\theta}} ELBO = \nabla_{\boldsymbol{\theta}} \mathbb{E}_{q_{\boldsymbol{\theta}}(\mathbf{W})} \Big[ \log p(\mathcal{D} \mid \mathbf{W}) \Big] \\
- \nabla_{\boldsymbol{\theta}} \mathbb{E}_{q_{\boldsymbol{\theta}}(\mathbf{W})} \Big[ \log q_{\boldsymbol{\theta}}(\mathbf{W}) \Big] + \nabla_{\boldsymbol{\theta}} \mathbb{E}_{q_{\boldsymbol{\theta}}(\mathbf{W})} \Big[ \log p(\mathbf{W}) \Big]
\end{aligned}
\tag{17}
$$

Thus, the gradient acts on three distinct terms, the first term simply corresponds to the NLL when learning on the data $\mathcal{D}$, and we refer to the other two terms as the *entropy* and the *cross-entropy*. In this specific case, the NLL and cross-entropy term can be approximated using an MC estimate, and their gradient computation through individual samples becomes feasible by using the *reparametrization trick*. However, as explained previously, the entropy term is difficult to compute since we do not have access to the density of $q_{\boldsymbol{\theta}}(\mathbf{W})$. This term can be rewritten as follows:

$$
\begin{aligned}
\nabla_{\boldsymbol{\theta}} \mathbb{E}_{q_{\boldsymbol{\theta}}(\mathbf{W})} \Big[ \log q_{\boldsymbol{\theta}}(\mathbf{W}) \Big] &= \nabla_{\boldsymbol{\theta}} \int_{\mathbf{w}} q_{\boldsymbol{\theta}}(\mathbf{w}) \log q_{\boldsymbol{\theta}}(\mathbf{w}) d\mathbf{w} \\
&= \nabla_{\boldsymbol{\theta}} \int_{\mathbf{z}} p(\mathbf{z}) \log q\big(f_{\mathrm{WG}}(\mathbf{z}, \boldsymbol{\theta}), \boldsymbol{\theta}\big) d\mathbf{z} \\
&= \int_{\mathbf{z}} p(\mathbf{z}) \nabla_{\boldsymbol{\theta}} \log q\big(f_{\mathrm{WG}}(\mathbf{z}, \boldsymbol{\theta}), \boldsymbol{\theta}\big) d\mathbf{z} \\
&= \mathbb{E}_{p(\mathbf{Z})} \Big[ \nabla_{\boldsymbol{\theta}} \log q\big(f_{\mathrm{WG}}(\mathbf{Z}, \boldsymbol{\theta}), \boldsymbol{\theta}\big) \Big] \\
&= \mathbb{E}_{p(\mathbf{Z})} \Big[ \nabla_{\boldsymbol{\theta}} \log q\big(f_{\mathrm{WG}}(\mathbf{Z}, \boldsymbol{\theta}), \hat{\boldsymbol{\theta}}\big)\big|_{\hat{\boldsymbol{\theta}} = \boldsymbol{\theta}} \Big] \\
&= \mathbb{E}_{p(\mathbf{Z})} \Big[ \nabla_{\mathbf{w}} \log q(\mathbf{W}, \boldsymbol{\theta})\big|_{W = f_{\mathrm{WG}}(\mathbf{Z}, \boldsymbol{\theta})} \nabla_{\boldsymbol{\theta}} f_{\mathrm{WG}}(\mathbf{Z}, \boldsymbol{\theta}) \Big]
\end{aligned}
\tag{18}
$$

where to get the second line we have used the following reparametrization $\mathbf{w} = f_{\mathrm{WG}}(\mathbf{z}, \boldsymbol{\theta})$ and $\mathbf{z} \sim p(\mathbf{Z})$, and rewritten $q_{\boldsymbol{\theta}}(\mathbf{W})$ as $q(\mathbf{W}, \boldsymbol{\theta})$ for clarity. Furthermore, to obtain the fifth line we computed the total derivative, and directly used the fact that $\mathbb{E}_{p(\mathbf{Z})} \big[ \nabla_{\boldsymbol{\theta}} \log q(\mathbf{W}, \boldsymbol{\theta})\big|_{\mathbf{w} = f_{\mathrm{WG}}(\mathbf{z}, \boldsymbol{\theta})} \big]$

cancels out. Here is the derivation for completeness:

$$\mathbb{E}_{p(\mathbf{Z})}\big[\nabla_{\boldsymbol{\theta}}\log q(\mathbf{W},\boldsymbol{\theta})\big|_{\mathbf{W}=f_{\mathrm{WG}}(\mathbf{Z},\boldsymbol{\theta})}\big]$$

$$= \mathbb{E}_{q_{\boldsymbol{\theta}}(\mathbf{W})}\big[\nabla_{\boldsymbol{\theta}}\log q(\mathbf{W},\boldsymbol{\theta})\big]$$

$$= \int_{\mathbf{w}} q_{\boldsymbol{\theta}}(\mathbf{w})\nabla_{\boldsymbol{\theta}}\log q(\mathbf{w},\boldsymbol{\theta})d\mathbf{w}$$

$$= \int_{\mathbf{w}} q_{\boldsymbol{\theta}}(\mathbf{w})\frac{1}{q_{\boldsymbol{\theta}}(\mathbf{w})}\nabla_{\boldsymbol{\theta}}q(\mathbf{w},\boldsymbol{\theta})d\mathbf{w}$$

$$= \nabla_{\boldsymbol{\theta}}\int_{\mathbf{w}} q(\mathbf{w},\boldsymbol{\theta})d\mathbf{w} = \nabla_{\boldsymbol{\theta}}1 = 0 \tag{19}$$

Coming back to Eq. 18, we see that two gradients need to be computed within the expectation. The second term $\nabla_{\boldsymbol{\theta}}f_{\mathrm{WG}}(\mathbf{z},\boldsymbol{\theta})$ is simply the gradient of the output of the weight-generator hypernetwork with respect to its own parameters, and can therefore be easily obtained using automatic differentiation. The expression $\nabla_{\mathbf{w}}\log q_{\boldsymbol{\theta}}(\mathbf{w})$ is not accessible for implicit distributions, and needs to be estimated.

SSGE provides a way to estimate $\nabla_{\mathbf{w}}\log q_{\boldsymbol{\theta}}(\mathbf{w})$ by considering a spectral decomposition:

$$\nabla_{w_i}\log q_{\boldsymbol{\theta}}(\mathbf{w}) = \sum_{j=1}^{\infty}\gamma_{ij}\varphi_j(\mathbf{w}) \tag{20}$$

where $\varphi$ are the eigenfunctions of a covariance kernel $k(\mathbf{w}^i,\mathbf{w}^j)$ with respect to $q_{\boldsymbol{\theta}}(\mathbf{w})$ and $\gamma_{ij}$ are the coefficients of the spectral series. Both need to be estimated. The Nyström method is used to approximate the eigenfunctions. For this, SSGE considers the following eigenvalue problem $K\mathbf{u}\approx\lambda\mathbf{u}$, where $K\in\mathbb{R}^{S\times S}$ is the Gram matrix: $K_{ij}=k(\mathbf{w}^i,\mathbf{w}^j)$, $S$ is the number of samples used for the gradient estimation. We only consider the radial basis function (RBF) kernel in this work. The $J$ eigenvectors $\mathbf{u}^1\ldots\mathbf{u}^J$ with the $J$ largest eigenvalues $\lambda_1\geq\ldots\geq\lambda_J$ are computed and later will be selected to approximate the spectral series. We discuss below how to set $J$. The $j$-th eigenfunction can now be estimated based on the Nyström method with:

$$\varphi_j(\mathbf{w}) = \frac{\sqrt{S}}{\lambda_j}\sum_{s=1}^{S} u_s^j k(\mathbf{w},\mathbf{w}^s) \tag{21}$$

where $u_s^j$ denotes the $s$-th element of the $j$-th eigenvector. Finally, Stein's identity allows finding the following expression for the coefficients $\gamma$:

$$\gamma_{ij} = -\frac{1}{\sqrt{S}\lambda_j}\sum_{n=1}^{S}\sum_{s=1}^{S}\nabla_{w_i^n}k(\mathbf{w}^n,\mathbf{w}^s)u_s^j \tag{22}$$

The estimated eigenfunctions and the corresponding coefficients can then be inserted back into a finite form of Eq. 20 to estimate $\nabla_{\mathbf{w}}\log q_{\boldsymbol{\theta}}(\mathbf{w})$, and multiplied by the gradient of $f_{\mathrm{WG}}$ with respect to $\boldsymbol{\theta}$ that we obtain via automatic differentation. Finally, taking the expectation over the base distribution $p(\mathbf{Z})$ allows obtaining an estimate for the gradient of the entropy term in the ELBO (Eq. 18). Note, that if $\boldsymbol{\theta}$ is the output of a TC network, then the gradient estimate $\nabla_{\boldsymbol{\theta}}\mathbb{E}_{q_{\boldsymbol{\theta}}(\mathbf{W})}\big[q_{\boldsymbol{\theta}}(\mathbf{W})\big]$ has to be further backpropagated to the parameters of the TC network, where it is accumulated with the gradient of the remaining loss terms, all of which have been automatically computed via automatic differentiation.

The use of SSGE introduces three extra hyperparameters that need to be tuned: the width of the RBF kernel, the number of samples $S$ used for the eigenvalue decomposition, and the number $J$ of eigenfunctions. For the kernel width, we explored setting it to some arbitrary small value or to the median of pairwise distances between all samples, as described in the original paper [79]. In our results, this choice did not considerably impact performance. For the number of eigenfunctions, we experimented with directly setting $J$ to some fixed value or, as suggested in the original paper, with setting it based on a certain ratio $\tau$ of cumulative eigenvalues (i.e., select the minimum number of eigenfunctions $J$ such that $\frac{\sum_{j=1}^{J}\lambda_j}{\sum_{j=1}^{S}\lambda_j}>\tau$).

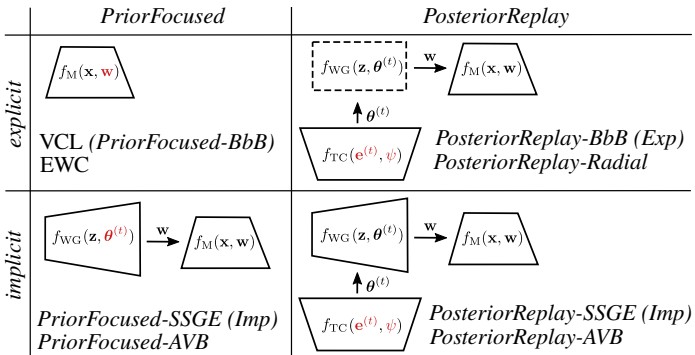

Figure S2: Summary of the different algorithms used to train BNNs, and how they fit into the described framework. We distinguish between approaches that learn a single shared posterior across tasks (*PriorFocused*) and approaches that learn task-specific posteriors (*PosteriorReplay*). We consider both *explicit* and *implicit* posterior approximations. Red indicates the parameters that are learned in each scenario.

### C.4.3 Support of implicit approximate posterior distributions

Special attention needs to be payed to the support of the approximate posterior distribution when it is parametrized by an auxiliary network. Indeed, when using a WG architecture for which the output size is larger than the input size, or that contains some bottleneck layer, the support of $q_{\boldsymbol{\theta}}(\mathbf{W})$ will be limited to a set of measure zero (cf. Lemma 1 in Arjovsky and Bottou [3]). This causes the *prior-matching term* of the ELBO to be ill-defined as the KL definition requires $q_{\boldsymbol{\theta}}(\mathbf{W})$ to be absolutely continuous with respect to $p(\mathbf{W})$. To overcome this limitation, we injected small noise perturbations to the outputs of the WG, e.g.,

$$\mathbf{w} \sim q_{\boldsymbol{\theta}}(\mathbf{W}) \Leftrightarrow \mathbf{w} = f_{\mathrm{WG}}(\mathbf{z}, \boldsymbol{\theta}) + \mathbf{u} \tag{23}$$

with $\mathbf{z} \sim p(\mathbf{Z})$ and $\mathbf{u} \sim p(\mathbf{U})$, where $p(\mathbf{U})$ is an additional noise distribution. Note, when using the *reparametrization trick* to rewrite expected values with respect to $q_{\boldsymbol{\theta}}(\mathbf{W})$, one now has to integrate over the joint of $\mathbf{Z}$ and $\mathbf{U}$.

Due to practical considerations, we use the following simplification in our implementation:

$$\mathbf{w} \sim q_{\boldsymbol{\theta}}(\mathbf{W}) \Leftrightarrow \mathbf{w} = f_{\mathrm{WG}}(\mathbf{z}_{:n_z}, \boldsymbol{\theta}) + \sigma_{\mathrm{noise}}\mathbf{z} \tag{24}$$

where $\mathbf{z}_{:n_z}$ refers to the first $n_z$ elements of the vector $\mathbf{z}$, and both $\sigma_{\mathrm{noise}}$ and $n_z$ are hyperparameters.

### C.5 Prior-Focused Continual Learning

*Prior-focused* CL [17] is an alternative Bayesian approach to CL that is commonly used in the literature. As opposed to *posterior meta-replay*, a single set of shared parameters is recursively updated, and finding suitable solutions therefore relies on the existence of trade-off solutions across tasks. We first describe VCL and EWC, two existing algorithms that use a Gaussian approximation for the shared posterior, and then discuss how our framework can be used to render *prior-focused* methods more flexible through the use of *implicit* distributions. Note that *prior-focused* approaches can be implemented within our framework (denoted *PriorFocused*) by directly learning a single set of parameters $\boldsymbol{\theta}$ that define the shared posterior distribution, which is recursively instantiated as prior. Since $\boldsymbol{\theta}$ is not task-specific, no TC network is required. An overview of how *PriorFocused* and *PosteriorReplay* approaches fit in our proposed framework is given in Fig. S2.

### C.5.1 Variational Continual Learning

Variational continual learning (VCL, Nguyen et al. [65], Swaroop et al. [83]) was introduced as a way to continually learn a single mean-field Gaussian posterior approximation across tasks by doing a recursive Bayesian update via VI. More specifically, VCL aims to learn a single approximate posterior $q_{\boldsymbol{\theta}^{(1:T)}}(\mathbf{W})$ by recursively considering the posterior of the previous task as the prior for the new task:

$$q_{\boldsymbol{\theta}^{(1:T)}}(\mathbf{W}) \propto p(\mathcal{D}^{(T)} \mid \mathbf{W}) q_{\boldsymbol{\theta}^{(1:T-1)}}(\mathbf{W}) \tag{25}$$

where $q_{\boldsymbol{\theta}^{(1:T-1)}}(\mathbf{W})$ is an approximation to the previous posterior $p(\mathbf{W} \mid \mathcal{D}^{(1:T-1)})$. Importantly, the CL requirements are not violated since only data from the current task $\mathcal{D}^{(T)}$ is required for the update. To learn the approximate posteriors, VCL uses BbB, and learns a mean-field Gaussian approximation parametrized by a mean and variance per weight, which are optimized by using the *reparametrization trick* (Eq. 5). An interesting contribution of this method is the mathematically sound posterior update based on coresets (i.e., a small set of samples stored from each task, [75, 4]) within their Bayesian framework. Importantly, the role of coresets in their approach is to help mitigate catastrophic forgetting. In contrast, in this work we mainly use coresets to facilitate task inference in a task-agnostic setting (see SM C.7 for details).

In our framework, VCL is realized via a main network and a simple WG function with parameters $\boldsymbol{\theta}$ as depicted in Fig. 2 for BbB. Conceptual differences between using BbB in a *prior-focused* setting (VCL), and using BbB using a *posterior meta-replay* approach (*PosteriorReplay-BbB*) are illustrated in Figure S1. We study this method in a multihead setting (*VCL-multihead*), where each head is task-specific and leads to a task-specific approximate posterior that induces epistemic uncertainty. Therefore, the predictive uncertainty of each head has a task-specific influence that we exploit for task inference. An interesting extension of this multihead setting (i.e., task-specific output parameters) could be CLAW [1] which builds on top of VCL and utilizes a set of task-specific neuronal parameters. Thus task-specific parameters are distributed throughout the whole network (not just the outputs), which might be beneficial for uncertainty-based task-inference. We also consider VCL with a growing head (**VCL-growing**), as described in SM C.5.2. Note, as we often apply likelihood-tempering in practice (SM D) for scalability reasons, our VCL version is often related to GVCL [54].

### C.5.2 Elastic Weight Consolidation

Kirkpatrick et al. [38] propose elastic weight consolidation (EWC), a CL algorithm that limits the plasticity of weights that are considered important for solving previous tasks, and therefore mitigates forgetting. In contrast to VCL, the algorithm is based on a Laplace approximation of the posterior [56] where approximations are restricted to diagonal covariance matrices.[7] For completeness, we detail here the derivation of this algorithm, specifically of its more mathematically sound variant *Online EWC* [32, 78], which we simply refer to as EWC in the tables for brevity. Afterwards, we explain how this algorithm is commonly used in the literature in a task-agnostic setting, and propose an alternative solution based on a combination of shared and task-specific parameters that leads to improved performance.

**Online EWC.** The core idea of EWC (and of *prior-focused* methods in general) simply consists in performing a recursive Bayesian update of a single posterior distribution as new tasks arrive:

$$p(\mathbf{W} \mid \mathcal{D}^{(1:T)}) = \frac{p(\mathcal{D}^{(T)} \mid \mathbf{W})p(\mathbf{W} \mid \mathcal{D}^{(1:T-1)})}{p(\mathcal{D}^{(T)} \mid \mathcal{D}^{(1:T-1)})} \tag{26}$$

where we have used the fact that $\mathcal{D}^{(1:T-1)}$ and $\mathcal{D}^{(T)}$ are conditionally independent given $\mathbf{W}$. For simplicity, we start by considering $T = 2$:

$$p(\mathbf{W} \mid \mathcal{D}^{(1:2)}) = \frac{p(\mathcal{D}^{(2)} \mid \mathbf{W})p(\mathbf{W} \mid \mathcal{D}^{(1)})}{p(\mathcal{D}^{(2)} \mid \mathcal{D}^{(1)})} \tag{27}$$

This is almost identical to the original formulation (cf. Eq 2. in Kirkpatrick et al. [38]), except that the denominator contains $p(\mathcal{D}^{(2)} \mid \mathcal{D}^{(1)})$ and not simply $p(\mathcal{D}^{(2)})$, since the datasets are only conditionally independent, as noted by Huszár [32]. Optimizing the parameters $\mathbf{W}$ corresponds to finding their most probable value given the data:

$$\arg\min_{\mathbf{W}} \left\{ -\log p(\mathbf{W} \mid \mathcal{D}^{(1:2)}) \right\} \Leftrightarrow \tag{28}$$

$$\arg\min_{\mathbf{W}} \left\{ -\log p(\mathcal{D}^{(2)} \mid \mathbf{W}) - \log p(\mathbf{W} \mid \mathcal{D}^{(1)}) \right\}$$

where we have dropped constant terms. Notice that $-\log p(\mathcal{D}^{(2)} \mid \mathbf{W})$ is simply the NLL and can easily be computed. The second term, $\log p(\mathbf{W} \mid \mathcal{D}^{(1)})$ is generally intractable, and for this reason

---

[7]An interesting extension with non-diagonal covariance matrices is described in Ritter et al. [76].

we consider a second order Taylor approximation around the minimum that was found after learning the first task $\mathbf{w}_{\mathrm{MAP}}^{(1)}$, corresponding to the maximum a posteriori (MAP) estimate. First order terms vanish around the minimum and we obtain the following expression:

$$\log p(\mathbf{W} \mid \mathcal{D}^{(1)}) \approx const+ \tag{29}$$

$$\frac{1}{2}(\mathbf{W} - \mathbf{w}_{\mathrm{MAP}}^{(1)})^T \mathcal{H}_{\log p(\mathbf{w}|\mathcal{D}^{(1)})}\Big|_{\mathbf{w}=\mathbf{w}_{\mathrm{MAP}}^{(1)}} (\mathbf{W} - \mathbf{w}_{\mathrm{MAP}}^{(1)})$$

where $\mathcal{H}_{\log p(\mathbf{W}|\mathcal{D}^{(1)})}$ denotes the Hessian of $\log p(\mathbf{W} \mid \mathcal{D}^{(1)})$, which can be rewritten as:

$$\mathcal{H}_{\log p(\mathbf{W}|\mathcal{D}^{(1)})} = \mathcal{H}_{\log p(\mathcal{D}^{(1)}|\mathbf{W})} + \mathcal{H}_{\log p(\mathbf{W})}$$

$$= \nabla_{\mathbf{w}} \nabla_{\mathbf{w}}^T \Big[ \sum_{n=1}^{N} \log p(\mathbf{y}^{(1,n)} \mid \mathbf{W}, \mathbf{x}^{(1,n)})$$

$$- \frac{1}{2\sigma_{prior}^2} \|\mathbf{W}\|_2^2 \Big]$$

$$= \sum_{n=1}^{N} \mathcal{H}_{\log p(\mathbf{y}^{(1,n)}|\mathbf{W},\mathbf{x}^{(1,n)})} - \frac{1}{\sigma_{prior}^2} I \tag{30}$$

where $\mathbf{x}^{(1,n)}$ denotes the $n$-th sample of the first task and where we have used the fact that all $N$ samples are independent given $\mathbf{w}$. Furthermore, we have assumed a Gaussian prior such that $p(\mathbf{W}) = \mathcal{N}(0, I\sigma_{prior}^2)$, where $I$ is the identity matrix. Recall the relationship between the Hessian and the Fisher information matrix $F$ (e.g., cf. Eq. 3/4 in Martens [59]):

$$F = -\mathbb{E}_{p(\mathbf{y}|\mathbf{W},\mathbf{x})}\Big[\mathcal{H}_{\log(\mathbf{y}|\mathbf{W},\mathbf{x})}\Big] \tag{31}$$

Note that so far we have considered $\mathbf{x}$ fixed, but the model should perform well with respect to the input distribution $p(\mathbf{x})$:

$$\mathbb{E}_{p(\mathbf{x})}\Big[ - F \Big] = \mathbb{E}_{p(\mathbf{x})p(\mathbf{y}|\mathbf{W},\mathbf{x})}\Big[\mathcal{H}_{\log p(\mathbf{y}|\mathbf{W},\mathbf{x})}\Big] \tag{32}$$

$$\approx \frac{1}{N} \sum_{n=1}^{N} \mathcal{H}_{\log p(\tilde{\mathbf{y}}^{(n)}|\mathbf{W},\mathbf{x}^{(n)})}$$

where the approximation in the last line comes from an MC estimate using $N$ samples drawn from the joint $p(\mathbf{y} \mid \mathbf{W}, \mathbf{x})p(\mathbf{x})$. Assuming $\mathbf{w}$ has been trained to optimality, i.e., $p(\mathbf{y} \mid \mathbf{W}, \mathbf{x}) \approx p(\mathbf{y} \mid \mathbf{x})$ for $\mathbf{x} \sim p(\mathbf{x})$, the samples used in Eq. 30 and Eq. 32 are essentially from the same joint distribution and we can write:

$$\frac{1}{N} \sum_{n=1}^{N} \mathcal{H}_{\log p(\mathbf{y}^{(n)}|\mathbf{W},\mathbf{x}^{(n)})} \approx \frac{1}{N} \sum_{n=1}^{N} \mathcal{H}_{\log p(\tilde{\mathbf{y}}^{(n)}|\mathbf{W},\mathbf{x}^{(n)})} \tag{33}$$

to obtain the following expression:

$$\mathcal{H}_{\log p(\mathbf{W}|\mathcal{D}^{(1)})} \approx -N F_{emp} - \frac{1}{\sigma_{prior}^2} I \tag{34}$$

where we have introduced the empirical Fisher, which is obtained using the sample points from the actual dataset $\mathcal{D}$, but given the optimality assumption above here it simply corresponds to $F_{emp} = \mathbb{E}_{p(\mathbf{x})}\big[F\big]$.[8] Plugging this result into Eq. 29, while using a diagonal approximation of the empirical Fisher matrix, and extending it to an arbitrary number of tasks by iteratively computing the posterior, we obtain the following expression (cf. Eq. 11 in Huszár [32]):

$$\log p(\mathbf{W} \mid \mathcal{D}^{(1:T)}) \approx const + \log p(\mathcal{D}^{(T)} \mid \mathbf{W}) - \tag{35}$$

$$\frac{1}{2} \sum_i \left( \sum_{t<T} N^{(t)} F_{emp_i^{(t)}} + \frac{1}{\sigma_{prior}^2} \right)(w_i - w_{\mathrm{MAP},i}^{(T-1)})^2$$

---

[8]Note, that for mathematical convenience we include the expectation over $p(\mathbf{x})$ in the definition of the empirical Fisher.

where $N^{(t)}$ denotes the size of $\mathcal{D}^{(t)}$. Thus, per weight $w_i$ there is a scalar importance value $\sum_{t<T} N^{(t)} F_{emp_i^{(t)}} + \frac{1}{\sigma_{prior}^2}$ which can be computed online such that there is no need to maintain the empirical Fisher matrices of individual tasks. This is in contrast to the original EWC formulation Kirkpatrick et al. [38], where each Fisher matrix had to be maintained in memory.

Crucially, in order to relate the sum over Hessians in Eq. 30 to the Fisher information matrix, we have to assume that $p(\mathbf{y} \mid \mathbf{W}, \mathbf{x})$ and $p(\mathbf{y} \mid \mathbf{x})$ are identical for $\mathbf{x} \sim p(\mathbf{x})$. In this case, the empirical Fisher and the (expected) Fisher are also identical and it mathematically does not matter which of the two is used for the algorithm.

**Multihead EWC.** When EWC is used in a task-agnostic inference setting, it is often trained with an output softmax that grows as new tasks are trained (e.g., van de Ven et al. [88]), which we refer to as *EWC-growing*. In this setting, where EWC performs poorly, the Bayesian assumption that the model class contains the ground-truth model is violated, since the model-class changes whenever the softmax changes in size. A simple way to overcome this issue when the number of tasks is known in advance, is to use a shared softmax that spans the outputs of all tasks from the beginning of training, which we refer to as **EWC-shared**. However, as we will show, this approach still performs poorly. We hypothesize this is due to the fact that the role played by output connections for their corresponding task is in conflict with that played for all other tasks (i.e., because the shared softmax pushes the weights of future heads to be highly negative). As a solution, we propose the use of a multihead when applying EWC to a task-agnostic inference setting (referred to as *EWC-multihead*). This results in a hybrid *prior-focused* approach, where the learned parameters $\mathbf{w}$ consist of a set of shared weights $\phi$ and a set of task-specific output heads with weights $\{\xi^{(t)}\}_{t=1}^T$. Now, Eq. 27 becomes:

$$p(\phi, \boldsymbol{\xi}^{(1:2)} \mid \mathcal{D}^{(1:2)}) \tag{36}$$
$$= \frac{p(\mathcal{D}^{(2)} \mid \phi, \boldsymbol{\xi}^{(2)}) p(\phi, \boldsymbol{\xi}^{(1)} \mid \mathcal{D}^{(1)}) p(\boldsymbol{\xi}^{(2)})}{p(\mathcal{D}^{(2)} \mid \mathcal{D}^{(1)})}$$

Repeating the procedure in the original derivation, i.e. considering a Taylor approximation around optimal parameters and recursively computing the posterior, we obtain the following expression:

$$\log p(\phi, \boldsymbol{\xi}^{(1:T)} \mid \mathcal{D}^{(1:T)}) \approx const \tag{37}$$
$$+ \log p(\mathcal{D}^{(T)} \mid \phi, \boldsymbol{\xi}^{(T)}) + \log p(\boldsymbol{\xi}^{(T)})$$
$$- \frac{1}{2} \sum_i \left( \sum_{t<T} N^{(t)} F_{emp_i^{(t)}} + \frac{1}{\sigma_{prior}^2} \right) (w_i - w_{\text{MAP},i}^{(T-1)})^2$$

where we have assumed that all parameters $\mathbf{w} = [\phi, \boldsymbol{\xi}^{(1)}, \dots, \boldsymbol{\xi}^{(T)}]$ share the same prior. Importantly, for any $t$, $F_{emp^{(t)}}$ is a square diagonal matrix of dimension $\dim(\mathbf{w})$, where all entries related to weights $\boldsymbol{\xi}^{(s)}$ for $s \neq t$ are zero.

Recall, that the EWC importance values are reminiscent of the entries of a precision matrix of a multivariate Gaussian with diagonal covariance matrix. Together with the final MAP estimate $\mathbf{w}_{\text{MAP}}^{(T)}$ of $p(\mathbf{W} \mid \mathcal{D}^{(1:T)})$ we can explicitly construct the following approximate posterior in (Online) multihead EWC:

$$\log p(\phi, \boldsymbol{\xi}^{(1:T)} \mid \mathcal{D}^{(1:T)}) = \tag{38}$$
$$\mathcal{N}\left( \mathbf{w}_{\text{MAP}}^{(T)}, \left[ \frac{1}{\sigma_{prior}^2} I + \sum_{t=1}^T N^{(t)} F_{emp^{(t)}} \right]^{-1} \right)$$

This posterior induces predictive uncertainty at each output head, which can be used for task inference in task-agnostic inference settings.

### C.5.3 Prior-focused CL with implicit distributions

Both VCL and EWC use a Gaussian approximation for the shared posterior. This means that, not only a trade-off solution across tasks needs to be found, but also that the expressivity of this posterior with respect to unknown future tasks is limited. To overcome this limitation, we explore *prior-focused*

methods with a more flexible variational family, a family of *implicit* distributions parametrized by a neural network. Although this approach does not overcome the need to find trade-off solutions across tasks, it can make better use of the existing overlaps by capturing, for example, multi-modality. Within our framework, *PriorFocused* methods with an *implicit* posterior distribution can be realised by directly learning the parameters $\boldsymbol{\theta}$ of a WG hypernetwork. This can be achieved using algorithms for learning with *implicit* distributions, such as AVB or SSGE, and we therefore refer to these methods as **PriorFocused-AVB** and **PriorFocused-SSGE**.

If not noted otherwise, we consider a hybrid approach with task-specific weights introduced by a multihead main network, such that task inference through task-specific predictive uncertainty is possible.

### C.6 Task inference

When confronted with an unseen input $\tilde{\mathbf{x}}$, algorithms that maintain task-specific solutions, such as our *posterior meta-replay* framework, first have to explicitly assign the input to a certain task. In this work, we exclusively consider inferring task identity via predictive uncertainty, and explore four different ways of quantifying uncertainty as outlined below.

The *Ent* criterion, only considered in classification tasks, can be computed directly from the posterior predictive distribution $p(\mathbf{y} \mid \mathcal{D}^{(t)}; \tilde{\mathbf{x}}) = \int_{\mathbf{W}} p(\mathbf{y} \mid \mathbf{W}; \tilde{\mathbf{x}}) p(\mathbf{W} \mid \mathcal{D}^{(t)}) \, d\mathbf{W}$. Due to intractability of this integral, we resort to an MC estimate using $K = 100$ models drawn from the approximate posterior parameter distribution $q_{\boldsymbol{\theta}^{(t)}}(\mathbf{W})$ of each task:

$$p(\mathbf{y} \mid \mathcal{D}^{(t)}; \tilde{\mathbf{x}}) \approx \frac{1}{K} \sum_{k=1}^{K} p(\mathbf{y} \mid \mathbf{w}^{(t,k)}; \tilde{\mathbf{x}}) \tag{39}$$

with $\mathbf{w}^{(t,k)} \sim q_{\boldsymbol{\theta}^{(t)}}(\mathbf{W})$. Note that for deterministic approaches such as *PosteriorReplay-Dirac*, only $K = 1$ is applicable. In the *Ent* criterion, the task leading to the lowest entropy is selected:

$$t_{Ent} = \underset{t \in 1..T}{\arg\min} \, \mathcal{H}\big\{ p(\mathbf{y} \mid \mathcal{D}^{(t)}; \tilde{\mathbf{x}}) \big\} \tag{40}$$

where $\mathcal{H}\{\}$ denotes the entropy functional.

Similarly, in the **Conf** criterion [26], also only considered for classification problems, the task leading to the highest confidence is selected:

$$t_{Conf} = \underset{t \in 1..T}{\arg\max} \, \underset{\mathbf{y} \in \mathcal{Y}}{\max} \, p(\mathbf{y} \mid \mathcal{D}^{(t)}; \tilde{\mathbf{x}}) \tag{41}$$

where $\mathcal{Y}$ denotes the set of class labels, which indeed could be task-specific.

These two criteria intuitively correspond to choosing the model with the most peaky predictive distribution, i.e. the highest certainty in the predicted class. However, uncertainty in the predictive distribution can also arise in-distribution due to noisy data, i.e., aleatoric uncertainty. In order to quantify epistemic uncertainty only, we additionally study model agreement as uncertainty measure. Note that, in regions where sufficient data has been observed, models drawn from the posterior parameter distribution should converge towards the data-generating distribution $p(\mathbf{Y} \mid \mathbf{X})$ and should therefore agree among each other. Conversely, regions where those models disagree have not observed enough data and can be considered OOD, assuming a rich enough prior in function space [11]. This intuition should be captured in *Agree*, where the task leading to the strongest agreement between models is selected. For $C$-way classification, we compute this quantity as the average standard deviation of predicted likelihood values for $K$ models $\mathbf{w}^{(t,k)} \sim q_{\boldsymbol{\theta}^{(t)}}(\mathbf{W})$ [81]:

$$t_{Agree} = \underset{t \in 1..T}{\arg\min} \, \frac{1}{C} \sum_{c=1}^{C} \mathrm{SD}\big\{ p(Y = c \mid \mathbf{w}^{(t,k)}; \tilde{\mathbf{x}}) \quad \forall k \big\} \tag{42}$$

where $\mathrm{SD}(\cdot)$ refers to the standard deviation of the given set of values.

Alternatively, Depeweg et al. [13] propose a neat decomposition of the entropy into two terms:

$$\mathcal{H}\big\{ p(\mathbf{y} \mid \mathcal{D}^{(t)}; \tilde{\mathbf{x}}) \big\} = \mathbb{E}_{p(\mathbf{W} \mid \mathcal{D}^{(t)})} \mathcal{H}\big\{ p(\mathbf{y} \mid \mathbf{W}; \tilde{\mathbf{x}}) \big\} + \mathcal{I}(\mathbf{y}, \mathbf{W}) \tag{43}$$

where $\mathcal{I}(\mathbf{y}, \mathbf{W})$ denotes the mutual information between $\mathbf{y}$ and $\mathbf{W}$, given by:

$$\mathcal{I}(\mathbf{y}, \mathbf{W}) = \mathrm{KL}\big(p(\mathbf{y}, \mathbf{W} \mid \mathcal{D}; \tilde{\mathbf{x}}) || p(\mathbf{W}|\mathcal{D}^{(t)})p(\mathbf{y} \mid \mathcal{D}^{(t)}; \tilde{\mathbf{x}})\big) \tag{44}$$

The two terms on the RHS of Eq. 43 are interpreted as aleatoric and epistemic uncertainty. Following the terminology from Malinin et al. [58], we refer to the mutual information term as *knowledge uncertainty* (noted **KU**). Note, that the remaining terms in Eq. 43 can be estimated via Monte-Carlo using the approximate posterior $q_{\boldsymbol{\theta}^{(t)}}(\mathbf{W})$ instead of $p(\mathbf{W}|\mathcal{D}^{(t)})$. Although we are unsure about whether these terms can be generally interpreted as epistemic and aleatoric uncertainty, we studied how this knowledge uncertainty estimate performs in practice in SplitMNIST-10 experiments.

Note, that choosing to use predictive uncertainty for task inference is a heuristic choice and, to the best of our knowledge, there is no guarantee that predictive uncertainty can lead to principled OOD detection in general [11]. However, except when exploring hybrid approaches, practitioners have to trade-off pros and cons when selecting a CL method that can be deployed in a task-agnostic setting. As discussed in the main text, if task identity can be inferred from inputs alone, in theory it seems reasonable to perform task inference directly on approximations of the unknown input distributions $p^{(t)}(\mathbf{x})$, e.g., via tractable density access [46]. While in practice this approach still seems to be out of reach for high-dimensional problems [64], generative models for data replay have been successfully applied to CL [80, 86, 91]. In this case, when a new task is trained, generated data from past tasks is replayed to train in a pseudo-i.i.d. setting. Because of the pseudo-parallel training on all tasks, task inference does not have to be handled explicitly anymore. Note, however, that a conceptual disadvantage exists when using replay approaches, i.e., the problem of training with non-i.i.d. data is not directly addressed and rather side-stepped completely for the target network by shifting CL to the generative model. Finally, we want to stress again that the task identity can also be externally provided to the TC, e.g., by an auxiliary system that processes context data (different than the main network's input) [24, 91].

## C.7 Posterior-Replay CL with Coreset Fine-Tuning

As opposed to the deterministic setting, already acquired knowledge can be updated in a principled way when using a Bayesian perspective. When continually learning a sequence of tasks, this allows approximate posteriors to be revisited as new data comes in. This can be used in our *posterior meta-replay* framework to mitigate forgetting by storing task-specific coresets and using them to refresh all task-specific posteriors in a small fine-tuning stage at the end of training.

Specifically, we consider a dataset split per task $\mathcal{D}^{(t)} \setminus \mathcal{C}^{(t)} \cup \mathcal{C}^{(t)}$, where $\mathcal{D}^{(t)} \setminus \mathcal{C}^{(t)}$ is used for the initial CL training phase, and $\mathcal{C}^{(t)}$ is maintained in memory for the fine-tuning stage. More explicitly, when continually learning our meta-model we approximate the following posteriors:

$$q_{\tilde{\boldsymbol{\theta}}^{(t)}}(\mathbf{W}) \approx p(\mathbf{W} \mid \mathcal{D}^{(t)} \setminus \mathcal{C}^{(t)}) \tag{45}$$

after which $\mathcal{D}^{(t)} \setminus \mathcal{C}^{(t)}$ can be discarded, and only a small-sized coreset $\mathcal{C}^{(t)}$ needs to be maintained in memory until all tasks are learned.

The fine-tuning stage at the end of training is then performed in a multitask fashion on the stored coresets, which are all simultaneously available. To see how this final update can be performed we write our posterior distribution as follows:

$$p(\mathbf{W} \mid \mathcal{D}^{(t)}) = \frac{p(\mathbf{W})p(\mathcal{D}^{(t)} \mid \mathbf{W})}{p(\mathcal{D}^{(t)})} = \frac{p(\mathbf{W})p(\mathcal{D}^{(t)} \setminus \mathcal{C}^{(t)} \mid \mathbf{W})p(\mathcal{C}^{(t)} \mid \mathbf{W})}{p(\mathcal{D}^{(t)} \setminus \mathcal{C}^{(t)}, \mathcal{C}^{(t)})} \tag{46}$$

$$\propto \frac{p(\mathbf{W})p(\mathcal{D}^{(t)} \setminus \mathcal{C}^{(t)} \mid \mathbf{W})p(\mathcal{C}^{(t)} \mid \mathbf{W})}{p(\mathcal{D}^{(t)} \setminus \mathcal{C}^{(t)})} = p(\mathbf{W} \mid \mathcal{D}^{(t)} \setminus \mathcal{C}^{(t)})p(\mathcal{C}^{(t)} \mid \mathbf{W})$$

where we have used the fact that $\mathcal{D}^{(t)} \setminus \mathcal{C}^{(t)}$ and $\mathcal{C}^{(t)}$ are conditionally independent given $\mathbf{W}$. Eq. 47 shows that the desired posteriors $p(\mathbf{W} \mid \mathcal{D}^{(t)})$ can be obtained via a Bayesian update of the continually learned posteriors $p(\mathbf{W} \mid \mathcal{D}^{(t)} \setminus \mathcal{C}^{(t)})$ by using the task-specific coreset $\mathcal{C}^{(t)}$. Recall that in VI, we approximate this posterior with a distribution $q_{\boldsymbol{\theta}^{(t)}}(\mathbf{W})$ by minimizing $\mathrm{KL}(q_{\boldsymbol{\theta}^{(t)}}(\mathbf{W}) \parallel p(\mathbf{W} \mid \mathcal{D}^{(t)}))$. Based on our dataset split, the expression to be minimized can be rewritten as $\mathrm{KL}(q_{\boldsymbol{\theta}^{(t)}}(\mathbf{W}) \parallel \zeta p(\mathbf{W} \mid \mathcal{D}^{(t)} \setminus \mathcal{C}^{(t)})p(\mathcal{C}^{(t)} \mid \mathbf{W}))$, where $\zeta$ is a normalization constant that accounts

for the fact that we dropped $\mathcal{C}^{(t)}$ from the denominator. Replacing $p(\mathbf{W} \mid \mathcal{D}^{(t)} \setminus \mathcal{C}^{(t)})$ by the approximations $q_{\tilde{\boldsymbol{\theta}}^{(t)}}(\mathbf{W})$ learned continually, we obtain the following VI objective:

$$\arg\min_{\boldsymbol{\theta}^{(t)}} \mathrm{KL}\big( q_{\boldsymbol{\theta}^{(t)}}(\mathbf{W}) \,\|\, \zeta q_{\tilde{\boldsymbol{\theta}}^{(t)}}(\mathbf{W}) p(\mathcal{C}^{(t)} \mid \mathbf{W}) \big) \tag{47}$$

$$= \arg\min_{\boldsymbol{\theta}^{(t)}} \left[ \mathrm{KL}\big( q_{\boldsymbol{\theta}^{(t)}}(\mathbf{W}) \,\|\, q_{\tilde{\boldsymbol{\theta}}^{(t)}}(\mathbf{W}) \big) - \mathbb{E}_{q_{\boldsymbol{\theta}^{(t)}}} \big[ \log p(\mathcal{C}^{(t)} \mid \mathbf{W}) \big] \right]$$

Note that the second term corresponds to integrating the evidence from the coreset into the new posterior, and therefore simply corresponds to minimizing the NLL on the coreset $\mathcal{C}^{(t)}$, while the first term ensures that the posterior does not change much, preventing forgetting. Notably, the KL term is now between two distributions from the same variational family, reminiscent of *prior-focused* learning as discussed in SM C.5. Therefore, it can be analytically evaluated for *PosteriorReplay-BbB* while other methods require estimation, e.g., using AVB or SSGE.

Crucially, having coresets available at the end of training can also be used to facilitate task inference based on predictive uncertainty (refer to SM C.6 for details). To do this, we perform the final update on a set of modified coresets, which encourage the prediction of a task-specific model on OOD data (i.e., on a coreset from another task) to be highly uncertain. More specifically, we use the modified coresets:

$$\tilde{\mathcal{C}}^{(t)} = \mathcal{C}^{(t)} \cup \bigcup_{s \in \{1,\dots,T\} \setminus \{t\}} \hat{\mathcal{C}}^{(s)} \tag{48}$$

where $\hat{\mathcal{C}}^{(s)}$ is constructed from the same inputs contained in $\mathcal{C}^{(s)}$ but the labels are replaced by high uncertainty labels. We consider per-task coresets of size 100 and enforce high uncertainty in OOD inputs by either setting high-entropy softmax labels (i.e., uniform distribution; eg., [45]), or by using different random labels per mini-batch.

Intriguingly, when training with data from other tasks, this data becomes in-distribution and cannot be considered OOD anymore. Therefore, we expect our task inference criterion *Agree*, which is designed to capture only epistemic uncertainty, to become less reliable. In other terms, what was previously OOD data, that could have been detected via epistemic uncertainty, now became in-distribution data with high aleatoric uncertainty due to the way we design training targets. This intuition is reflected in our empirical observations (cf. SM D).

### C.8  Experience Replay

Experience replay [52] refers to the idea of using a replay buffer (or coresets) to store current experiences that can later be replayed in order to mitigate forgetting. Implementations of this method might slightly differ in how coresets are assembled and how replayed data is incorporated into the loss [e.g., 2, 7, 9]. Our implementation of experience replay (**Exp-Replay**) is designed to be comparable to our proposed coreset fine-tuning (SM C.7). Therefore, we use task-specific coresets of size 100 (coresets are build using a random subset of a task's training set). A coreset only contains input samples, and a checkpointed model from the previous task is used to generate distillation targets [28]. Thus, a distillation loss is added to the overall loss. This distillation loss uses a mini-batch that is created by randomly sampling from all available coresets in a stratified manner.

## D  Supplementary Experiments and Results

In this section, we extend the results presented in the main text, and provide a more detailed discussion. Furthermore, we report additional baselines and supplementary experiments such as PermutedMNIST (cf. SM D.4).

**Baselines.** In addition to the methods and baselines that have already been introduced, we consider the following variations. To investigate the role played by the shared meta-model, we consider independently trained models per task (*SeparatePosteriors*); a baseline that by design is not affected by catastrophic interference, and therefore leads to identical *TGiven-During* and *TGiven-Final* scores. In this setting, task-specific solutions cannot benefit from knowledge transfer, but they are also less limited because the capacity of the meta-model is not shared with previous tasks.

We distinguish two cases for this baseline. In the first case, there is no TC network and $\boldsymbol{\theta}$ is trained directly (denoted *SeparatePosteriors-<method>*, e.g., **SeparatePosteriors-BbB**). However, one has

to keep in mind that the underlying architecture in combination with the chosen weight prior defines a prior in function space that will affect predictive uncertainty (cf. Sec. 3 and [93]), and it is therefore unclear how comparable task inference scores based on predictive uncertainty are in this case, e.g., between *SeparatePosteriors-BbB* and *PosteriorReplay-BbB*. For this reason, we include a second *SeparatePosteriors* baseline where the *TC* network remains but has no dedicated functional purpose (denoted *SeparatePosteriors-TC-<method>*, e.g., **SeparatePosteriors-TC-BbB**). Note that for this baseline, one TC network with a single task embedding is learned per task.

As a potential upper-bound we also consider **Fine-Tuning**, which simply refers to the continuous deterministic training of a main network without undertaking any measures against catastrophic forgetting. Thus, the achieved *TGiven-During* score can benefit from transfer while not being restricted to finding any trade-off that accommodates past tasks. In this case, the hyperparameter configuration is selected based on the best *TGiven-During* score. We only searched hyperparameters for this baseline in the PermutedMNIST and SplitCIFAR experiments, because the *TGiven-During* scores of other experiments are often maxed out. Importantly, whenever applicable and unless noted otherwise, all hyperparameter configurations have been selected based on the *TInfer-Final (Ent)* criterion. Hence, reported task-given (*TGiven*) scores do not necessarily reflect the ability of a method to combat forgetting. However, in most cases, forgetting does not seem to be a major challenge in the experiments we consider, and the reported *TGiven-During* and *TGiven-Final* scores are often very close. Please refer to SM E for details on the experimental setup and hyperparameter searches.

**Tempering.** All results involving Bayesian methods use a standard Gaussian prior $p(\mathbf{W}) = \mathcal{N}(\mathbf{0}, I)$. For high-dimensional problems and for methods trained with variational inference, we explore tempering the posterior to increase training stability (cf. SM E.3). Specifically, we downscale the *prior-matching term*, which effectively increases the emphasis on matching the data well (cf. SM E in Wenzel et al. [92]). Note, no such posterior tempering is used when studying the low-dimensional problems in SM D.1 and SM D.2.

### D.1   1D Polynomial Regression

Table S2: Mean-squared-error (MSE) values for the 1D polynomial regression experiments. *TInfer* refers to the MSE when making predictions using the task embedding leading to lowest uncertainty for the given input. The column *ACC-inference* reports the accuracy for how often the correct task-embedding was chosen for (in-distribution) test inputs. All methods use a singlehead. Note, that singlehead *PriorFocused* methods do not require task inference, and thus the distinction between *TGiven* and *TInfer* does not apply (Mean $\pm$ SEM, *ACC-inference* in %, $n = 10$). PR refers to *PosteriorReplay* and PF to *PriorFocused* methods.

|  | TGiven-During | TGiven-Final | TInfer-Final | ACC-inference |
|---|---|---|---|---|
| PR-Dirac | $0.01007 \pm 0.00170$ | $0.01037 \pm 0.00162$ | N/A | N/A |
| PR-BbB | $0.01036 \pm 0.00112$ | $0.01037 \pm 0.00121$ | $0.01175 \pm 0.00103$ | $98.07 \pm 0.53$ |
| PR-Radial | $0.01238 \pm 0.00184$ | $0.01108 \pm 0.00090$ | $0.03996 \pm 0.02377$ | $96.19 \pm 1.77$ |
| PR-AVB | $0.00504 \pm 0.00061$ | $0.00712 \pm 0.00181$ | $0.00712 \pm 0.00181$ | $100.00 \pm 0.00$ |
| PR-SSGE | $0.00236 \pm 0.00016$ | $0.00251 \pm 0.00020$ | $0.00476 \pm 0.00208$ | $99.80 \pm 0.13$ |
| EWC | $0.08004 \pm 0.02444$ | $2.55570 \pm 1.07003$ | N/A | N/A |
| VCL | $0.09787 \pm 0.00982$ | $0.23889 \pm 0.06711$ | N/A | N/A |
| PF-AVB | $0.24125 \pm 0.00315$ | $0.46187 \pm 0.07851$ | N/A | N/A |
| PF-SSGE | $0.01546 \pm 0.00082$ | $2.09433 \pm 0.78611$ | N/A | N/A |

In this section we expand on the discussion from Sec. 4.1 on continually learning a set of low-dimensional regression tasks. Details about the dataset can be found in SM E.1. Quantitative results for a ReLU MLP-10,10 main network can be found in Table S2.

*PosteriorReplay* methods are all able to fit the polynomials well and do not seem to be affected by catastrophic interference. Notably, due to the choice of Gaussian likelihood with fixed variance (cf. Eq. 9), the likelihood function is not able to represent $x$-dependent (aleatoric) uncertainty. Therefore, all $x$-dependent uncertainty corresponds to parameter uncertainty, which is not captured by *PosteriorReplay-Dirac*. Interestingly, the reported *PosteriorReplay-AVB* run is always able to pick the right task embedding in a random-seed robust way, such that *TGiven-Final* and *TInfer-Final* scores

are identical. All reported *PriorFocused* methods use a singlehead main network and therefore do not require task inference. Instead, they have to learn a single posterior continually that captures well all data across tasks. Here, we observe that these methods greatly suffer from the stability-plasticity dilemma [71]: having enough plasticity to accommodate new tasks causes catastrophic interference with existing knowledge, while excessive protection of the existing shared posterior does not give the flexibility required to fit new data.

Supplementary qualitative plots showing the final approximate posteriors found by the *PosteriorReplay* methods are depicted in Fig. S3. While these plots convey the intuition behind uncertainty-based task inference, it should be noted that the ground-truth posterior shape is unknown, and that hyperparameters have a strong influence on these plots.

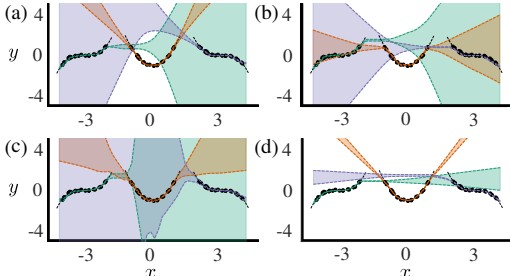

Figure S3: 1D polynomial regression task, where three polynomials need to be learned consecutively. Different colors represent the task-specific posterior approximations within the final model when training with different *PosteriorReplay* algorithms: **(a)** *PosteriorReplay-BbB*, **(b)***PosteriorReplay-Radial*, **(c)** *PosteriorReplay-AVB*, **(d)** *PosteriorReplay-SSGE*. The illustrations sketch the idea behind task inference via predictive uncertainty, i.e., for an input point $x$ the posterior with the lowest predictive uncertainty can be chosen to make predictions. Note, if the input does not lie in the in-distribution space of any task, predictive uncertainty will be high for all posteriors.

Finally, to highlight a potential advantage of *implicit* methods, we qualitatively investigate the found posterior approximations by *PosteriorReplay-AVB* in Fig. S4. We provide joint density plots (using kernel density estimation) for random pairs of weights as well as Pearson correlation matrices for a random subset of weights. Note that these plots are not cherry-picked and are representative of the found posterior approximations. Weights in the posteriors of all tasks seem to be highly correlated, which is also to be expected for the unknown ground-truth posterior, since individual weight changes will affect the behavior of the neural network function and should therefore be coordinated to maintain stable in-distribution predictions. This is in contrast to mean-field approximations such as *PosteriorReplay-BbB* and *PosteriorReplay-Radial*, which by design are not able to capture weight correlations. In addition, the joint distribution plots for *PosteriorReplay-AVB* often exhibit multi-modality. Presumably, this could be beneficial for OOD detection and thus task inference, since different modes in the posterior landscape may represent very different functions that exert vastly different behavior on OOD data and, indeed, when studying low-dimensional problems, we generally found it easier to find viable hyperparameter configurations with *implicit* methods than with *explicit* ones. However, such argument is purely speculative and should be considered with care given that single high-dimensional posterior modes might be flat in many directions and thus also be able to capture a diverse set of functions [29, 68].

### D.2   2D Mode Classification

Next, we reconsider the 2D mode classification introduced in Sec. 4.2. All results are obtained with a ReLU MLP-10,10 main network. Details about this synthetic dataset can be found in SM E.2, and quantitative results are reported in Table S3.

As reflected in the *TGiven-Final* scores, forgetting is not a concern in this benchmark. Instead, we use this experiment to gain insights on how task inference based on predictive uncertainty works in classification tasks. Recall, that the softmax likelihood function can represent arbitrary **x**-dependent discrete distributions, which (in contrast to Sec. 4.1) allows the model to exhibit **x**-dependent aleatoric uncertainty. Deterministic approaches such as *PosteriorReplay-Dirac* only have access to aleatoric uncertainty, and therefore rely solely on it for task inference. However this has certain

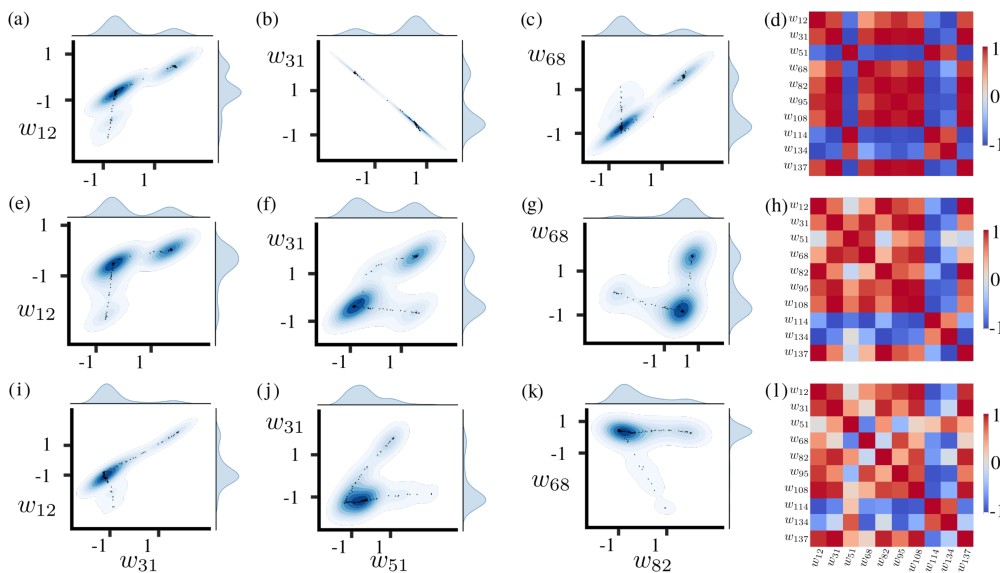

Figure S4: Example illustrations of weight distributions captured by the final approximate posteriors when learning the 1D polynomial regression task with *PosteriorReplay-AVB*. **(a)-(c)** Joint density plots of task 1 for three random pairs of weights. **(d)** Weight correlations in task 1 for a subset of randomly chosen weights. Same for the approximate posterior of task 2 **(e)-(h)** and task 3 **(i)-(l)**. Note, that unlike unimodal mean-field approximations (cf. *PosteriorReplay-BbB* and *PosteriorReplay-Radial*), *implicit* methods can capture correlations between weights and multi-modality.

Table S3: Accuracies for the 2D mode classification experiments (Mean $\pm$ SEM in %, $n = 10$). PR refers to *PosteriorReplay*, PF to *PriorFocused* methods and SP to *SeparatePosteriors*.

|  | TGiven-During | TGiven-Final | TInfer-Final (Ent) | TInfer-Final (Conf) | TInfer-Final (Agree) |
|---|---|---|---|---|---|
| PR-Dirac | $100.0 \pm 0.00$ | $99.78 \pm 0.21$ | $44.90 \pm 5.74$ | $45.43 \pm 5.84$ | N/A |
| PR-BbB | $100.0 \pm 0.00$ | $100.0 \pm 0.00$ | $81.07 \pm 6.78$ | $81.07 \pm 6.78$ | $90.02 \pm 3.57$ |
| PR-Radial | $95.08 \pm 2.38$ | $95.08 \pm 2.38$ | $54.50 \pm 5.01$ | $54.50 \pm 5.01$ | $76.33 \pm 4.18$ |
| PR-AVB | $100.0 \pm 0.00$ | $100.0 \pm 0.00$ | $98.57 \pm 1.33$ | $98.57 \pm 1.33$ | $99.93 \pm 0.06$ |
| PR-SSGE | $100.0 \pm 0.00$ | $100.0 \pm 0.00$ | $100.0 \pm 0.00$ | $100.0 \pm 0.00$ | $100.0 \pm 0.00$ |
| SP-Dirac | N/A | $100.0 \pm 0.00$ | $70.92 \pm 4.96$ | $74.17 \pm 3.78$ | N/A |
| SP-BbB | N/A | $100.0 \pm 0.00$ | $85.13 \pm 2.58$ | $85.13 \pm 2.58$ | $87.92 \pm 2.29$ |
| SP-AVB | N/A | $100.0 \pm 0.00$ | $95.00 \pm 2.09$ | $95.00 \pm 2.09$ | $98.62 \pm 1.23$ |
| VCL-multihead | $100.0 \pm 0.00$ | $100.0 \pm 0.00$ | $45.17 \pm 3.80$ | $45.17 \pm 3.80$ | $47.13 \pm 4.26$ |
| PF-AVB-multihead | $100.0 \pm 0.00$ | $98.17 \pm 1.57$ | $64.53 \pm 4.99$ | $64.53 \pm 4.99$ | $65.40 \pm 5.05$ |
| PF-SSGE-multihead | $100.0 \pm 0.00$ | $95.92 \pm 1.71$ | $50.00 \pm 3.33$ | $49.40 \pm 3.06$ | $50.02 \pm 3.33$ |

limitations since aleatoric uncertainty is only calibrated in-distribution. For instance, the typical cross-entropy loss criterion used for classification tasks can be linked to the minimization of the quantity: $\mathbb{E}_{p(\mathbf{X})}\big[\mathrm{KL}\big(p(\mathbf{Y} \mid \mathbf{X})||p(\mathbf{Y} \mid \mathbf{W};\mathbf{X})\big)\big]$, where $p(\mathbf{X})p(\mathbf{Y} \mid \mathbf{X})$ denotes the unknown underlying data-generating process and $p(\mathbf{Y} \mid \mathbf{W};\mathbf{X})$ is the model likelihood. Thus, the behaviour of aleatoric uncertainty as reflected in $p(\mathbf{Y} \mid \mathbf{W};\mathbf{X})$ for OOD data (e.g., outside the support of $p(\mathbf{X})$) is not calibrated and can indeed be harmful for OOD detection (e.g., [25]). This intuition is validated by our *PosteriorReplay-Dirac* results, which exhibit rather arbitrary behavior on OOD data (Fig. S6). This might also explain the notable difference in the results reported for *PosteriorReplay-Dirac* and *SeparatePosteriors-Dirac* in Table S3, while probabilistic *PosteriorReplay* methods behave similarly. We provide uncertainty maps for all studied probabilistic *PosteriorReplay* methods in Fig. S5.

A desirable behavior for good OOD detection would entail having low uncertainty only where the training data of the corresponding task resides, while having high uncertainty elsewhere. As opposed to the deterministic case, Bayesian approaches reflect this intuition (cf. Fig. S6), despite not being

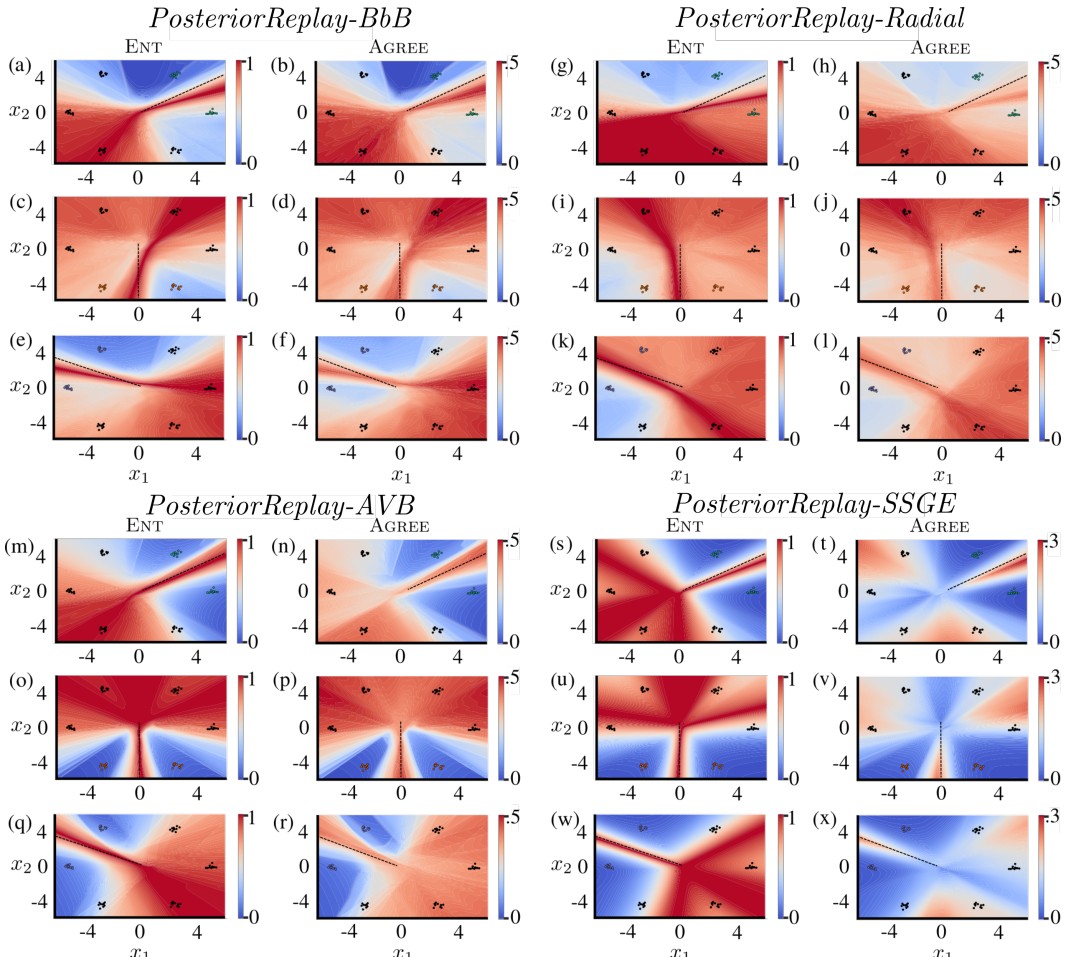

Figure S5: 2D mode classification uncertainty maps for probabilistic *PosteriorReplay* approaches. **(a)** Entropy of the predictive distribution (*Ent*) of task 1 across the input space covering in-distribution domains of all tasks, when training with *PosteriorReplay-BbB*. Dots represent training points and colors indicate task-affiliation. The dashed line represents the decision boundary for task 1. **(b)** Same as (a) but uncertainty here reflects model agreement (*Agree*). The same uncertainty maps are produced using the final approximate posteriors for task 2 **(c)**-**(d)** and 3 **(e)**-**(f)**. To show qualitative results for several probabilistic *PosteriorReplay* approaches, the uncertainty maps (a)-(f) are repeated for *PosteriorReplay-Radial* **(g)**-**(l)**, *PosteriorReplay-AVB* **(m)**-**(r)** and *PosteriorReplay-SSGE* **(s)**-**(x)**.

perfect OOD detectors. Note that, since the uncertainty map of the true Bayesian posterior is unknown, it remains unclear whether this imperfection originates from the approximate nature of the used posteriors, or whether it is innate to the real posterior. Importantly, the displayed uncertainty maps represent aggregated results over many models drawn from the approximate posteriors. Therefore, an interesting question arises, i.e., whether the depicted uncertainty maps result from models having high aleatoric uncertainty on OOD data, or from individual models behaving very differently on OOD data. To answer this question, we plot the softmax entropy maps for individual models drawn from the posterior of *PosteriorReplay-AVB* in Fig. S7.

Uncertainty maps from individual models look very different, suggesting that epistemic uncertainty is crucial for OOD detection in this experiment. This is supported by the fact that the *Ent* uncertainty map looks very similar to that obtained with *Agree*, even though *Ent* is supposed to capture both aleatoric and epistemic uncertainty. However, one needs to keep in mind that these are qualitative results which strongly depend the selected hyperparameter configuration, as can be seen for the chosen *PosteriorReplay-SSGE* results.

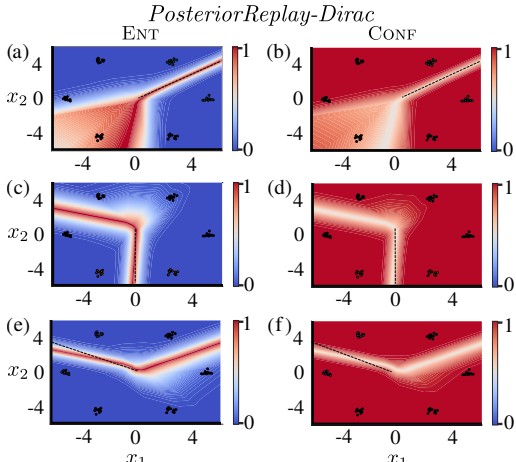

Figure S6: 2D mode classification uncertainty maps for *PosteriorReplay-Dirac*. **(a)** Entropy of the predictive distribution (*Ent*) of task 1 across the input space covering in-distribution domains of all tasks. Dots represent training points, colors task-affiliation and the dashed line the decision boundary for task 1. **(b)** Same as (a) but uncertainty here reflects the confidence of the predictive distribution (*Conf*). Same uncertainty maps with the final approximate posteriors for task 2 **(c)**-**(d)** and 3 **(e)**-**(f)**.

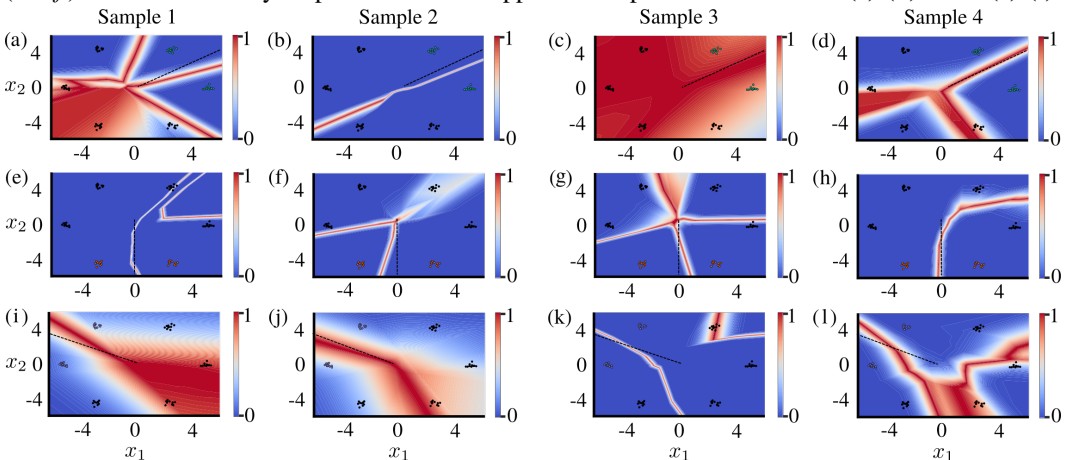

Figure S7: 2D mode classification uncertainty maps of single sample points drawn from the final approximate posteriors trained with *PosteriorReplay-AVB*. **(a)** Entropy of the softmax output for a sample point $\mathbf{w} \sim q_{\boldsymbol{\theta}^{(1)}}(\mathbf{W})$ of task 1 across the input space covering in-distribution domains of all tasks. **(b)**-**(d)** same as (a) for different sample points $\mathbf{w} \sim q_{\boldsymbol{\theta}^{(1)}}(\mathbf{W})$. Posterior draws for task 2 using $q_{\boldsymbol{\theta}^{(2)}}(\mathbf{W})$ are shown in **(e)**-**(h)** and for task 3 using $q_{\boldsymbol{\theta}^{(3)}}(\mathbf{W})$ are shown in **(i)**-**(l)**, respectively. Intriguingly, individual models tend to be overconfident (similar to Fig. S6), and different models exhibit very different uncertainty behavior. Thus, the overall uncertainty reflected in the predictive distribution (cf. Fig. S5) is likely induced through parameter uncertainty.

In this experiment, *PriorFocused* methods perform worse in uncertainty-based task inference than *PosteriorReplay* methods. A reason could be the impaired ability of *PriorFocused* methods to capture task-specific epistemic uncertainty. This may be due to the fact that data from all tasks is used to train a shared body, and thus the posterior models drawn from it should lead to similar hidden activations (i.e., low epistemic uncertainty) for data from all tasks. Thus, epistemic uncertainty, which we saw above is crucial for task inference, will mainly be introduced by the task-specific outputs heads which process those hidden activations. In contrast, the posterior of all weights in *PosteriorReplay* methods has only ever seen data from one task and can therefore exhibit high epistemic uncertainty when computing hidden activations for data from different tasks.

We also experiment with *EWC-multihead*, for which we obtain chance level predictions. To understand this effect, let's recall that EWC relies on the post-hoc construction of the posterior, for which the

computation of the empirical Fisher information matrix is required in order to approximate the loss curvature. In our results, we observe that some of the computed Fisher values are zero. Looking at Eq. 38 one can see that whenever Fisher values are very low for a certain weight, the variance of the posterior for that weight will be dominated by the variance of the prior, which was set to one in our experiments. We hypothesize that such high variance for certain weights in the constructed posterior is the reason for the low performance displayed by *EWC-multihead* in such a small network. Note, that the empirical Fisher values are computed as a sum of squared log-likelihood gradients computed across the whole training set. These gradients are zero for some weights, which might be due to overfitting; a conceivable problem given the low dataset size. We observed the same issue when testing *EWC-multihead* for the 1D regression experiment. Intriguingly, this observation also implies that the empirical Fisher approximation failed to appropriately approximate the Hessian and thus capture loss curvature.

## D.3 SplitMNIST

Here we provide additional results obtained for the SplitMNIST experiment. In particular, we compare the different methods for three different main network architectures: Table S4 contains results obtained with a ReLU MLP with two hidden layers of 100 neurons (MLP-100,100), Table S5 for an MLP-400,400 and Table S6 for a Lenet [44] with a kernel size of 5, 20 resp. 50 feature maps in the two convolutional layers and 500 units in the penultimate fully-connected layer.

Table S4: Accuracies of SplitMNIST experiments when using an MLP-100,100 (Mean $\pm$ SEM in %, $n = 10$). Column *TInfer-Final* represents *TInfer-Final (Ent)* accuracies if applicable. Otherwise this column is used to report results of methods that learn a shared softmax across all tasks. PR refers to *PosteriorReplay* and PF to *PriorFocused* methods.

|                    | TGiven-During   | TGiven-Final    | TInfer-Final    | TInfer-Final (Conf) | TInfer-Final (Agree) |
| ------------------ | --------------- | --------------- | --------------- | ------------------- | -------------------- |
| PR-Dirac           | $99.72 \pm 0.01$ | $99.72 \pm 0.01$ | $63.41 \pm 1.54$ | $48.41 \pm 1.02$ | N/A              |
| PR-BbB             | $99.75 \pm 0.01$ | $99.75 \pm 0.01$ | $70.07 \pm 0.56$ | $65.34 \pm 0.66$ | $70.11 \pm 0.54$ |
| PR-Radial          | $99.55 \pm 0.06$ | $99.43 \pm 0.10$ | $63.00 \pm 1.56$ | $64.03 \pm 1.19$ | $63.06 \pm 1.55$ |
| PR-SSGE            | $99.66 \pm 0.02$ | $99.65 \pm 0.02$ | $66.15 \pm 0.92$ | $47.89 \pm 0.67$ | $66.20 \pm 0.92$ |
| EWC-growing        | N/A             | N/A             | $28.15 \pm 0.51$ | N/A             | N/A              |
| EWC-shared         | N/A             | N/A             | $29.67 \pm 0.86$ | N/A             | N/A              |
| EWC-multihead      | $99.01 \pm 0.03$ | $97.79 \pm 0.30$ | $46.61 \pm 1.26$ | $46.63 \pm 1.27$ | $19.97 \pm 0.01$ |
| VCL-multihead      | $97.83 \pm 0.03$ | $96.05 \pm 0.21$ | $51.45 \pm 1.11$ | $51.45 \pm 1.11$ | $51.02 \pm 1.22$ |
| PF-SSGE-multihead  | $99.74 \pm 0.01$ | $96.48 \pm 0.54$ | $51.26 \pm 1.72$ | $49.64 \pm 1.72$ | $51.59 \pm 1.64$ |
| PR-BbB-BW          | $99.75 \pm 0.01$ | $99.75 \pm 0.01$ | $99.75 \pm 0.01$ | $99.75 \pm 0.01$ | $99.75 \pm 0.01$ |
| PR-BbB-CS          | $99.41 \pm 0.04$ | $98.70 \pm 0.05$ | $90.42 \pm 0.19$ | $90.42 \pm 0.19$ | $59.09 \pm 0.64$ |
| Exp-Replay         | N/A             | N/A             | $86.84 \pm 0.51$ | N/A             | N/A              |

Some general trends can be observed independent of the main network architecture used. Catastrophic forgetting is not a major issue in this experiment, as illustrated by similar *TGiven-During* and *TGiven-Final* scores across methods. Only *prior-focused* approaches seem slightly affected by forgetting and sometimes exhibit a slight drop in *TGiven-Final* accuracies, which however stay above 96% for all methods and architectures. Much wider variations can be observed in the task-agnostic setting where task identity needs to be inferred. Despite the fact that we could improve upon previously reported results for the *PosteriorReplay-Dirac* baseline in MLP-400,400 (termed HNET+ENT in von Oswald et al. [91]), this deterministic solution generally performs worse at inferring task identity than probabilistic approaches, notably *PosteriorReplay-BbB*. Interestingly, *PosteriorReplay-BbB* consistently outperforms *PosteriorReplay-Radial* in this experiment by a large extent, which is in disagreement with the performance gains reported by Farquhar et al. [18] on other datasets. A potential cause for the differences in performance can be that, as opposed to Farquhar et al. [18], we tempered the posteriors [92]. This was done because successful training for both *PosteriorReplay-Radial* and *PosteriorReplay-BbB* could only be accomplished when notably reducing the prior influence in the loss computation. When using *implicit* posterior distributions via *PosteriorReplay-SSGE*, performance gains can be observed over *PosteriorReplay-BbB* for MLP-400,400 and Lenet main networks. Using SSGE to learn a single *implicit* posterior distribution (*PriorFocused-SSGE-multihead*) leads however

Table S5: Accuracies of SplitMNIST experiments when using an MLP-400,400 (Mean $\pm$ SEM in %, $n = 10$). Column *TInfer-Final* represents final accuracies when task identity is inferred with the *Ent* criterion, if applicable. Otherwise this column is used to report results of methods that learn a shared softmax across all tasks. Results denoted with a * are taken from van de Ven and Tolias [87] and those denoted with a ** are taken from von Oswald et al. [91]. SI stands for *synaptic intelligence* [96], and DGR for *deep generative replay* Shin et al. [80]. HNET+TIR and HNET+R are CL methods based on hypernetwork-protected replay proposed in von Oswald et al. [91]. PR refers to *PosteriorReplay*, PF to *PriorFocused* methods and SP to *SeparatePosteriors*.

| | TGiven-During | TGiven-Final | TInfer-Final | TInfer-Final (Conf) | TInfer-Final (Agree) |
|---|---|---|---|---|---|
| PR-Dirac | 99.65 ± 0.02 | 99.65 ± 0.01 | 70.88 ± 0.61 | 56.56 ± 0.64 | N/A |
| PR-BbB | 99.73 ± 0.01 | 99.72 ± 0.02 | 71.73 ± 0.87 | 67.42 ± 0.68 | 71.73 ± 0.85 |
| PR-Radial | 99.64 ± 0.01 | 99.64 ± 0.01 | 66.01 ± 0.92 | 61.56 ± 0.38 | 66.22 ± 0.93 |
| PR-SSGE | 99.78 ± 0.01 | 99.77 ± 0.01 | 71.91 ± 0.79 | 55.15 ± 0.67 | 71.43 ± 0.77 |
| EWC-growing | N/A | N/A | 27.32 ± 0.60 | N/A | N/A |
| EWC-shared | N/A | N/A | 30.21 ± 0.52 | N/A | N/A |
| EWC-multihead | 99.70 ± 0.01 | 96.40 ± 0.62 | 47.67 ± 1.52 | 47.67 ± 1.52 | 47.52 ± 1.48 |
| VCL-multihead | 96.66 ± 0.19 | 96.45 ± 0.13 | 58.84 ± 0.64 | 58.84 ± 0.64 | 56.54 ± 0.94 |
| PF-SSGE-multihead | 99.79 ± 0.01 | 99.02 ± 0.16 | 62.70 ± 1.32 | 61.62 ± 1.31 | 62.76 ± 1.31 |
| PR-Dirac-SR | 99.65 ± 0.01 | 99.64 ± 0.01 | 71.34 ± 0.49 | 58.05 ± 0.35 | N/A |
| PR-BbB-SR | 99.73 ± 0.02 | 99.73 ± 0.02 | 72.38 ± 0.77 | 66.50 ± 0.76 | 72.39 ± 0.77 |
| PR-BbB-BW | 99.73 ± 0.01 | 99.72 ± 0.02 | 99.72 ± 0.02 | 99.72 ± 0.02 | 99.72 ± 0.02 |
| PR-BbB-CS | 99.34 ± 0.05 | 98.50 ± 0.09 | 90.83 ± 0.24 | 90.83 ± 0.24 | 59.74 ± 0.59 |
| SP-Dirac | N/A | 99.77 ± 0.01 | 70.39 ± 0.27 | 63.69 ± 0.10 | N/A |
| SP-BbB | N/A | 99.81 ± 0.00 | 68.40 ± 0.23 | 63.37 ± 0.85 | 68.37 ± 0.24 |
| SP-SSGE | N/A | 99.76 ± 0.04 | 71.53 ± 1.34 | 68.50 ± 0.99 | 71.36 ± 1.31 |
| SP-TC-Dirac | N/A | 99.79 ± 0.02 | 71.84 ± 0.49 | 50.69 ± 1.23 | N/A |
| SP-TC-BbB | N/A | 99.77 ± 0.01 | 72.74 ± 0.45 | 72.87 ± 0.57 | 72.79 ± 0.45 |
| Exp-Replay | N/A | N/A | 88.85 ± 0.39 | N/A | N/A |
| EWC-growing* | N/A | N/A | 19.96 ± 0.07 | N/A | N/A |
| SI-growing* | N/A | N/A | 19.99 ± 0.06 | N/A | N/A |
| DGR* | N/A | N/A | 91.79 ± 0.32 | N/A | N/A |
| PR-Dirac** | N/A | N/A | 69.48 ± 0.80 | N/A | N/A |
| HNET+TIR** | N/A | N/A | 89.59 ± 0.59 | N/A | N/A |
| HNET+R** | N/A | N/A | 95.30 ± 0.13 | N/A | N/A |

to a significant drop in performance across architectures, which highlights the potential of a system that learns task-specific posteriors. Yet, compared to *prior-focused* approaches that do not use *implicit* distributions, *PriorFocused-SSGE-multihead* generally performs better, illustrating that more flexible posterior approximations can lead to improved performance when trade-off solutions need to be found.

When applied with a growing (*EWC-growing*) or shared softmax (*EWC-shared*), EWC leads to very poor results in all architectures, even though we manage to considerably improve upon the existing *EWC-growing*\* baseline. The use of a multihead significantly improves performance, but even in this case the performance of EWC remains well below other *PriorFocused* methods such as *VCL-multihead*. A potential cause is the post-hoc construction of the posterior in EWC, as noted in SM D.2.

Note that, if task inference performance is above chance-level for individual inputs, it can be improved as a simple statistical effect by looking at multiple samples belonging to the same task. Indeed, in all three architectures, we observe that performance in a task-agnostic setting can be dramatically improved when task inference is performed on a set of 100 samples rather than individual ones. Task inference becomes perfect in all three architectures, leading to identical *TGiven-Final* and *TInfer-Final* accuracies. Even though we only report results for BbB (**PosteriorReplay-BbB-BW**), all methods equally benefit from such aggregated task inference. An additional way to considerably

Table S6: Accuracies of SplitMNIST experiments when using a Lenet (Mean $\pm$ SEM in %, $n =$ 10). Column *TInfer-Final* represents final accuracies when task identity is inferred with the *Ent* criterion, if applicable. Otherwise this column is used to report results of methods that learn a shared softmax across all tasks. PR refers to *PosteriorReplay*, PF to *PriorFocused* methods and SP to *SeparatePosteriors*.

| | TGiven-During | TGiven-Final | TInfer-Final | TInfer-Final (Conf) | TInfer-Final (Agree) |
|---|---|---|---|---|---|
| PR-Dirac | 99.92 ± 0.01 | 99.87 ± 0.04 | 72.33 ± 2.75 | 57.22 ± 3.25 | N/A |
| PR-BbB | 98.96 ± 0.90 | 99.20 ± 0.67 | 74.09 ± 1.38 | 67.39 ± 1.39 | 74.13 ± 1.33 |
| PR-Radial | 99.85 ± 0.03 | 99.78 ± 0.05 | 68.99 ± 2.06 | 64.63 ± 2.79 | 69.41 ± 2.12 |
| PR-SSGE | 99.89 ± 0.01 | 99.89 ± 0.01 | 77.56 ± 1.01 | 57.29 ± 1.89 | 77.55 ± 1.02 |
| EWC-growing | N/A | N/A | 27.62 ± 0.55 | N/A | N/A |
| EWC-shared | N/A | N/A | 26.01 ± 1.02 | N/A | N/A |
| EWC-multihead | 98.09 ± 0.75 | 97.17 ± 0.54 | 49.78 ± 2.20 | 49.77 ± 2.20 | 49.85 ± 2.11 |
| VCL-multihead | 98.03 ± 0.07 | 97.43 ± 0.12 | 63.05 ± 0.63 | 63.05 ± 0.63 | 62.24 ± 0.73 |
| PF-SSGE-multihead | 99.94 ± 0.00 | 99.37 ± 0.10 | 74.18 ± 1.00 | 38.15 ± 0.90 | 74.12 ± 1.01 |
| PR-BbB-BW | 98.96 ± 0.90 | 99.20 ± 0.67 | 99.20 ± 0.67 | 99.20 ± 0.67 | 99.20 ± 0.67 |
| PR-BbB-CS | 99.76 ± 0.02 | 99.62 ± 0.02 | 95.73 ± 0.05 | 95.73 ± 0.05 | 60.69 ± 1.13 |
| SP-Dirac | N/A | 99.92 ± 0.00 | 85.50 ± 0.28 | 70.98 ± 0.34 | N/A |
| SP-BbB | N/A | 99.93 ± 0.00 | 85.52 ± 0.45 | 71.22 ± 1.41 | 85.47 ± 0.45 |
| SP-TC-Dirac | N/A | 99.91 ± 0.00 | 82.85 ± 0.60 | 55.38 ± 0.25 | N/A |
| SP-TC-BbB | N/A | 99.92 ± 0.00 | 84.16 ± 0.42 | 84.06 ± 0.41 | 84.04 ± 0.38 |

improve task inference performance is the use of coresets in a fine-tuning stage at the end of training (*CS*). This approach leads to results that are comparable to prior work based on generative replay (e.g., DGR*, HNET+TIR**). Although this approach leads to lower performance than the use of batches for inferring the task, the use of coresets does not rely on the assumption that a set of samples belongs to the same task, and when such an assumption is plausible both tricks can be simultaneously used to further boost performance.

The relative behavior of the three criteria for quantifying uncertainty is also stable across architectures. *Ent* and *Agree* generally lead to very similar results, while *Conf* often performs worse, except for *EWC-multihead* and *VCL-multihead*, where all three approaches behave similarly. Interestingly, when using coresets to facilitate task inference, OOD data becomes in-distribution, and hinders the ability of the *Agree* criterion to infer task identity, as shown by our *TInfer-Final (Agree)* results for *PosteriorReplay-BbB-CS*.

Training separate posteriors per task (*SeparatePosteriors*) controls for the influence of the shared system on performance and uncertainty-based task inference. Our results for MLP-400,400 indicate that the CL performance of *PosteriorReplay-Dirac* and *PosteriorReplay-SSGE* is very close to what is achieved when a different model can be allocated per task. To our surprise, however, BbB exhibits lower performance when used to learn independent posteriors (*SeparatePosteriors-BbB*), than when used in a CL setting (*PosteriorReplay-BbB*). Since *PosteriorReplay-SSGE* does not exhibit the same trend, the cause is likely not rooted in Bayesian inference for this model class, but rather in the particular approximation used. Interestingly, the *SeparatePosteriors* performance achieved in the deterministic case and with BbB (*SeparatePosteriors-Dirac*, *SeparatePosteriors-BbB*) improves marginally when the parameters of the approximate posteriors are not learned directly, but generated by a hypernetwork (*SeparatePosteriors-TC-Dirac*, *SeparatePosteriors-TC-BbB*). The results are somewhat different for a Lenet, where learning separate posteriors leads to a much larger performance improvement in *SeparatePosteriors-Dirac* and *SeparatePosteriors-BbB* compared to the *PosteriorReplay* setting. In this case, however, the use of a hypernetwork to generate main network weights does not lead to increased *SeparatePosteriors* performance, highlighting the complicated influence of the architectural setup on uncertainty, and therefore task inference.

For the MLP-400,400 we provide *PosteriorReplay* results for the case where, in each update, the CL regularization is only applied to a subset of tasks chosen randomly, denoted as stochastic

Table S7: Accuracies of SplitMNIST experiments when using an MLP-400,400 (Mean $\pm$ SEM in %, $n = 10$) using the *KU* task-inference criterion.

| | TInfer-Final (KU) |
|---|---|
| PR-BbB | $46.53 \pm 1.04$ |
| PR-SSGE | $53.68 \pm 0.43$ |
| PR-BbB-CS | $42.86 \pm 1.31$ |

regularization (**SR**).[9] Thus, the regularization cost does not increase with the number of tasks. For both *PosteriorReplay-Dirac-SR* and *PosteriorReplay-BbB-SR*, the *TGiven-Final* accuracies are almost identical to the runs with the full regularizations. But interestingly enough, in a task-agnostic setting, the stochastic regularization leads to better *TInfer-Final* accuracies in both methods. We hypothesize that the reason for this performance increase might be related to the reason why stochastic gradient descent performs in practice better than gradient descent. Specifically, this stochastic regularization may make it easier to escape local minima. Overall, this shows that a stochastic CL regularization of the hypernetwork outputs not only allows to considerably reduce computation, but can also lead to improved results.

When comparing the behavior of individual methods for different architectures, we generally observe improved uncertainty-based task inference with increasing main network complexity (i.e., the results obtained for MLP-100,100 are generally worse than for MLP-400,400, which in turn are worse than for Lenet). This is especially noticeable for *PosteriorReplay-Dirac*, *PosteriorReplay-SSGE*, *VCL-multihead* and *PriorFocused-SSGE-multihead*, whereas the performance of *EWC-multihead* in all three considered settings is similar. We speculate that improved performance in increasingly complex architectures is due to differences in the resulting inductive biases [93]. For example, compared to an MLP, a convolutional architecture such as Lenet is supposed to have a much better inductive bias towards data with local structure (e.g., SplitMNIST), and might therefore be better suited for detecting whether an image belongs to a certain task or not.

For the MLP-400,400 we provide *TInfer-Final* results using the *KU* criterion described in SM C.6. Results can be found in Table S7. Perhaps surprisingly, we observe that this uncertainty estimate leads to quite poor task-inference results, even lower than the confidence criterion (*Conf*), suggesting that epistemic uncertainty is not properly captured by this metric on OOD data. We also tested this criterion with our coreset method (*PR-BbB-CS*), which explicitly calibrates for high aleatoric uncertainty in the data that was originally OOD (i.e., other tasks) and therefore causes the OOD data to become in-distribution (cf. SM C.7). Thus, if on in-distribution data aleatoric uncertainty is properly discounted by the proposed uncertainty estimate, poor task-inference can be expected. We indeed observe this in our new results (the task-agnostic performance obtained with this estimate is very low, i.e. 42% vs. 90% for *Ent* or 60% for *Agree*).

### D.4 PermutedMNIST-10

In this section, we consider the PermutedMNIST benchmark [20], another adaptation of MNIST to CL, where different tasks are obtained by applying, for each task, a different pixel permutation to the input digits. The results of learning ten different tasks with an MLP with two hidden layers of 100 neurons (MLP-100,100) are presented in Table S8, and with an MLP with two hidden layers of 1000 neurons (MLP-1000,1000) in Table S9. Note that, for compatibility with previous literature [87, 91], the experiments with the MLP-1000,1000 are done by padding the original MNIST images with zeros before applying the permutation, which results in inputs of size $32 \times 32$ instead of the original $28 \times 28$ dimensions.

In the MLP-100,100 experiments (Table S8), catastrophic forgetting is successfully prevented by all *PosteriorReplay* methods, as indicated by the similar performance in *TGiven-During* and *TGiven-Final* accuracies. Note, that if no explicit CL strategy is applied as in *Fine-Tuning*, severe catastrophic interference can be observed for this small network architecture. In a setting where task iden-

---
[9]The results reported here have been obtained by selecting one task embedding at random for regularization at each iteration.

Table S8: Accuracies of PermutedMNIST-10 experiments when using an MLP-100,100 (Mean $\pm$ SEM in %, $n = 10$). PR refers to *PosteriorReplay*.

|  | TGiven-During | TGiven-Final | TInfer-Final (Ent) | TInfer-Final (Conf) | TInfer-Final (Agree) |
|---|---|---|---|---|---|
| PR-Dirac | $95.70 \pm 0.03$ | $95.05 \pm 0.04$ | $75.84 \pm 0.51$ | $75.14 \pm 0.50$ | N/A |
| PR-BbB | $95.44 \pm 0.04$ | $94.35 \pm 0.06$ | $89.90 \pm 0.33$ | $88.38 \pm 0.33$ | $70.52 \pm 0.86$ |
| PR-Radial | $94.31 \pm 0.03$ | $94.30 \pm 0.03$ | $81.78 \pm 0.36$ | $79.87 \pm 0.33$ | $76.73 \pm 0.45$ |
| PR-SSGE | $93.58 \pm 0.10$ | $92.88 \pm 0.10$ | $78.94 \pm 0.73$ | $77.43 \pm 0.73$ | $68.93 \pm 0.88$ |
| Fine-Tuning | $97.44 \pm 0.01$ | $47.89 \pm 0.46$ | N/A | N/A | N/A |

Table S9: Accuracies of PermutedMNIST-10 experiments when using an MLP-1000,1000 (Mean $\pm$ SEM in %, $n = 10$). Column *TInfer-Final* represents final accuracies when task identity is inferred with the *Ent* criterion, if applicable. Otherwise this column is used to report results of methods that learn a shared softmax across all tasks. Results denoted with * are taken from van de Ven and Tolias [87] and those denoted with ** are taken from von Oswald et al. [91]. SI stands for *synaptic intelligence* [96], and DGR for *deep generative replay* Shin et al. [80]. HNET+TIR and HNET+R are CL methods based on hypernetwork-protected replay proposed in von Oswald et al. [91]. PR refers to *PosteriorReplay*.

|  | TGiven-During | TGiven-Final | TInfer-Final | TInfer-Final (Conf) | TInfer-Final (Agree) |
|---|---|---|---|---|---|
| PR-Dirac | $96.78 \pm 0.03$ | $96.73 \pm 0.03$ | $94.15 \pm 0.18$ | $93.41 \pm 0.16$ | N/A |
| PR-BbB | $96.33 \pm 0.02$ | $96.21 \pm 0.03$ | $96.14 \pm 0.03$ | $95.86 \pm 0.05$ | $85.92 \pm 0.32$ |
| PR-Radial | $97.19 \pm 0.02$ | $97.19 \pm 0.02$ | $92.92 \pm 0.20$ | $92.68 \pm 0.19$ | $92.26 \pm 0.19$ |
| PR-SSGE | $97.57 \pm 0.02$ | $97.39 \pm 0.02$ | $93.58 \pm 0.13$ | $93.39 \pm 0.12$ | $93.02 \pm 0.10$ |
| EWC-multihead | $96.87 \pm 0.02$ | $94.73 \pm 0.11$ | $81.12 \pm 0.62$ | $79.32 \pm 0.59$ | $79.46 \pm 0.57$ |
| VCL-multihead | $95.15 \pm 0.02$ | $89.72 \pm 0.24$ | $85.40 \pm 0.49$ | $81.66 \pm 0.56$ | $79.80 \pm 0.86$ |
| Fine-Tuning | $98.13 \pm 0.01$ | $90.08 \pm 0.45$ | N/A | N/A | N/A |
| EWC-growing* | N/A | N/A | $33.88 \pm 0.49$ | N/A | N/A |
| SI-growing* | N/A | N/A | $29.31 \pm 0.62$ | N/A | N/A |
| DGR* | N/A | N/A | $96.38 \pm 0.03$ | N/A | N/A |
| PR-Dirac** | N/A | N/A | $91.75 \pm 0.21$ | N/A | N/A |
| HNET+TIR** | N/A | N/A | $97.59 \pm 0.01$ | N/A | N/A |
| HNET+R** | N/A | N/A | $97.76 \pm 0.76$ | N/A | N/A |

tity is given (*TGiven*), all methods perform similarly, except for *PosteriorReplay-SSGE* which exhibits slightly lower performance. However, whenever task identity needs to be inferred, all three probabilistic methods outperform *PosteriorReplay-Dirac*, highlighting the advantages of using a Bayesian approach for inferring task identity via predictive uncertainty. In agreement with our SplitMNIST results, *PosteriorReplay-BbB* performs considerably better than *PosteriorReplay-Radial*. *PosteriorReplay-SSGE*, despite using more flexible *implicit* distributions, performs poorly compared to *PosteriorReplay-BbB*. Note that, here, individual tasks are more difficult than in SplitMNIST, and a method as complex as SSGE might be disproportionately affected by an increase in task difficulty. As opposed to our SplitMNIST results, when comparing different methods to quantify uncertainty for task inference one can observe that *Ent* systematically yields the best results, closely followed by *Conf*. However, when applicable, accuracies based on the *Agree* criterion lead to lower performance. A potential reason could be that all employed approximate inference methods lead to poor approximations that do not capture the space of admissible solutions well.

Similar trends can be observed for the MLP-1000,1000 (Table S9). Catastrophic forgetting only seems to be an issue for the prior-focused methods *EWC-multihead* and *VCL-multihead*. Interestingly, the *Fine-Tuning* baseline shows that, despite not adopting any strategy for CL, catastrophic interference is also not very severe due to the capacity of this large network. We could not find viable hyperparameter configurations for an *implicit* prior-focused method, i.e., *PriorFocused-SSGE*, and

therefore do not report results for this method. The deterministic solution *PosteriorReplay-Dirac* performs surprisingly well, specially in a task-agnostic setting (*TInfer-Final* in the Table), for which we significantly improve upon the previous baseline (*PosteriorReplay-Dirac\*\**), and that is only out-performed by *PosteriorReplay-BbB* and methods based on replay. Interestingly, despite having similar performance when task identity is given (especially *EWC-multihead*), *PriorFocused* approaches perform considerably worse than *PosteriorReplay* approaches whenever task identity needs to be inferred. The performance we obtain for *EWC-multihead* in a task-agnostic setting is, however, vastly superior to that reported by prior work (*EWC-growing\** and *SI-growing\**), highlighting the benefits of using a multihead architecture in *prior-focused* CL. Again, when task identity needs to be inferred, *Ent* results in the best performance, but as opposed to the MLP-100,100 case, *Agree* often leads to comparable results. While results reported by prior work when using generative replay methods (DGR, HNET+TIR and HNET+R) perform best, the gap with *PosteriorReplay-BbB*, which does not use generative models, is small.

When comparing the results obtained with both architectures, as expected, performance is slightly higher for the larger MLP when task identity is given. This gap in performance is much more noticeable when task identity needs to be inferred. However, this could also be explained by the fact that for the larger MLP, the dimensionality of the input images is considerably larger due to padding, which might render task inference easier.

## D.5 PermutedMNIST-100

To study whether our *posterior meta-replay* approach scales to longer task sequences, we also experimented with *PosteriorReplay-BbB* in PermutedMNIST with 100 tasks. Results for an MLP-1000,1000 are reported in Table S10. The results are obtained using the stochastic regularization *SR* in order to avoid a linear runtime increase with the number of tasks.

Table S10: Accuracies of PermutedMNIST-100 experiments when using an MLP-1000,1000 (Mean $\pm$ SEM in %, $n = 5$). PR refers to *PosteriorReplay*.

|  | TGiven-During | TGiven-Final | TInfer-Final (Ent) | TInfer-Final (Conf) | TInfer-Final (Agree) |
|---|---|---|---|---|---|
| PR-Dirac-SR | $96.73 \pm 0.02$ | $95.90 \pm 0.06$ | $70.08 \pm 0.52$ | $69.82 \pm 0.51$ | N/A |
| PR-BbB-SR | $96.92 \pm 0.05$ | $96.84 \pm 0.05$ | $85.84 \pm 0.31$ | $84.35 \pm 0.31$ | $81.33 \pm 0.52$ |

We observe that, even though task inference becomes considerably more difficult (as shown by lower *TInfer-Final* compared to PermutedMNIST-10), accuracy when task identity is provided is high. For the considered methods almost no forgetting occurs despite the large number of tasks (similar *TGiven-During* and *TGiven-Final*). Notably, we see a substantially better task inference performance of *PosteriorReplay-BbB* compared to *PosteriorReplay-Dirac*, emphasizing the importance of using principled uncertainties. The same experiment was conducted in von Oswald et al. [91], also considering a task-agnostic inference setting (termed CL3, cf. Fig. A4b in von Oswald et al. [91]). They observe that common replay methods such as DGR [80] drop to chance level because the underlying generative model is retrained on its own data, causing a drift that accumulates over many tasks. Only the proposed method HNET+R performs well with *TInfer-Final* $96.00 \pm 0.03$. This method is based on task-conditioned replay models that are regularized via Eq. 4. While this performance may appear vastly superior to the performance of *PosteriorReplay*, it should be noted that the underlying MNIST data can be easily learned with simple generative models. In general, however, the generative task of learning $p(\mathbf{X})$ is more difficult than the discriminative one $p(\mathbf{Y} \mid \mathbf{X})$, which is why we hypothesize the performance gap will shrink or even reverse for more complicated input data (e.g., natural images). However, investigating this question by scaling generative models to natural images is beyond the scope of this study.

## D.6 SplitCIFAR-10

Our results from the SplitCIFAR-10 experiment (cf. Sec. 4.4) conducted with a Resnet-32 as main network [23] are reported in Table S11. We use batch normalization in all convolutional layers. The batch normalization weights are part of the trainable weights captured by the WG system. The

Table S11: Accuracies of SplitCIFAR-10 experiments on a Resnet-32 (Mean $\pm$ SEM in %, $n = 10$). PR refers to *PosteriorReplay* and SP to *SeparatePosteriors*.

| | TGiven-During | TGiven-Final | TInfer-Final | TInfer-Final (Conf) | TInfer-Final (Agree) |
|---|---|---|---|---|---|
| PR-Dirac | 94.59 $\pm$ 0.10 | 93.77 $\pm$ 0.31 | 54.83 $\pm$ 0.79 | 54.83 $\pm$ 0.79 | N/A |
| PR-BbB | 95.59 $\pm$ 0.08 | 95.43 $\pm$ 0.11 | 61.90 $\pm$ 0.66 | 61.90 $\pm$ 0.64 | 61.36 $\pm$ 0.71 |
| PR-Radial | 94.82 $\pm$ 0.12 | 94.67 $\pm$ 0.15 | 52.89 $\pm$ 1.19 | 52.89 $\pm$ 1.19 | 50.92 $\pm$ 1.50 |
| PR-SSGE | 94.25 $\pm$ 0.07 | 92.83 $\pm$ 0.16 | 51.95 $\pm$ 0.53 | 51.93 $\pm$ 0.52 | 51.81 $\pm$ 0.48 |
| PR-BbB-BW | 95.59 $\pm$ 0.08 | 95.43 $\pm$ 0.11 | 92.94 $\pm$ 1.04 | 91.34 $\pm$ 1.45 | 93.57 $\pm$ 0.83 |
| PR-BbB-CS | 95.15 $\pm$ 0.11 | 92.48 $\pm$ 0.13 | 64.76 $\pm$ 0.34 | 64.76 $\pm$ 0.34 | 41.21 $\pm$ 0.85 |
| SP-Dirac | N/A | 95.42 $\pm$ 0.13 | 58.67 $\pm$ 0.94 | 58.62 $\pm$ 0.93 | N/A |
| SP-BbB | N/A | 96.06 $\pm$ 0.06 | 61.35 $\pm$ 0.91 | 61.36 $\pm$ 0.91 | 59.24 $\pm$ 1.05 |
| EWC-growing | N/A | N/A | 20.40 $\pm$ 0.95 | N/A | N/A |
| VCL-growing | N/A | N/A | 19.84 $\pm$ 0.53 | N/A | N/A |
| VCL-multihead | 95.78 $\pm$ 0.09 | 61.09 $\pm$ 0.54 | 15.97 $\pm$ 1.91 | 15.86 $\pm$ 1.90 | 15.86 $\pm$ 1.88 |
| EWC-Dirac | 82.50 $\pm$ 0.27 | 82.50 $\pm$ 0.26 | 25.46 $\pm$ 0.52 | 25.46 $\pm$ 0.52 | – |
| Exp-Replay | N/A | N/A | 41.38 $\pm$ 2.80 | N/A | N/A |
| Fine-Tuning | 96.59 $\pm$ 0.03 | 60.25 $\pm$ 0.77 | N/A | N/A | N/A |

batchnorm statistics are checkpointed and stored at the end of training for each task. Specifically, in the final model, each task embedding has an associated set of batchnorm statistics that will be loaded into the main network when the task embedding is selected.

In this experiment, *PosteriorReplay-BbB* performs best, followed by *PosteriorReplay-Dirac*. As for PermutedMNIST, *PosteriorReplay-SSGE* struggles, potentially due to the increased task difficulty. All three task inference criteria perform similarly, with *Ent* usually leading to the best results. The *Fine-Tuning* baseline reveals that *PosteriorReplay* methods slightly suffer from a stability-plasticity dilemma, and the use of a shared meta-model seems to affect *TGiven-During* accuracies. Also here we see that when using batches of samples for task inference as in *PosteriorReplay-BbB-BW*, one can almost match task-given (*TGiven*) and task-inferred (*TInfer*) scores. Note, that this is just a statistical accumulation effect based on the fact that we choose the correct task identity for individual sample points above chance level, and therefore the improvements due to batch-wise inference are not directly linked to the task-difficulty. We also observe improvements in *TInfer-Final (Ent)* when fine-tuning on coresets (*PosteriorReplay-BbB-CS*). However, these improvements are not as large as for SplitMNIST (cf. Table S5). In addition, the small coresets interfere with the *TGiven-Final* accuracy, suggesting that this coreset size is not sufficient to capture the richness and diversity of the data from a task.

*VCL-multihead* performs poorly in this benchmark. The correct task identity is chosen below chance level, which indicates that inducing task-specific uncertainty through the output head only is not sufficient for this challenging benchmark. We also experimented with *EWC-multihead* but observed instabilities similar to those reported in SM D.2. In particular, some weights had very low Fisher values, which led to high variance in the predictions made by the post-hoc constructed posterior, even in-distribution. We therefore consider a heuristic modification of *EWC-multihead*, where the proper post-hoc posterior construction is omitted (cf. Eq. 38), and a Dirac posterior based on the current MAP solution is used, called **EWC-Dirac**. In this case, there is still task-specific aleatoric uncertainty (similar to *PosteriorReplay-Dirac*), but no epistemic uncertainty and therewith no instability issues arising from the posterior construction using the EWC importance values. In the growing softmax baselines, *EWC-growing* and *VCL-growing*, only instances from the last task are correctly classified.

As consistently observed in all our experiments, *PosteriorReplay-Radial* underperforms *PosteriorReplay-BbB*. While this is in disagreement with the superior performance obtained with *radial* posteriors in [18], their results were obtained for a medical dataset, and are therefore not comparable to the experiments that we consider. We leave it open for future work to investigate under which scenarios *PosteriorReplay-Radial* might be a useful replacement for *PosteriorReplay-BbB*.

Table S12: Accuracies of SplitCIFAR-10 experiments on a WRN-28-10 (Mean $\pm$ SEM in %, $n = 5$). PR refers to *PosteriorReplay* and SR denotes *StochasticRegularization* on a subset of randomly selected past tasks.

| | TGiven-During | TGiven-Final | TInfer-Final (Ent) | TInfer-Final (Conf) | TInfer-Final (Agree) |
|---|---|---|---|---|---|
| PR-Dirac-SR | $96.23 \pm 0.12$ | $95.75 \pm 0.20$ | $57.50 \pm 2.42$ | $54.09 \pm 2.43$ | N/A |
| PR-BbB-SR | $93.77 \pm 0.51$ | $92.24 \pm 0.93$ | $50.23 \pm 3.96$ | $50.20 \pm 3.96$ | $49.95 \pm 3.68$ |

To study whether the *posterior meta-replay* approach can scale to even more complex architectures, we also performed SplitCIFAR-10 experiments with a Wide Resnet [95] (WRN-28-10) containing a total of 36.5 million weights. The results for *PosteriorReplay-Dirac* and *PosteriorReplay-BbB* are shown in Table S12. Here, the trend starts to reverse and even the mean-field BNN *PosteriorReplay-BbB* starts to encounter scalability issues. We hypothesize that these scalability issues can be mitigated by a more carefully chosen prior, e.g., a mean-field prior that adapts the variance of each weight by the layer's fan-in to counteract exploding or vanishing activations as they are particularly harmful in wide architectures. We again use stochastic regularization (*SR*) to showcase that even for these very complex models the full regularization as described by Eq. 1 is not necessary.

**Results reported in related work.** We are aware of several papers that considered SplitCIFAR-10 to benchmark CL algorithms under varying experimental conditions (such as the used architecture). The purpose of this paragraph is to give a brief overview over previously reported results, while appealing to the reader's experience for comparing numbers obtained under such varying conditions. For instance, Li et al. [50] consider a class-incremental scenario [87] and report results consistent with ours for a variety of well established regularization approaches such as *EWC-growing*. Best results in this study are achieved when using large random coresets (size 1000), which still perform below 45% *TInfer-Final*. Aljundi et al. [2] propose an approach for building coresets with which they obtain 49% accuracy for an overall coreset size of 1000. Also Mundt et al. [63] studies several approaches for coreset selection considering various coreset sizes, e.g., they report 53% *TInfer-Final* for a coreset of size 1500. De Lange and Tuytelaars [12] use nearest neighbor-based prediction on a set of continually evolving prototypes, obtaining up to 49% *Final* accuracy. In addition, some papers study this benchmark in a domain-incremental setting [87]. In this case, the overall problem is a binary classification, where objects with labels 0,2,4,6,8 belong to the negative class and objects with labels 1,3,5,7,9 belong to the positive class. Since in a class-incremental setting the inputs additionally need to be assigned to the correct task, the domain-incremental setting is simpler. Our framework can be readily adapted to the domain-incremental evaluation setting by changing the way the accuracy is computed. Specifically, our *TInfer-Final* accuracies necessarily become higher in the domain-incremental setting since, even if the incorrect output head has been chosen, there is still a 50% chance that the correct binary class is predicted. In this domain-incremental setting, Borsos et al. [7] provide a careful comparison of coreset methods, where all reported domain-incremental accuracies are below 40%. Chen et al. [10] propose Discriminative Representation Loss, a CL method based on decreasing gradient diversity, and report a *Final* accuracy of 40% when training in an online fashion (i.e., training the model with a single epoch on the training data). Finally, we would like to mention the work of Prabhu et al. [74], which proposes GDumb where models are trained from scratch on stored coresets. While this method is not trained on non-i.i.d. data, and thus not strictly comparable to a CL method, it provides a simple baseline whose performance questions the effectiveness of all modern CL methods. For instance, they achieve 61.3% *TInfer-Final* by training on only 1000 samples from CIFAR-10.

### D.7 SplitCIFAR-100

We conducted additional experiments on the more challenging SplitCIFAR-100 benchmark, which considers the CIFAR-100 dataset split into 10 tasks of 10 classes each. The results of this experiment conducted with a Resnet-18 as main network [23] (where the first layer has only a kernel size of 3 with stride 1 and the max-pooling is dropped as in Verma et al. [90]) are reported in Table S13.

We again see a minor improvement in performance when using *PosteriorReplay-BbB* compared to *PosteriorReplay-Dirac*. Interestingly, the trend is reversed when considering the *SeparatePosteriors*

Table S13: Accuracies of SplitCIFAR-100 experiments on a Resnet-18 (Mean $\pm$ SEM in %, $n = 10$). PR refers to *PosteriorReplay* and SP to *SeparatePosteriors*.

| | TGiven-During | TGiven-Final | TInfer-Final (Ent) | TInfer-Final (Conf) | TInfer-Final (Agree) |
|---|---|---|---|---|---|
| PR-Dirac | $85.25 \pm 0.34$ | $85.16 \pm 0.34$ | $40.35 \pm 0.37$ | $39.69 \pm 0.35$ | N/A |
| PR-BbB | $84.97 \pm 0.17$ | $84.78 \pm 0.19$ | $42.36 \pm 0.26$ | $42.00 \pm 0.23$ | $41.78 \pm 0.25$ |
| PR-Dirac-SR | $84.56 \pm 0.12$ | $84.57 \pm 0.12$ | $40.68 \pm 0.09$ | $39.46 \pm 0.08$ | N/A |
| PR-BbB-SR | $86.68 \pm 0.09$ | $86.56 \pm 0.10$ | $45.22 \pm 0.18$ | $44.55 \pm 0.21$ | $44.77 \pm 0.19$ |
| SP-Dirac | N/A | $89.52 \pm 0.05$ | $50.80 \pm 0.16$ | $47.79 \pm 0.15$ | N/A |
| SP-BbB | N/A | $82.73 \pm 0.10$ | $38.86 \pm 0.19$ | $38.52 \pm 0.20$ | $38.57 \pm 0.20$ |
| EWC-Dirac | $66.86 \pm 0.25$ | $66.83 \pm 0.24$ | $16.96 \pm 0.19$ | $17.46 \pm 0.18$ | N/A |
| Fine-Tuning | $91.10 \pm 0.05$ | $24.97 \pm 0.47$ | $0.98 \pm 0.01$ | $0.99 \pm 0.01$ | N/A |

baseline. We hypothesize that this result is due to optimization difficulties. In *SeparatePosteriors-BbB*, one has to train ten approximate posterior distributions independently per hyperparameter configuration. For most hyperparameter configurations that were tested, some of these ten posterior approximations failed in solving the respective task. In contrast, for *PosteriorReplay-BbB* we observed that all tasks were successfully learned if the first task could be learned (due to transfer in the meta-model). To optimally boost the performance among considered prior-focused methods, we again considered *EWC-Dirac* (SM D.6), which performs far worse than *PosteriorReplay* methods.

An overview on how other continual learning methods perform on this benchmark can be found in concurrent studies [89, 90], where EFT exhibits very similar performance to our method. Note, the results in van de Ven et al. [89] are based on pretrained models and thus not directly comparable. Another recent study that considers this benchmark and evaluates the effect of coresets on common class-incremental methods is Masana et al. [60].

## D.8 Task boundary detection during training

So far, we assumed that task identity is known during training. To overcome this constraint, methods for task boundary detection can be used, and the use of uncertainty for this purpose was already proposed by Farquhar and Gal [16].

Here we analyze, for PermutedMNIST, the feasibility of task boundary detection based on the uncertainty computed on a training batch, and compare it to using the loss as a boundary indicator. We consider the entropy of the predictive distribution as uncertainty measure (*Ent*). In both cases, the transitions to new tasks can be detected based on peaks in the evolution of the signals. The criterion for boundary detection can be implemented as a simple threshold crossing. As both loss and uncertainty will stay high in the initial phase of training on a new task, the detection algorithm is paused for a grace period after a detected transition. Furthermore, to improve detection stability, loss and uncertainty can be considered over a window of several training iterations.

To compare how useful loss and uncertainty values are for detecting task boundaries, we analyze the sensitivity of the two approaches to the choice of the detection threshold when using a grace period of 1000 iterations and a detection window of 10 iterations (Fig. S8). The threshold value that detects task boundaries without errors is bounded from above by the minimum of the values at actual task boundaries (such that all boundaries are detected), and from below by the maximum value outside of task boundary periods (such that false positives are avoided). While perfect boundary detection is possible with both signals, we found that the range of viable thresholds is much larger for the uncertainty signal (65.80% of the signal range) than for the loss signal (14.11% of the signal range).

These results suggest that task boundary detection based on an uncertainty measure might be a more robust way to detect task boundaries during training than a loss criterion. Interestingly, using uncertainty for task boundary detection does not require access to labelled data, as opposed to a loss-based criterion. However, an uncertainty criterion relies on the assumption that input distributions of subsequent tasks are sufficiently distinct, which might not always be case. Crucially, both of these simple threshold criteria assume some sort of continuity for the training of individual tasks.

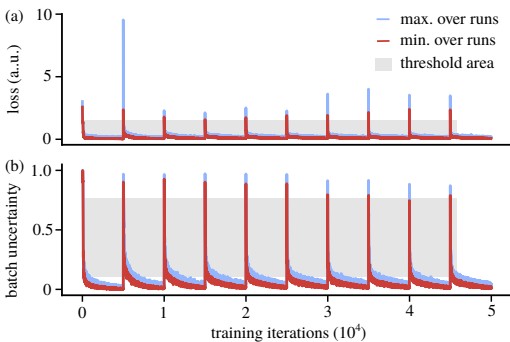

Figure S8: Task boundary detection in PermutedMNIST-10. Evolution of **(a)** the loss and **(b)** batch uncertainty when training tasks for 5000 iterations. The detection threshold of a potential task boundary detection system is bounded by the minimum (red) and maximum (black) values over 10 runs. The gray area indicates the range in which a threshold would successfully detect all task boundaries without causing false positives or false negatives.

## E  Experimental Details

In this section, we provide detailed descriptions of the two low-dimensional problems considered in this paper, and explain how hyperparameter configurations were chosen for all methods and experiments. The exact hyperparameter choice of all reported experiments can be found in the code repository accompanying the paper, which also contains the instructions to reproduce all results.

### E.1  1D Polynomial Regression Dataset

The 1D polynomial regression dataset consists of three tasks, each defined by a different ground-truth function $g^{(t)}(x)$ that operates on a specific input domain $p^{(t)}(X)$:

$$
\begin{aligned}
g^{(1)}(x) &= (x+3)^3 & p^{(1)}(X) &= \mathcal{U}(-4, -2) & (49)\\
g^{(2)}(x) &= 2x^2 - 1 & p^{(2)}(X) &= \mathcal{U}(-1, 1)\\
g^{(3)}(x) &= (x-3)^3 & p^{(3)}(X) &= \mathcal{U}(2, 4)
\end{aligned}
$$

where $\mathcal{U}(a, b)$ denotes a continuous uniform distribution between the bounds $a$ and $b$. Note that input domains are non-overlapping, and therefore task identity can be inferred by looking at the inputs alone.

Twenty noisy training samples are collected from each of these polynomials according to $\{(x, y) \mid y = g^{(t)}(x) + \epsilon$ with $\epsilon \sim \mathcal{N}(0, \sigma^2), x \sim p^{(t)}(X)\}$. We set $\sigma = 0.05$ in this experiment. We also scale the NLL properly using $\sigma_{\mathrm{ll}} = 0.05$ (cf. Eq. 9) to circumvent model misspecification when testing Bayesian approaches.

### E.2  2D Mode Classification Dataset

The 2D mode classification experiment is comprised of three binary classification tasks. The 2D inputs $\mathbf{x}$ are sampled from a Gaussian mixture model with six modes and uniform mixing coefficients:

$$
p(\mathbf{X}) = \sum_{t=1}^{6} \frac{1}{6} \mathcal{N}(\mathbf{X}; \boldsymbol{\mu}^{(t)}, \boldsymbol{\Sigma}^{(t)}) \tag{50}
$$

We set $\boldsymbol{\Sigma}^{(t)} = (0.2)^2 I$ for $t \in \{1, .., 6\}$ and equidistantly locate the modes in a circle of radius five around the origin. Specifically, the angular position of each mode was determined by $\alpha^{(t)} = \frac{t-0.5}{6} 2\pi$, and the corresponding Cartesian coordinates by: $\boldsymbol{\mu}^{(t)} = [5\sin(\alpha^{(t)}), 5\cos(\alpha^{(t)})]^T$. The dataset contains 10 training samples per mode, i.e., 20 training samples per task.

Table S14: Hyperparameter values scanned across methods in our basic search. Note that these values were subsequently further tuned for each method and experiment. BNNs refer to all probabilistic methods (i.e. all considered methods except *PosteriorReplay-Dirac*). *PR* refers to all *PosteriorReplay* methods, which therefore require a TC network. *Implicit* refers to all methods using implicit posterior approximations, which therefore require a WG network.

| Methods | Hyperparameter | Searched values |
|---|---|---|
| All | learning rate | 1e-5, 1e-4, 1e-3, 1e-2 |
| | batch size $N_{mb}$ | 32, 64, 128 |
| | clip gradient norm | None, 100 |
| | main network activation function | ReLU |
| | optimizer | Adam |
| | number of iterations (1D-Regression) | 5000, 10000 |
| | number of iterations (2D Mode Class.) | 2000, 5000 |
| | number of iterations (SplitMNIST) | 2000, 5000 |
| | number of iterations (PermutedMNIST) | 5000, 10000 |
| | number of epochs (SplitCIFAR-10) | 20, 40, 60 |
| BNNs | prior variance $\sigma^2_{prior}$ | 1 |
| | *prior-matching term* scaling factor | 1e-6, 1e-5, ... 1e0 |
| | | (1 for low-dim. experiments) |
| BNNs (except VCL and PR-BbB) | number of samples for *prior-matching term* estimation | 1, 10, 20 |
| | number of MC samples for NLL estimation $K$ | 1, 10, 20 |
| PR-BbB and VCL | CL regularizer | MSE, fKL, rKL, W2 |
| | local reparametrization trick | True (only for MLPs), False |
| EWC | regularization strength | 1e-4, 1e-3, ..., 1e3, 1e4 |
| All PR | CL regularizer strength $\beta$ | 1e-3, 1e-2, ... 1e2, 1e3 |
| | TC hidden layers | None, 10-10, 50-50, 100-100 |
| | TC activation function | ReLU, sigmoid |
| | size of task embeddings $\mathbf{e}^{(t)}$ | 16, 32, 64 |
| | initial SD of task embeddings $\mathbf{e}^{(t)}$ | 0.1, 1 |
| | size of TC chunk embeddings $\mathbf{c}^{(l)}$ | 16, 32, 64 |
| | initial SD of TC chunk embeddings $\mathbf{c}^{(l)}$ | 0.1, 1 |
| All *implicit* | variance $\sigma_{\text{noise}}$ of WG output perturbation | None, 1e-2, 1e-1 |
| | dimensionality of the latent vector $\mathbf{z}$ | 8, 16, 32 |
| | initial SD of the latent vector $\mathbf{z}$ | 0.1, 1 |
| | WG hidden layers | None, 10-10, 50-50, 100-100 |
| | WG activation function | ReLU, sigmoid |
| | size of WG chunk embeddings $\mathbf{c}^{(l)}$ | 16, 32, 64 |
| | initial SD of WG chunk embeddings $\mathbf{c}^{(l)}$ | 0.1, 1 |
| AVB | batch size $K$ for D training | 1, 10 |
| | number of D training steps | 1, 5 |
| | use batch statistics | False, True |
| | D hidden layers | 10-10, 100-100 |
| SSGE | kernel width | 0.1, 1 |
| | use heuristic kernel width | True, Ralse |
| | number of samples $S$ for gradient estimation | 10, 20, 50 |
| | threshold $\tau$ for eigenvalue ratio of eigenvalues $J$ | 1, 5, All |
| PR-BbB-CS | number of iterations for fine-tuning stage | 2000, 5000 |
| | method for OOD uncertainty increase | random labels, high entropy targets |
| | *prior-matching term* scaling factor during fine-tuning | 1e-6, ..., 1e0 |
| Exp-Replay | regularization strength | 1e-1, 1e0, 1e1, 1e2 |
| | coreset batch size | 8, 16, 32, 128 |
| | fixed mini-batch size for regularizing | True, False |

### E.3 Hyperparameter selection

To gather the results, we performed extensive hyperparameter searches for all the methods across all experiments. Unless noted otherwise, we selected the hyperparameter configurations according to the best final performance in a task-agnostic setting using the *Ent* criterion. In Table S14 we describe the basic grid of hyperparameter values used for all methods. Note that we only report the initial search grid from which a random subset of 100 calls was generated. The results of this initial search were thoroughly evaluated and the grid was fine-tuned individually for each method and experiment if parameter choices appeared inappropriate.

Notably, the network architectures differ considerably across methods and across experiments, and the lists below are not exhaustive. We use chunked hypernetworks (cf. Sec. C.2) with architectures where the total number of parameters of the TC system is smaller than $\dim(\mathbf{w})$, except for low-dimensional problems where we only experiment with non-compressive MLP hypernetworks. In addition, we experiment with principled hypernetwork initializations (e.g., cf. Chang et al. [8]), that we adapted for chunked hypernetworks. However, we do not find a noticeable influence of the initialization on the training outcome when using the Adam optimizer.

The hyperparameter searches were performed on the ETH Leonhard scientific compute cluster. For exact configurations of the reported results, please refer to our list of command line calls provided in the README files of the accompanying code base.

## F  Further Discussion and Remarks

### F.1  On Posterior Meta-Replay as a Bayesian method

Proper Bayesian inference is intractable and approximations are needed in practice, and our approach is no exception to this. However, we believe *posterior meta-replay* can be considered a Bayesian method and outline in this section why.

First of all, just like *Online EWC*, our method can be derived through a series of approximations from a probabilistic graphical model, which additionally assumes the existence of discrete tasks and therefore has as additional variable the task identifier $t$ (a design choice resulting in high performance gains). The joint can be written as $p(t)(\mathbf{x} \mid t)p(\mathbf{W} \mid t)p(\mathbf{y} \mid \mathbf{W}, \mathbf{x})$ with $p(\mathbf{x} \mid t)$ being the task-specific input distribution and $p(\mathbf{y} \mid \mathbf{W}, \mathbf{x})$ being the likelihood. $p(\mathbf{W} \mid t)$ is a Dirac distribution such that there is one ground-truth model $\mathbf{W}^{(t)}$ per task, allowing each task to be represented by a dataset $\mathcal{D}^{(t)}$ drawn i.i.d. from $p(\mathbf{x} \mid t)p(\mathbf{y} \mid \mathbf{W}^{(t)}, \mathbf{x})$. This setting naturally induces task-specific posteriors $p(\mathbf{W} \mid \mathcal{D}^{(t)})$ (cf. Fig. 1). To bring this graphical model to a practical CL method, we apply several approximations. First, the intractable posteriors $p(\mathbf{W} \mid \mathcal{D}^{(t)})$ are approximated by $q_{\boldsymbol{\theta}^{(t)}}(\mathbf{W})$ using variational inference (where we remain flexible regarding the choice of variational family). Second, as storing separate posteriors is undesirable from a CL perspective, we entangle all posteriors within a shared hypernetwork, which is trained continually. Third, in the case of task-agnostic inference (e.g., class-incremental learning), the task identity $t$ has to be explicitly inferred from the current input $\tilde{\mathbf{x}}$, which requires access to the posterior $p(t \mid \tilde{\mathbf{x}}) = \frac{p(\tilde{\mathbf{x}}|t)p(t)}{\sum_{t'} p(\tilde{\mathbf{x}}|t')p(t')}$. Since explicit modelling of $p(\tilde{\mathbf{x}} \mid t)$ is difficult, we heuristically opt for uncertainty-based task inference.

Second, *posterior meta-replay* solutions can approximate the true posteriors. Indeed, although the parameters of the TC system influence both the ELBO and the regularization term, the approximate posterior of the task being currently learned only appears in the ELBO. Therefore our optimization objective still aims to learn an approximate posterior using a proper lower bound, and if forgetting of the learned posteriors is prevented, the final per-task solutions correspond to valid posterior approximations (indendently of which mechanism is used against forgetting). This is the case in the non-parametric limit (the TC system being a universal function approximator) if the objective is optimally minimized; in this case the introduction of the hypernetwork does not impose any constraints compared to the separate posterior view prescribed by the graphical model. Although in practice the capacity of the TC system is limited and optimization is not perfect, we empirically show

that the introduced errors only marginally affect performance.[10] This is also the reason why we did not consider more sophisticated forms of regularization (for instance, as outlined in SM F.3) for our empirical analysis.

Nevertheless, forgetting is happening in practice and our current approach does not allow such forgetting to be reflected in uncertainty estimates, which would be an interesting avenue for future research. Specifically, distributions outputted by the TC system are always assumed to be approximations to the task-specific posteriors and forgetting can currently only be detected by having access to a withheld test set for each past task.

## F.2 Runtime and storage complexity

A BNN has an intrinsic runtime disadvantage during inference compared to a deterministic network due to the incorporation of parameter uncertainty. As in practical scenarios the posterior predictive distribution cannot be analytically evaluated, it has to be approximated via an MC estimate (cf. Eq. 39). Thus, if the MC estimate incorporates $K$ samples from the approximate parameter posterior, then inference of every input is approximately $K$ times as expensive as for a deterministic model.

Task inference via predictive uncertainty comes into play as an additional factor during inference, since the predictive distribution has to be estimated for every task in order to chose the prediction corresponding to lowest uncertainty. Hence, inference time is additionally increased by a factor $T$. Note, this is not the case for the considered multihead *PriorFocused* methods, where in every forward pass the output of all $T$ heads is computed anyways. Certainly, proper parallelization on modern graphics hardware can ensure that these extra demands are not noticeably reflected in the actual runtime.

During training via variational inference, BNNs require an MC estimate of the NLL term (cf. Eq. 3), which also leads to a linear increase for the loss computation compared to the deterministic case. However, it should be noted that this hyperparameter $K_{\text{NLL}}$ (the MC sample size) is often chosen to be rather small, e.g., $K_{\text{NLL}} = 1$.

Apart from these general remarks, we also would like to comment on method-specific resource demands.

**BbB and Radial posteriors.** Both methods are very similar in their implementation as well as their resource complexity. A notable difference is that an analytic expression for the *prior-matching term* is known for BbB when using certain priors, while the training of radial posteriors requires an MC estimate of the involved cross-entropy term (cf. SM C.3.2). Compared to the deterministic case, the number of trainable parameters is doubled (a mean and variance per weight) in both cases.

**Implicit methods.** Implicit methods, such as AVB and SSGE, require an additional neural network that in combination with the base distribution forms the approximate posterior. The architecture of this WG network is a hyperparameter. Note, that the choice of architecture has a considerable influence on runtime and storage complexity. Every sample drawn from such an approximate posterior requires a forward pass through the WG network.

AVB requires yet another network during training, the discriminator. Also here, the architecture of the discriminator is a hyperparameter. In every training iteration, the discriminator is optimized inside an inner-loop for a predefined number of steps. For loss computation, the *prior-matching term* has to be evaluated via an MC sample, where forward passes through the discriminator are required.

SSGE requires an eigendecomposition at every training iteration which has cubic runtime complexity in the number of samples $S$ used for the construction of the kernel matrix (cf. SM 20). Most of our results were obtained for $S = 10$ or $S = 20$.

**Posterior meta-replay.** To evaluate whether the use of task-conditioned hypernetworks requires increased computational and memory resources, we evaluated the runtime and memory usage of several methods in SplitMNIST runs (Table S15). To enable a fair comparison, we used the same set

---

[10]We empirically show this through the SP and SP-TC baselines, e.g. Table S5, where all posteriors are trained separately (i.e., the objective is only the ELBO) such that the influence of the regularizer (SP-TC baseline) or both the regularizer and TC-system (SP baseline) can be understood.

of hyperparameters across all methods. Furthermore, in methods requiring an MC estimate of the NLL or *prior-matching term*, we run experiments using a single sample, but note that both runtime and memory resources will increase with this hyperparameter. For methods using SSGE, we use 10 weight samples to construct the kernel matrix.

Table S15: Resources needed by *PosteriorReplay* in terms of runtime and memory usage. Results are reported for SplitMNIST results using an MLP-400,400 and identical hyperparameters (if applicable) for all methods (Mean $\pm$ SD in %, $n = 5$). PR refers to *PosteriorReplay* and PF to *PriorFocused* methods.

|  | Runtime (s) | Memory usage (MiB) |
|---|---|---|
| Fine-Tuning | $95.51 \pm 1.32$ | 389.0 |
| HNET Fine-Tuning | $185.7 \pm 1.9$ | 397.0 |
| PR-Dirac | $426.1 \pm 1.9$ | 439.0 |
| PR-Dirac-SR | $314.5 \pm 2.1$ | 407.1 |
| PR-BbB | $542.3 \pm 6.6$ | 473.0 |
| PR-SSGE | $1201.2 \pm 9.6$ | 537.6 |
| PF-SSGE-multihead | $834.2 \pm 4.8$ | 482.2 |
| VCL-multihead | $221.4 \pm 4.9$ | 452.6 |
| EWC-multihead | $289.0 \pm 4.1$ | 392.1 |

All results are obtained using the same compute hardware and the provided code base. Therefore, the computational complexity depends on the efficiency of this implementation and does not necessarily reflect theoretical time or space complexity. We especially want to stress that for the sake of flexibility and simplicity we use a naive hypernetwork implementation, where we first generate all main network weights before feeding them into the main network. A more efficient implementation or specialized hardware may improve the reported runtimes substantially [22].

Compared to *Fine-Tuning*, where a main network is updated without any CL protection, *PosteriorReplay-Dirac* leads to a four-fold increase in runtime, but only a slight increase in memory usage. To disentangle whether the considerable increase in runtime is linked to the addition of the hypernetwork or the computation of the CL regularizer, we also evaluated the runtime of a system consisting of a main network and a task-conditioned hypernetwork, but where no CL protection is applied (i.e., equivalent to fine-tuning the hypernetwork, denoted *HNET Fine-Tuning*). Memory usage is again in the same ballpark, but runtime only constitutes this time a two-fold increase with respect to the *Fine-Tuning* baseline that has no hypernetwork. Altogether these results show that, although memory usage is not noticeably affected by the use of our *posterior meta-replay* framework, the introduction of the hypernetwork and the CL regularization each lead to a two-fold increase in the runtime for this particular experiment.

We show that the runtime of *posterior meta-replay* experiments can be considerably shortened by doing a stochastic CL regularization and considering a subset of tasks for computing the regularizer (noted *SR* and reported here for a subset of size 1). Indeed, runtime for *PosteriorReplay-Dirac-SR* is larger than when no regularizer is computed (*HNET Fine-Tuning*), but only about 70% of the runtime when all tasks are regularized upon (*PosteriorReplay-Dirac*). Compared to the deterministic *PosteriorReplay* solution, *PosteriorReplay-BbB* requires about 20% more memory and 30% longer runtime, which can partly be explained by the fact that, when the same hyperparameters are used (including main network size), BbB has to generate twice as many parameters for the main network parametrization as *PosteriorReplay-Dirac* (since it generates means and variances, and not weight values directly). Finally, *PriorFocused* methods such as *VCL-multihead* or *EWC-multihead* again have similar memory usage, but considerably shorter runtimes than *PosteriorReplay-Dirac*. However, compared to the solution with stochastic regularization *PosteriorReplay-Dirac-SR*, the runtime of *EWC-multihead* is only about 8% faster; a negligible amount compared to the gains in performance that can be achieved. SSGE requires substantially more resources, especially in terms of runtime. This is to be expected as in every training iteration an estimate of the log-density has to be computed. It is important to keep in mind that these values will vary substantially for other hyperparameters and experiments, but we expect the trend across methods to hold.

### F.3 Continual learning regularization in distribution space

The goal of the CL regularization in the *posterior meta-replay* framework is to ensure that found posterior approximations do not change when learning new tasks, as discussed in Sec. C.2. This desideratum can be directly enforced if, for the considered variational family, an analytic expression for a divergence measure is accessible (e.g., cf. Sec. C.3.1/ Eq. 1). Whenever such divergence measure is not available, we avoid sample-based regularization, and thus regularize at the level of the output of the TC system, resorting to an L2 regularization of the distributional parameters (cf. Eq. 2). In our experiments this L2 regularization is sufficient, as we do not observe that forgetting is a major problem. A potential reason for the success of this simple regularization could be the discovery of flat minima as speculated in Ehret et al. [15]. However, a sound application of the *posterior meta-replay* framework should regularize towards closeness in distribution space, as the chosen parameterization of the variational family is arbitrary, and it is unclear how perturbations in parameter space affect the encoded distributions. Therefore, this section provides guidance for future work on how a simple regularization acting on the outputs of the TC system can be interpreted as enforcing closeness in distribution space. Our discussion on this topic is inspired by the derivation of the natural gradient [72], that allows to cast parameter updates into distribution updates.

Recall that the goal of regularization is to ensure that (cf. Eq. 1):

$$\min_{\boldsymbol{\theta}^{(t)}} \mathrm{D}\big(q_{\boldsymbol{\theta}^{(t,*)}}(\mathbf{W}) || q_{\boldsymbol{\theta}^{(t)}}(\mathbf{W})\big) \tag{51}$$

where $q_{\boldsymbol{\theta}^{(t,*)}}(\mathbf{W})$ denotes the approximate posterior of task $t$ obtained from a checkpoint of the TC network before learning the current task, and D denotes some divergence measure. In particular, we consider the KL as a choice for the divergence measure, which for small parameter perturbations behaves like a metric in distribution space.[11] Assuming the outputs $\boldsymbol{\theta}^{(t,*)}$ do not change much when learning task $t$ (i.e. $\boldsymbol{\epsilon} = \boldsymbol{\theta}^{(t)} - \boldsymbol{\theta}^{(t,*)}$ is sufficiently small), we consider a Taylor approximation of Eq. 51 around $\boldsymbol{\theta}^{(t,*)}$. To compute this expression, we use the fact that the KL between identical distributions is zero, and we compute the first and second order terms of the approximation as follows. For the first order term we require the first derivative of the KL with respect to $\boldsymbol{\theta}^{(t)}$ at $\boldsymbol{\theta}^{(t)} = \boldsymbol{\theta}^{(t,*)}$:

$$\Big[\nabla_{\boldsymbol{\theta}^{(t)}} \mathrm{KL}\big(q_{\boldsymbol{\theta}^{(t,*)}}(\mathbf{W}) || q_{\boldsymbol{\theta}^{(t)}}(\mathbf{W})\big)\Big]\Big|_{\boldsymbol{\theta}^{(t)}=\boldsymbol{\theta}^{(t,*)}} \tag{52}$$

$$= -\int q_{\boldsymbol{\theta}^{(t,*)}}(\mathbf{W}) \big[\nabla_{\boldsymbol{\theta}^{(t)}} \log q_{\boldsymbol{\theta}^{(t)}}(\mathbf{W})\big]\Big|_{\boldsymbol{\theta}^{(t)}=\boldsymbol{\theta}^{(t,*)}} d\mathbf{W}$$

$$= 0$$

For the second order term we need the respective second derivative:

$$\Big[\nabla_{\boldsymbol{\theta}^{(t)}} \nabla_{\boldsymbol{\theta}^{(t)}}^T \mathrm{KL}\big(q_{\boldsymbol{\theta}^{(t,*)}}(\mathbf{W}) || q_{\boldsymbol{\theta}^{(t)}}(\mathbf{W})\big)\Big]\Big|_{\boldsymbol{\theta}^{(t)}=\boldsymbol{\theta}^{(t,*)}} \tag{53}$$

$$= -\int q_{\boldsymbol{\theta}^{(t,*)}}(\mathbf{W})$$

$$\big[\nabla_{\boldsymbol{\theta}^{(t)}} \nabla_{\boldsymbol{\theta}^{(t)}}^T \log q_{\boldsymbol{\theta}^{(t)}}(\mathbf{W})\big]\Big|_{\boldsymbol{\theta}^{(t)}=\boldsymbol{\theta}^{(t,*)}} d\mathbf{W}$$

$$= -\int q_{\boldsymbol{\theta}^{(t,*)}}(\mathbf{W}) \big[\mathcal{H}_{\log q_{\boldsymbol{\theta}^{(t)}}(\mathbf{W})}\big]\Big|_{\boldsymbol{\theta}^{(t)}=\boldsymbol{\theta}^{(t,*)}} d\mathbf{W}$$

$$= -\mathbb{E}_{q_{\boldsymbol{\theta}^{(t,*)}}(\mathbf{w})} \big[\mathcal{H}_{\log q_{\boldsymbol{\theta}^{(t,*)}}(\mathbf{w})}\big]$$

And we therefore obtain the following local approximation of the KL:

$$\mathrm{KL}\big(q_{\boldsymbol{\theta}^{(t,*)}}(\mathbf{W}) || q_{\boldsymbol{\theta}^{(t)}}(\mathbf{W})\big) \tag{54}$$

$$= \mathrm{KL}\big(q_{\boldsymbol{\theta}^{(t,*)}}(\mathbf{W}) || q_{\boldsymbol{\theta}^{(t,*)}+\boldsymbol{\epsilon}}(\mathbf{W})\big)$$

$$\approx -\frac{1}{2}\boldsymbol{\epsilon}^T \mathbb{E}_{q_{\boldsymbol{\theta}^{(t,*)}}(\mathbf{w})} \big[\mathcal{H}_{\log q_{\boldsymbol{\theta}^{(t,*)}}(\mathbf{w})}\big]\boldsymbol{\epsilon}$$

---

[11]As we will see below, the KL divergence is for infinitesimal parameter perturbations approximately equal to the Rao distance [66].

Thus, regularization in distribution space as demanded by Eq. 51 can be achieved for small $\epsilon$ using the quadratic form in Eq. 54. We denote the regularization matrix by $R^{(t)} \equiv \mathbb{E}_{q_{\boldsymbol{\theta}^{(t,*)}}(\mathbf{w})}\left[\mathcal{H}_{\log q_{\boldsymbol{\theta}^{(t,*)}}(\mathbf{w})}\right]$ to highlight the elegance and computational simplicity of the final CL regularizer:

$$\min_{\boldsymbol{\theta}^{(t)}} \frac{1}{2}(\boldsymbol{\theta}^{(t)} - \boldsymbol{\theta}^{(t,*)})^T R^{(t)}(\boldsymbol{\theta}^{(t)} - \boldsymbol{\theta}^{(t,*)}) \tag{55}$$

Note that $R^{(t)}$ has to be computed once after task $t$ has been trained on,[12] and for $R^{(t)} = I$ we recover the isotropic regularization of Eq. 2.

When interpreting $q_{\boldsymbol{\theta}^{(t,*)}}(\mathbf{W})$ as a likelihood function, it can be seen that $R^{(t)}$ is the corresponding Fisher information matrix (cf. Eq. 31) and can be computed via:

$$R^{(t)} = \mathbb{E}_{q_{\boldsymbol{\theta}^{(t,*)}}(\mathbf{W})}\Big[\big(\nabla_{\boldsymbol{\theta}^{(t,*)}} \log q_{\boldsymbol{\theta}^{(t,*)}}(\mathbf{W})\big) \tag{56}$$
$$\big(\nabla_{\boldsymbol{\theta}^{(t,*)}} \log q_{\boldsymbol{\theta}^{(t,*)}}(\mathbf{W})\big)^T\Big]$$

Note, that the quantity estimated by algorithms such as SSGE (cf. Sec. C.4.2) is $\nabla_{\mathbf{W}} \log q_{\boldsymbol{\theta}^{(t)}}(\mathbf{W})$ and not the score function $\nabla_{\boldsymbol{\theta}^{(t)}} \log q_{\boldsymbol{\theta}^{(t)}}(\mathbf{W})$. It is therefore not straightforward to see how to approximate the Fisher information matrix and thus the regularization matrix $R^{(t)}$.

However, as deep learning often benefits from coarse approximations, we propose the following heuristic to estimate $R^{(t)}$ if the simple regularization of Eq. 2 is not sufficient. Assuming the training of each task is successful and the variational family contains the correct posterior, we have that $q_{\boldsymbol{\theta}^{(t,*)}}(\mathbf{W}) \approx \frac{1}{Z} p(\mathcal{D}^{(t)} \mid \mathbf{W}) p(\mathbf{W})$, where $Z$ is some unknown normalization constant. We can therefore rewrite the Hessian as $\mathcal{H}_{\log q_{\boldsymbol{\theta}^{(t,*)}}(\mathbf{W})} \approx \mathcal{H}_{\log p(\mathcal{D}^{(t)}|\mathbf{W})} + \mathcal{H}_{\log p(\mathbf{W})}$. The term $\mathcal{H}_{\log p(\mathbf{W})}$ has a simple analytic expression for a Gaussian prior and $\mathcal{H}_{\log p(\mathcal{D}^{(t)}|\mathbf{W})}$ can be computed for a given variate $\mathbf{w}$ as long as the data $\mathcal{D}^{(t)}$ is available and the model is twice differentiable. If $R^{(t)}$ is computed right after a task has been learned, it can be safely assumed that $\mathcal{D}^{(t)}$ is still available (similar to the Fisher computation in EWC (cf. Sec. C.5.2)). Hence, $R^{(t)}$ can be approximated via an MC estimate since sampling from $q_{\boldsymbol{\theta}^{(t,*)}}(\mathbf{W})$ is always possible. Crucially, the actual CL regularization via Eq. 56 is efficient to compute and does not require sampling from $q_{\boldsymbol{\theta}^{(t,*)}}(\mathbf{W})$.

### F.4  Optimization considerations in Posterior-Replay

We discuss here important aspects regarding the ease of optimization of our *posterior meta-replay* framework. Chang et al. [8] found that the initialization of the hypernetwork is important for training stability when optimizing via SGD. We therefore use the Adam optimizer throughout, with which the choice of initialization does not seem to be crucial. While we do not suffer noticeably from instabilities (note, that we apply gradient clipping), we do observe that certain hyperparameter choices are crucial for loss minimization (improper choices might cause the optimizer to plateau at chance level performance). For instance, as already noted in von Oswald et al. [91], the choice of hypernetwork architecture strongly influences how easy it is to find suitable hyperparameter configurations. To the best of our knowledge, there is currently no principled or agreed upon approach for choosing such architecture and the choice thus solely relies on the experimenter's expertise. Interestingly, von Oswald et al. [91] studied the sensitivity of PR-Dirac to changes in the regularization strength, and found that the accuracies are robust for a wide range of values (Fig. SM A2 in von Oswald et al. [91]).

### F.5  Deep Ensembles

A successful approach for OOD detection are deep ensembles [82], which in the simplest case correspond to several deterministic networks trained independently, and whose predictions are aggregated during inference [42]. Due to the non-convexity of the loss landscape and the stochasticity

---

[12]In the strict sense, $R^{(t)}$ is specific to the used checkpoint $q_{\boldsymbol{\theta}^{(t,*)}}$, which can change whenever a new checkpoint of the TC system is made and thus whenever a new task arrives. We ignore this detail for computational simplicity and due to the approximate nature of the proposed regularization.

innate to maximum-likelihood training, distinct solutions are obtained despite training with the same dataset. Although, to the best of our knowledge, such an ensemble of models cannot be interpreted as a sample from the Bayesian posterior, having access to a set of diverse predictions makes it possible to infer task identity based on model disagreement, which is not possible in the plain deterministic case. In addition, deep ensembles are just as easy to train as single deterministic solutions, but do require more training resources. This is in contrast to the increased training difficulty of BNNs, where continuous weight distributions are sought. However, despite their attractiveness, ensembles lack some of the advantages that come with a Bayesian approach. These include the ability to drop i.i.d. assumptions within tasks or the ability to revisit and update existing knowledge.

Analogously to *PosteriorReplay-Dirac*, our framework can also be adapted to work with ensembles. In this case, the TC system maintains multiple embeddings per task; one per ensemble member. Although not in the context of CL, this idea has already been explored by Oswald et al. [68], who showed improved OOD detection compared to the single model baseline. To combine the advantages of BNNs and deep ensembles, Wilson and Izmailov [93] proposed to train an ensemble of BNNs. This is an intriguing future direction for our framework, as the beneficial CL properties of BNNs are preserved. For instance, a simple and scalable approach such as *PosteriorReplay-BbB* could be turned into an ensemble of Gaussian posterior approximations by allowing the use of more than one embedding vector per task, which may allow to capture multiple modes of the posterior and ultimately lead to better task inference.

### F.6 Continual learning in function space

The focus of this paper is on Bayesian continual learning methods that operate primarily in weight space. However, especially from a prior-focused perspective (where a single shared solution is desired), methods like Titsias et al. [84] and Pan et al. [69], which operate in function space, offer a compelling alternative. This function space view might also be beneficial for *PosteriorReplay* methods for several reasons. First of all, Bayesian inference in function space might make it easier to encode (task-specific) prior information (e.g., via Gaussian processes). Furthermore, the function space view offers a better interpretability of OOD capabilities [11], which might ultimately be used for improving uncertainty-based task inference. For those reasons, it might be an interesting future direction to study *PosteriorReplay* methods while focusing on the function- rather than weight-space.

### F.7 Graceful forgetting

Should a CL algorithm have the built-in ability to forget past knowledge in order to facilitate learning from new data? This is an intriguing question whose answer depends on the precise definition of "continual learning". As described in Sec. 1, we focus on supervised learning and under the term CL study algorithms capable of learning from a stationary unknown data distribution $p(\mathbf{X})p(\mathbf{Y} \mid \mathbf{X})$ using a non-i.i.d. sample. In our specific case (and in line with the vast majority of the CL literature), this non-i.i.d. sample is constrained to be a series of i.i.d. samples (called tasks), e.g., $p(\mathbf{X}, \mathbf{Y}) = \frac{1}{T} \sum_{t=1}^{T} p^{(t)}(\mathbf{X})p^{(t)}(\mathbf{Y} \mid \mathbf{X})$.[13] Following this strict definition, there is no notion of *temporal valence* that we can assign to sample points. Or more specifically, there is no intrinsic reason why we should trade performance on old tasks (forgetting/stability) for learning new tasks (plasticity).

On the other hand, some studies consider a different, but related problem, namely learning from a non-stationary data distribution ([48, 41]). A good example of this scenario is learning from observations that are subject to sensor drift. The goal here is clearly different from typical CL (as defined above): the learner should adapt to current observations quickly while overwriting (possibly conflicting) previously acquired knowledge. The emphasis here is on *adapting quickly*, which requires transferring (or generalizing) acquired concepts to new observations. The contrast between these two types of online learning is clearly highlighted by the different ways in which performance is computed: in typical CL (as defined above) performance is measured across all data seen so far, whereas in settings subject to concept drift performance is measured only on the most recent data.

---

[13]Note, the overall (unknown) data distribution to be learned is stationary. However, the instantaneous data distribution that is sampled from may be considered non-stationary with discrete, non-continuous transitions (task boundaries).

Since CL algorithms should ideally also have built-in mechanisms for transfer learning, it is well justified to ask whether they can be modified to *gracefully forget* past data in order to facilitate new observations. *Prior-focused* methods incorporate past knowledge via the prior, which is the posterior of the previous task. As illustrated in Fig. 1, this prior progressively becomes more confident, which impedes learning. Therefore, Li et al. [48] suggest to broaden this prior by tempering the current prior $p(\mathbf{W})$ when task boundaries are detected: $p^{\beta}(\mathbf{W}) \equiv \frac{1}{Z_{\beta}} p(\mathbf{W})^{\beta}$, with inverse temperature $\beta$ and normalization $Z_{\beta}$. Plugging this expression into the *prior-matching term* of Eq. 3 yields:

$$-\mathrm{KL}\big(q_{\boldsymbol{\theta}}(\mathbf{W}) \,||\, p^{\beta}(\mathbf{W})\big) = \qquad\qquad\qquad (57)$$
$$- \mathbb{E}_{q_{\boldsymbol{\theta}}(\mathbf{W})}\big[\log q_{\boldsymbol{\theta}}(\mathbf{W}) - \beta \log p(\mathbf{W})\big] + \log Z_{\beta}$$

Hence, whenever the *prior-matching term* is estimated via an MC sample, this type of tempering can be achieved by scaling the log-prior density. Note that this is different from the type of tempering that we apply (cf. Sec. D), which scales the *prior-matching term* as a whole.

Also Kurle et al. [41] proposes forgetting mechanisms for prior-focused learning, the first of which is similar in spirit to the one described above (cf. Eq. 11 in Kurle et al. [41]).

In this context, it is worth stressing that *PosteriorReplay* methods do not explicitly suffer the same plasticity (ability to learn new tasks) vs. stability (prevention of forgetting) trade-off. Instead, the prior of each task can be freely chosen without sacrificing previously acquired knowledge. For instance, the prior can be a weighted average of an arbitrary prior and previously learned posteriors, i.e., it can be broad to encourage exploration of solutions that fit upcoming data well while being biased towards solutions of previous tasks to exploit prior knowledge.

As a final remark, it is important to note that algorithms are always applied to systems with limited capacity. Therefore, it is natural to ask what happens to a CL algorithm if the system's capacity is exceeded but new data still arrives, i.e., what compromise will it find. The CL algorithms discussed in this paper have no active mechanisms to free up capacity, and therefore their behavior in the limiting regime can only be partially controlled via the regularization strength. For *PriorFocused* methods, the regularization strength can be tuned via tempering. Since *PosteriorReplay* methods explicitly regularize all previous tasks when learning on new data, forgetting can be concentrated on a subset of tasks by choosing task-specific regularization strengths. Therefore, also *PosteriorReplay* methods can employ *graceful forgetting* by discounting the regularization strength depending on the age of a task.