# OpenReview forum: "Posterior Meta-Replay for Continual Learning"
_NeurIPS.cc/2021/Conference — NeurIPS 2021 Poster_

### Official Review · Reviewer_mFdX · 2021-07-04

**Rating:** 3
**Confidence:** 4

**Summary:**

This submission proposes a continual learning method based on a shared hypernetwork that parameterises a distribution over the network parameters for each task given a learned task embedding. It presents this as a Bayesian method, which is **not correct** as the objective function does not minimize a divergence from the posterior. Practically, the method is mostly **incremental** over (von Oswald et al., 2020), replacing the weight outputs of the hypernetwork with the parameters of a distribution over these. Finally, the paper omits large parts of the literature on continual Bayesian deep learning, displaying a disappointing **lack of scholarship**. Overall, this is a **clear rejection**.

Reference:
J. von Oswald et al. Continual learning with hypernetworks. In ICLR 2020.

**Main Review:**

**Technical correctness**

The paper argues that it approximates the posterior for each task. Considering that the objective introduces an ad-hoc term to prevent forgetting, this assertion is not correct. Optimizing the parameters of a distribution w.r.t. the variational lower bound is a principled approximation as this is equivalent to minimizing the KL divergence between the approximate and true posterior, hence we get the exact result if the variational family is sufficiently expressive. Optimizing a distribution over the parameters does not automatically constitute a Bayesian method, even if the variational lower bound appears in the objective function, as the addition of ad-hoc terms to the objective breaks the property of minimizing the deviation from the posterior. This is exacerbated here by changing the approximate posterior on a task when further training on subsequent tasks. I'd be curious if a similar objective as the one used in this work could be derived from a hierarchical probabilistic model, however there is no discussion pointing towards this unfortunately.

The paper further proposes to infer the task identity based on the predictive uncertainty of each task-specific weight distribution. This is referred to as "principled" without further explanation. To me, a principled approach would involve modelling the task identity as part of the probabilistic model, approximating the posterior given a data point and then accordingly marginalizing(!) over all task-specific posteriors for the final predictions. The paper does not justify its approach theoretically. Semantics aside, the major drawback of Bayesian neural networks is the computational cost of having to calculate multiple forward passes. The proposed approach further aggravates this by requiring multiple forward passes  _for each task_ observed so far, which is entirely impractical considering that ensembling of a single model is already prohibitive in many settings. The computational cost should at least be discussed and not swept under the rug (there are some runtime results in the appendix, but those appear to be for the setting with given task identities at test time).

**Originality**

The core of the method is a modification of [1] (referred to as "PR-Dirac" in this paper), which parameterizes the network parameters via a meta-network. PR-Exp instead outputs a per-parameter mean and a variance, samples the parameters from the corresponding factorized Gaussian and regularizes with a divergence measure towards previous distributions instead of a quadratic penalty, both of which are a rather incremental change over [1]. More interestingly, PR-Imp outputs the parameters of yet another hypernetwork, which in turn samples the parameters of the network of interest based on a noise input. However, this seems to introduce little additional difficulty and judging from the paper can be achieved with similar techniques as the original hypernetwork in [1]. Further, hierarchical models for neural network weights have been considered in various works on Bayesian neural networks, see also the scholarship section.

**Clarity**

Overall the paper is reasonably clear in its description of the method. Key variables are not defined unambiguously in some places, e.g. $\mathcal{L}_{task}$ is only specified in the appendix. Also, which divergence was used for PR-Exp in eq. (1)? The text insinuates the KL divergence, but how did you choose between forward and reverse KL? The equation implies that you're requiring the old distribution to be covered by the new one, but that seems like it would allow for parameter settings that perform poorly on the old data?
Finally, the pervasive use of custom acronyms makes the paper rather cumbersome to read.

**Significance**

The methodological advances over [1] are incremental. The proposed method for test-time task inference is prohibitively expensive and appears to only work on MNIST, placing the significance of this work at the low end of the spectrum.

**Scholarship**

There are numerous prior works on Bayesian methods for continual deep learning, which the paper mostly ignores [2-7]. It is  also  inaccurate to state that EWC (in constrast to online EWC) finds a shared posterior for all data due to the per-task terms (see the note by Huszar referenced as [25] in the paper). Besides works on Bayesian continual learning, a discussion of federated learning methods, e.g. [8,9], would seem relevant, as these typically work with per-task posteriors as well. [10] similarly consolidates per-task Gaussians into a shared Gaussian that it carries forward, so PR-Exp could be related as an amortized version of that procedure (which maintains the ability of producing per-task Gaussians). Finally, works on hierarchical weight models for neural networks are again not discussed [11-13]

**Further comments**

- The references are formatted inconsistently (venue names, title capitalization, inclusion/omission of page/issue/etc numbers) and look like they have been copy-pasted from Google scholar.
- Figure 2c suggests that the implicit posterior is potentially multi-modal, however the paper presents no evidence that this is the case in the experiments beyond toy datasets, making the figure potentially misleading. Given the similar performance of PR-Imp to PR-Exp I would find it highly surprising if the distribution was indeed multimodal.
- Are you sure that Figure 3a shows PR-Exp and not PR-Imp? I have not seen a mean-field Gaussian variational posterior with a remotely comparable increase in predictive uncertainty away from the data anywhere in the literature or my own experiments before. If this is indeed the result for PR-Exp, this would point towards this not really being a variational method as discussed above.

References

[1] von Oswald et al. Continual learning with hypernetworks. In ICLR 2020.

[2] Titsias et al. Functional regularisation for continual Learning with Gaussian processes. In ICLR 2020.

[3] Loo et al. Generalized variational continual learning. In ICLR 2021. (on arxiv since Nov 2020)

[4] Pan et al. Continual deep learning by functional regularisation of memorable past. In NeurIPS 2020.

[5] Adel et al. Continual learning with adaptive weights. In ICLR 2020.

[6] Ritter et al. Online structured Laplace approximations for overcoming catastrophic forgetting. In NeurIPS 2018.

[7] Zeno et al. Task agnostic continual learning using online variational Bayes with fixed-point updates. arXiv preprint arXiv:2010.00373, 2020.

[8] Bui et al. Partitioned variational inference: A unified framework encompassing federated and continual learning. arXiv preprint arXiv:1811.11206, 2018.

[9] Yurochkin et al. Bayesian nonparametric federated learning of neural networks. In ICML 2019.

[10] Lee et al. Overcoming catastrophic forgetting by incremental moment matching. In NeurIPS 2017.

[11] Dwaracherla et al. Hypermodels for exploration. In ICLR 2020.

[12] Karaletsos et al. Probabilistic meta-representations of neural networks. arXiv preprint arXiv:1810.00555, 2018.

[13] Karaletsos & Bui. Hierarchical Gaussian process priors for Bayesian neural network weights. In NeurIPS 2020.

**Time Spent Reviewing:**

5

---

> ### Author Response · Authors · 2021-08-10
> **Reply to reviewer mFdX**
>
> We thank the reviewer for the considerable amount of time taken to critically assess our manuscript. We found the input very valuable, as it has helped us strengthen several parts of the manuscript. Below we address all points individually.
>
> Whether our posterior meta-replay approach can be considered **Bayesian** was questioned by the reviewer, which rightly pointed out the fact that an extra regularization term is added to the optimization objective. Interestingly, although the parameters of the TC system influence both the ELBO and the regularization term, the approximate posterior of the task being currently learned only appears in the ELBO. Therefore our optimization objective still aims to learn an approximate posterior using a proper lower bound. However, the concern is still valid as there is an implicit constraint on the variational family due to factors such as the TC architecture or the regularization term. Simply speaking, only in the nonparametric limit can the TC system realize any desired output mapping, and therefore in practice not every member of the variational family can be realised. This implicit shrinkage of the variational family is now discussed in the paper.  Note, that we empirically control for these influences via the SP and SP-TC baselines (e.g., Table S5). In this case, all posteriors are trained separately (i.e., the objective is only the ELBO) such that the influence of the regularizer (SP-TC baseline) or both the regularizer and TC-system (SP baseline) can be understood. An interesting alternative that sidesteps the need to use the regularizer is to train a task-specific posterior solely via the ELBO and distill the solution after completing the current task into the TC-system. This option is supported by our code base, but we empirically observed lower performance with this alternative in a set of preliminary experiments. To appropriately address this concern beyond the comments added to the main text, we have now added a discussion to the SM.
>
> The reviewer also raised concerns about calling our **task-inference system principled**, and argued that a principled approach would require including task identity as part of the probabilistic model. We thank the reviewer for this valuable comment with which we fully agree (we even comment on this in the “Task Inference” paragraph of our discussion, line 405). However, as mentioned in the same paragraph, because of existing limitations for learning generative models on high-dimensional data, we opted for using the predictive uncertainty for task-inference instead. Indeed, an important message we aim to stress in the paper is that the OOD capabilities of the Bayesian posterior depend on the induced prior in function space, and that it remains to be proven that, for the explored settings, the induced priors in function space lead to high epistemic uncertainty in OOD regions.  In this regard, we acknowledge that calling our task-inference approach “principled” can indeed be misleading. We thank the reviewer for pointing this out. We have now carefully scanned the text and corrected this.
>
> We thank the reviewer for interesting pointers to **related literature**. We agree that having a related work section as complete as possible is crucial, and therefore incorporated most of the listed studies. We briefly outline how we think these studies relate to our work.
> [3], [5] and [6] are further prior-focused methods that operate in weight space. Interestingly, [5] utilizes a set of task-specific neuronal parameters, which could be a promising extension of the EWC/VCL-multihead setting when considering explicit task-inference via uncertainty. [7] is an online prior-focused learning method, and could therefore be applied within our framework to learn *within* tasks in a non-iid fashion (cf. line 432). [2] and [4] are prior-focused methods that operate in function space. Function space priors are interesting because they are considered more interpretable, and could also lead to benefits in terms of OOD detection. However, functional regularisation methods currently only approximate the posterior in function space for a chosen input domain, which is undesirable for OOD detection. Within our framework, they could be used to set desirable priors for the task-specific posteriors. [10] learns task-specific posteriors which are post-hoc merged into a single one. [11], [12] and [13] explore alternative ways of using hypernetworks to model BNNs, although not in the context of continual learning. We now incorporated these references into SM line 909. Although we are not familiar with the federated learning literature (e.g., [8] and [9]), we agree that these algorithms can also be useful when studying the problem of continual learning. For instance, the method proposed in [8] can be seen as a prior-focused approach when applied to CL (cf. Sec. 4.3 in [8]).
> We hope this captures the links between related literature and our work that the reviewer had in mind, and remain open to further discussion.
>
> The reviewer raises concerns about the **originality** of our work, arguing that our PR-Dirac baseline is identical to the method HNET+ENT proposed in [1] (as we state in the paper) and that the methods proposed are a probabilistic extension of this baseline. However, we don't see our contribution as a mere methodological extension of PR-Dirac to the probabilistic setting, but also as a thorough exploration of the range of possibilities that open up with an explicit treatment of parameter uncertainty.
> Some examples are the incorporation of coresets to update knowledge in a prior-focused way or the ability to exploit knowledge about the type of OOD data to facilitate task inference (cf. SM C.7). Furthermore, we highlight that epistemic uncertainty is an important factor for uncertainty-based task-inference as illustrated in Sec. 4.2. Finally, we also consider the thorough empirical evaluation of methods that fall into our framework as being of interest to the community.
>
> Regarding concerns about the **computational cost** of our method, we fully agree that the need to evaluate all posteriors when working in a task-agnostic inference setting is one of the main limitations of our approach compared to other Bayesian CL approaches, which we openly discuss in our “Limitations” paragraph of the discussion. Note, however, that the choice of inference setting does not affect training time, which constitutes the bulk of the runtime (the runtimes reported in SM F.1 are indeed for the task-agnostic inference setting).
>
> Regarding the **significance of our results**, we believe our experiments extensively show that our method works in datasets beyond MNIST. Indeed, our SplitCIFAR-10 results are to the best of our knowledge state-of-the-art, as we outline in the related work paragraph of SM D.6. While we understand that the PR-Dirac/HNET+ENT baseline might currently be more appealing for computational reasons, we highlight that results via this baseline on benchmarks other than MNIST have not been reported before. Finally, our new SplitCIFAR-100 results provide further evidence on the scalability of our method.
>
> Because no performance benefits were obtained for **implicit methods** in complex datasets, it is indeed relevant to ask whether the obtained posteriors exploit the conferred flexibility by, for example, exhibiting multimodality. Interestingly, we do observe multimodality in the obtained posteriors with SSGE in more complex datasets such as SplitMNIST, showing that limitations are not linked to posteriors being unimodal. Illustrative plots (similar to Fig. S4) can be readily added to the SM if deemed useful by the reviewer.
>
> Regarding the **divergence measure** in Eq. 1 when working with PR-Exp, we did not choose a priori one particular measure, but set it as a hyperparameter that could take one of the following forms: forward KL, reverse KL, Wasserstein-2 distance and mean-squared error (see SM Table S12). In practice, we did not observe that the choice of divergence measure is crucial (cf. SM line 861). We now comment on this in the Methods section.
>
> We also thank the reviewer for pointing out the ambiguity regarding **EWC/Online EWC** in the main text. Indeed, we refer to Online EWC as EWC throughout the paper for brevity (SM C.5.2, line 1071). We now state this in the main text to make it clear for the reader. The exact implementation we consider is derived in SM C.5.2.
>
> We confirm that **Fig. 3a shows PR-Exp** is obtained via Bayes-by-Backprop. Crucially, the posterior approximation depends on many factors, such as the $x/y$ domains of the chosen polynomial, the ground-truth likelihood variance and training set size (cf. Eq. 9), as well as architecture and weight prior. Furthermore, we confirmed the qualitative behavior of the posteriors on OOD data with similar results obtained using the SP baseline which obtains a separate posterior per task via ELBO maximization.
>
> Finally, we fixed all remaining text issues such as bibliography formatting, variables definition.
>
> We remain available to answer any further questions or elaborate more extensively if our clarifications are not yet sufficient.

---

> > ### Comment · Reviewer_mFdX · 2021-09-04
> > **Response**
> >
> > **tl;dr** The technical exposition and positioning of the paper require extensive changes, I am still convinced that it should be rejected and will keep my score. In its current state the paper is misleading to non-expert readers regarding the technical justifications of the method and therefore not fit for publication.
> >
> > Thank you for your detailed response, I have re-read the paper and discussed the rebuttal with the other reviewers. While I agree that adding regularization terms to the lower bound does not change that it is a lower bound on the log evidence, this only applies to the per-task objectives. The distributions over the network weights for previous tasks are still changed in an entirely ad-hoc manner when observing more data, so they can be no longer considered approximate posterior distributions. This is of course not problematic as such, not every continual learning method needs to be Bayesian, but this paper quite clearly positions itself as such. This positioning needs to be justified by describing a joint probabilistic model for all data and proposing a procedure for approximating the posterior over the unknown variables -- in the case of variational inference this would involve deriving the objective that is optimized. If forgetting is entirely prevented by ad-hoc terms in the objective, I don't see any justification for declaring this a "Bayesian" continual learning method. I don't mind at all if there are tweaks to improve performance for the final algorithm, but the core method needs a well-justified foundation.
> >
> > Rigor matters in a scientific piece of writing, a consistent use of terminology forms the basis of efficient communication. The proposed method as such is of course not unreasonable (i.e. what the paper does/implements), however the technical explanations simply don't hold up (i.e. why it works/what it does conceptually). This is extremely problematic, as acceptance at a conference leads to the claims in a paper being cited and propagated through the field. To me, the paper would need to be fundamentally re-written and either positioned as a "continual deep learning with distributions" method or carefully justify its full objective function based on a probabilistic model. This is also the reason why I pointed out the lack references to Bayesian continual learning works. The point is not to just mention each of them with a sentence in the related work section in the appendix to increase their citation count, but for including the most relevant ones in the methodology section to highlight what methods the paper builds on and where it diverges from prior work. This is done well for the (von Oswald et al., 2020) reference, but missing entirely for Bayesian continual learning methods (e.g. the original EWC method which maintains a "posterior" per task for regularization, but always predicts with the current one might be particularly relevant). Again, as the paper positions itself explicitly as Bayesian, this is the most relevant body of related work -- how exactly the hypernetwork is constructed is ultimately just an implementation detail if an existing method is used.
> >
> > Minor additional notes:
> > * I did not mean to suggest that the method does not work beyond MNIST in my review. However, the only instance where an uncertainty based method (PR-Exp/Imp) significantly outperforms the deterministic baseline PR-Dirac in the TInfer setting for a non-toy dataset is PR-Exp on Split-CIFAR (plus the small fully-connected architecture on MNIST, but that one underperforms overall). As PR-Dirac is a baseline, this limits the significance of the results to me and draws the claim of the paper that uncertainty helps with task identification into question (whether that is a general problem or specific to the proposed method is of course impossible to tell).
> > * I forgot to mention this in my original review, but I find the discussion around Eq. 2 quite hand-wavy and poorly justified and would encourage you to either remove it or put it on a more principled footing. Considering the identity an approximation to the Fisher means that we could also refer to plain SGD as a natural gradient method. Placing an L2 penalty on changes to the parameters of a distribution may work empirically, but it does not work because it minimizes a KL divergence -- just how SGD often works well in practice, but not because it takes advantage of the information geometry of the problem at hand. Again, rigor and terminology matter and approximations need to be justified.

---

> > > ### Author Response · Authors · 2021-09-06
> > > **On Posterior Meta-Replay as a Bayesian method**
> > >
> > > We thank reviewer *mFdX* for the critical feedback, which challenges us to further clarify our positioning and design choices. However, we respectfully disagree with the view that it is unjustified to call our CL method Bayesian, and explain below why.
> > >
> > > **Just like Online EWC, our method can be derived through a series of approximations from a probabilistic graphical model**, which additionally assumes the existence of discrete tasks and therefore has as additional variable the task identifier $t$ (a design choice resulting in high performance gains). The joint can be written as $p(t) p(x \mid t) p(W \mid t) p(y \mid W, x)$ with $p(x \mid t)$ being the task-specific input distribution and $p(y \mid W, x)$ being the likelihood. $p(W \mid t)$ is a Dirac distribution such that there is one ground-truth model $W^{(t)}$ per task, allowing each task to be represented by a dataset $\mathcal{D}^{(t)}$ drawn i.i.d. from $p(x \mid t) p(y \mid W^{(t)}, x)$. This setting naturally induces task-specific posteriors $p(W \mid \mathcal{D}^{(t)})$ (cf. Fig. 1). To bring this graphical model to a practical CL method, we apply several approximations. First, the intractable posteriors $p(W \mid \mathcal{D}^{(t)})$ are approximated by $q_{\theta^{(t)}}(W)$ using variational inference (where we remain flexible regarding the choice of variational family). Second, as storing separate posteriors is undesirable from a CL perspective, we entangle all posteriors within a shared hypernetwork, which is trained continually. Third, in the case of task-agnostic inference (e.g., class-incremental learning), the task identity $t$ has to be explicitly inferred from the current input $x$, which requires access to the posterior $p(t \mid x) = \frac{p(x \mid t) p(t)}{\sum_{t'} p(x \mid t') p(t')}$. Since explicit modelling of $p(x \mid t)$ is difficult, we heuristically opt for uncertainty-based task inference, which we empirically study throughout the paper, outlining limitations, influencing factors, and drafting ways for improvements.
> > >
> > > **Posterior meta-replay solutions can approximate the true posteriors.**
> > > As acknowledged by reviewer *mFdX*, our per-task objectives still correspond to a proper lower bound. Therefore, if forgetting of the learned posteriors is prevented, the final per-task solutions correspond to valid posterior approximations (no matter how ad-hoc the mechanism against forgetting is). This is the case in the non-parametric limit (the TC system being a universal function approximator) if the objective is optimally minimized; in this case the introduction of the hypernetwork does not impose any constraints compared to the separate posterior view prescribed by the graphical model. Although in practice the capacity of the TC system is limited and optimization is not perfect, we empirically show that the introduced errors only marginally affect performance. This is also the reason why we did not consider more sophisticated forms of regularization (for instance, as outlined in SM F.2) for our empirical analysis.
> > >
> > > **It is equally justified to question whether Online EWC is Bayesian.**
> > > Online EWC (cf. SM C.5.2) recursively applies a Laplace approximation with diagonal covariance. More specifically, capturing the true posterior via such approximation would require considering $p(W \mid \mathcal{D}^{(1)}, \dots, \mathcal{D}^{(T)}) \approx p(\mathcal{D}^{(T)} \mid W) p(W \mid \mathcal{D}^{(1)}, \dots, \mathcal{D}^{(T-1)})$, while Online EWC applies a Laplace approximation on $p(\mathcal{D}^{(T)} \mid W) q_{\theta^{(1:T-1)}}(W)$. The relation of the final approximation $q_{\theta^{(1:T)}}(W)$ with the posterior $p(W \mid \mathcal{D}^{(1)}, \dots, \mathcal{D}^{(T)})$ is not discussed in the original studies. In addition, the effect of this recursive accumulation of approximation errors is hard to empirically quantify (for instance, there is no unique Laplace approximation of $p(W \mid \mathcal{D}^{(1)}, \dots, \mathcal{D}^{(T)})$ for deep networks to compare to). In contrast, we apply an established Bayesian approximation per task and can empirically measure the accumulated effect of our CL regularizer on those approximations during continual learning.
> > >
> > > Proper Bayesian inference is intractable and approximations are needed in practice. At what point those approximations deviate too much from the underlying graphical model, such that it is unjustified to consider a method as Bayesian, is a philosophical question that we do not intend to answer in this response. However, as long as approximations are clearly stated, future work can improve upon those in order to more faithfully capture the underlying probabilistic model. To acknowledge the relevance of the Bayesianity question, we detail the content of this discussion in a new SM section.

---

### Official Review · Reviewer_KFBY · 2021-07-11

**Rating:** 7
**Confidence:** 3

**Summary:**

The paper proposes to tackle task-agnostic continual learning from a Bayesian deep learning perspective. In contrast to previous work along these lines the paper proposes to parameterize the posterior (over network weights) using a mixture distribution with one mixture component per task (seen so far). Three main options are explored - a point estimate (i.e. a single set of network weights) per task, a parametric distribution (Gaussian w. diagonal covariance) per task, and an “implicit” distribution parameterized by a neural network (using the reparametrization trick essentially). All three options can be unified/generalized by learning a weight-generator which, when conditioned on the task, produces a distribution of weights for the network that solves the task (e.g. the classifier). Importantly, the parameters of the weight-generator are produced by another neural network (the hyper-network) whose input is conditioned on the task. The question is how this task-conditioning is obtained: the paper proposes to automatically infer the task via the (epistemic) uncertainty modeled by the distribution of functions (classifiers) defined by the task-conditioned posterior over parameters (produced by the hyper-network). Experiments on synthetic data and continual learning versions of MNIST and CIFAR-10 are shown - with lots of ablations and control experiments. Results look promising, though they hint at some scalability issues for implicit distributions (an issue commonly observed in the Bayesian Deep Learning literature), as well as issues with reliable task inference via epistemic uncertainty on OOD data (another important open problem whose solution is beyond the scope of this paper). Limitations and conclusions are well discussed and related literature if well summarized.


**Contributions:**

1) Extension of a previously published approach to Bayesian CL via hyper-networks. The main novelty is to allow for the hyper-networks to parameterize distributions over networks via the weight-generator (including the earlier work as a special case - the MAP case with point-estimates for the posterior). The necessity of this extension is well motivated and the benefits are demonstrated nicely on synthetic data and result in significant performance gains on the image classification tasks. Significance: a main drawback of a SOTA method was identified and solved in a well-motivated and satisfactory manner (particularly from a Bayesian perspective). While similar approaches can be found in the literature (as discussed in the paper), the combination with automatic task inference is novel, and crucial relies on this innovation. I think the method will receive a fair amount of attention in the Bayesian CL community.

2) Automatic task inference via the epistemic uncertainty on OOD data given by the Bayesian ensemble (parameterized by the hyper-network). While previous CL methods have proposed to use task inference - the combination with contribution 1) allows for a potentially powerful and principled solution to the problem. Benefits (and some remaining issues) are shown in the results. Significance: the approach is very reasonable from a theoretical/principled point of view. The combination of contributions 1) and 2) is perhaps one of the most promising advantages of Bayesian CL. The particular way this is executed in the paper is novel and original (to the best of my knowledge), and it is well motivated and critically analysed in the paper. The results obtained here will be important for the Bayesian CL community, and allow for a clear separation of the task identification problem such that improvements in reliable modeling of epistemic uncertainty (which is an important topic outside CL) will directly contribute to improvements of Bayesian CL.

3) Solving the problem of catastrophic forgetting and task interference with a regularizer (posterior meta-replay) that acts on the level of the hyper-network (parametrization of the posterior over classifier weights). Significance: solving the problem is crucial to successful CL - the approach taken here is sound and empirical results underline the success of the regularizer. This is an interesting result for the Bayesian CL community.

4) Well-written and comprehensive discussion of the two main innovations, including a range of specific implementations for each. These variants are discussed, related with the current literature and evaluated in a large number of experiments. Results are critically evaluated and limitations are nicely discussed. Significance: given how well the paper is written and executed, and the wealth of experimental results, ablations and control experiments, I think the paper has the potential to become a seminal next step in Bayesian CL. My only slight worry is that the paper certainly pushes against the constraints of a conference-format paper, with many interesting parts of the paper pushed to the appendix, and a quite high information density in the main text.


**Ethical Concerns:**

No ethical concerns.

**Limitations And Societal Impact:**

Limitations are nicely discussed in a separate section, including societal impact.

**Main Review:**

**Originality, Quality, Clarity, Correctness:**
To the best of my knowledge, the precise method proposed in the paper is novel and original. There are some highly related methods in the literature, which are discussed in the paper and the innovations/distinctions are made very clear. The method is well motivated and aligns well with a principled Bayesian CL approach. The paper is very well written, though the density of material in the main text is very high and the supplementary material is extensive. The main innovations are well motivated and these motivations are supported with illustrations and experiments on synthetic data. Rather than simply aiming to “beat some benchmark numbers” the paper conducts a very differentiated analysis through a set of control experiments and ablations. The main results of this are promising and in favor of the method. Limitations and caveats are well discussed in the paper.

**Verdict:**
To me, this is a well written paper, with clearly articulated main ideas, which are critically analyzed and discussed. The main problem (continual learning) is timely and important, and a Bayesian solution is sound and promising - the focus in the paper is on sequences of image classification tasks on the same data-set, where i.i.d. sampling is possible within each sub-task. This is an important limitation, but is clearly pointed out in the paper, and helps to make the focus of the work clear. Perhaps the main value of the paper is to provide a framework with various choices for parameterizing the weight generator, implementing the task inference, etc. Future improvements for these individual parts can easily be incorporated, and I would expect to see a fair number of follow-up work in this direction (e.g. normalizing-flow based implicit distributions). Given my current understanding of the literature I am clearly in favor of accepting the paper. I have some small comments and one concrete suggestion for improving estimation of epistemic uncertainty. I am, of course, happy to reconsider my verdict based on the other reviews and the authors’ response. I am not a top expert on the Bayesian CL literature (but very familiar with Bayesian DL and uncertainty estimation), hence my confidence rating.

**Pros:**

 * Principled approach, well embedded in the current literature and SOTA methods for Bayesian CL and Bayesian neural networks.
 * Well explained and illustrated motivations. Good explanation of the individual parts of the method, what role each part plays, and what reasoning went into the choices made for each part (and potential alternatives). This allows future work to easily connect/improve by focusing on individual parts.
 * Promising results, backed up and critically analyzed through a range of control experiments, comparisons and ablations.
 * Great discussion of current limitations (and potential societal impacts), relations to current literature made very clear.

**Cons:**

 * Paper at the limit of the conference format - main paper quite dense, loads of interesting discussion, experiments, etc. pushed to appendix (which is itself quite extensive)
 * No results beyond CIFAR-10, which is a common limitation in Bayesian DL
 * Some issues with reliable estimation of epistemic uncertainty and scalability of variational inference for the implicit parametrization remain (though this is clearly shown and discussed, and these are two important unsolved problems in the wider literature)

**Improvements:**

1) Improvement to estimate epistemic uncertainty via the entropy (line 233-236). Ent (entropy over predictive uncertainty) captures epistemic and aleatoric uncertainty (which is undesirable). The two factors can be teased apart fairly easily by a scheme similar to Eq. (3) in [1]. The main idea is to take the difference between the entropy of the average prediction and the average entropy of the individual predictions, where the average is taken w.r.t. the MC samples from q_\theta(W)). For high aleatoric uncertainty but no epistemic uncertainty both quantities should be equal and cancel out by subtracting. High epistemic uncertainty is quantified with a strictly positive quantity (a mutual info term, similar to an entropic term).

[1] Decomposition of Uncertainty in Bayesian Deep Learning for Efficient and Risk-sensitive Learning, Depeweg et al. 2018.

2) Experiments beyond CIFAR-10 would be nice (though I personally don’t regard them as crucial for publication).

3) The main text is fairly dense (and given this density it is very well written). The method consists of a number of components (with various choices per component), which further increases the complexity of the text (again, given this complexity I think the text is very well written). Additionally, many results and some discussion does not fit into the main paper at all, leading to a fairly lengthy appendix. Overall I think the paper would have benefited from a publication format that allows for longer main text. Given the constraints of a conference format I think the authors did a great job, and I certainly think that the work is an excellent fit for NeurIPS. Just wanted to mention this here for consideration for future publications.


**Comments:**

1) Potential drawback that should be discussed: for some tasks a lot of the task-specific posteriors might look very similar - having a single joint posterior in this case could lead to efficiency gains in terms of learning and transfer, compared to having a separate posterior per task. These gains might be partially harnessable through the hyper-network (which, after all, is shared across tasks) but it is unclear to which degree this would happen.

2) Which inference metric was used for identifying the task in 4.3 - Ent or Agreement? Perhaps using BALD (as suggested in the minor improvement above) leads to improvements (particularly over Ent).

3) I found Fig. 2 very helpful :)


**Time Spent Reviewing:**

3

---

> ### Author Response · Authors · 2021-08-10
> **Reply to reviewer KFBY**
>
> We thank the reviewer for the effort taken to carefully review our paper, and for the positive evaluation, which we found very encouraging.
>
> Given that one of the main focuses of the paper is the use of uncertainty for task-inference, we found the suggestion about **alternative ways to quantify uncertainty** very interesting. The neat decomposition of the entropy in [1] is indeed very appealing. However, we are unsure about whether these terms can indeed be interpreted as epistemic and aleatoric uncertainty. Especially on OOD data, these interpretations may fail, given that aleatoric uncertainty is a property of the data and only defined on the support of $p(x)$ (otherwise the conditional $p(y|x)$ is undefined).
> Nevertheless, we were curious to see how this estimate of epistemic uncertainty (noted KU in the table below) performs in practice, and obtained the following results for SplitMNIST-10 with an MLP-400,400.
>
> |           	| TGiven-During 	| TGiven-Final 	| Tinfer-Final (Ent) 	| Tinfer-Final (KU) 	| Tinfer-Final (Conf) 	| Tinfer-Final (Agree) 	|
> |-----------	|---------------	|--------------	|--------------------	|-------------------	|---------------------	|----------------------	|
> | PR-Exp    	| 99.73 ± 0.01  	| 99.73 ± 0.01 	| 72.27 ± 0.81       	| 46.53 ± 1.04      	| 61.74 ± 1.07        	| 72.27 ± 0.82         	|
> | PR-Imp    	| 99.80 ± 0.01  	| 99.80 ± 0.01 	| 71.30 ± 0.58       	| 53.68 ± 0.43      	| 55.98 ± 0.48        	| 71.05 ± 0.59         	|
> | PR-Exp-CS 	| 99.36 ± 0.06  	| 98.37±0.11   	| 90.26 ± 0.20       	| 42.86 ± 1.31      	| 90.26 ± 0.20        	| 60.02 ± 0.90         	|
>
> Perhaps surprisingly, we observe that this uncertainty estimate leads to quite poor task-inference results, even lower than the confidence criterion (Conf), suggesting that epistemic uncertainty is not properly captured on OOD data. We also tested this criterion with our coreset method (PR-Exp-CS), which explicitly calibrates for high aleatoric uncertainty in the data that was originally OOD (i.e. other tasks) and therefore causes the OOD data to become in-distribution (cf. SM line 1232). Thus, if on in-distribution data aleatoric uncertainty is properly discounted by the proposed uncertainty estimate, poor task-inference can be expected. We indeed observe this in our new results (the task-agnostic performance obtained with this estimate is very low, i.e. 42\% vs. 90\% for entropy or 60\% for model disagreement). We are curious to hear whether the reviewer has any thoughts on our interpretation of these results, and are happy to add a corresponding SM section if deemed valuable by the reviewer.
>
> The reviewer also inquired about the **knowledge transfer abilities** of our method. As rightly pointed out, transfer via the hypernetwork can occur since the model is shared across tasks  (also see Fig. 4 and 5 in [2]), but this transfer is indeed implicit and difficult to control for. Interestingly, our probabilistic extension allows us to explicitly transfer knowledge via the task-specific prior, which can be chosen to be the posterior of a similar task. If it is known that incoming data belongs to any previously observed task, prior-focused learning within a task-specific posterior can be employed rather than learning a new posterior. Finally, if posteriors of tasks are deemed similar a posteriori, they may be merged into a single one, and distilled into the TC system (using the same regularizer to prevent forgetting of the remaining posteriors). Although we comment on some of these aspects in several places of the manuscript, no section is currently devoted to transfer. If desired, this could be added to the SM.
>
> Unless noted otherwise, all *TInfer-Final* results including those in Sec. 4.3 are obtained with the entropy criterion (Ent). We now made this more explicit in the manuscript.
>
> Finally, we expanded our experiments to a **more challenging benchmark** by adding SplitCIFAR-100 results (see general response to all reviewers).
>
> *************************************************************************
>
> [1] Depeweg et al., Decomposition of Uncertainty in Bayesian Deep Learning for Efficient and Risk-sensitive Learning, PMLR 2018.
>
> [2] von Oswald et al. Continual learning with hypernetworks, ICLR 2020.

---

> > ### Comment · Reviewer_KFBY · 2021-08-18
> > **Thanks for the clarifications and additional results.**
> >
> > I want to thank the authors for their comments, in particular the additional results using the epistemic uncertainty estimation method (KU) and the Split-CIFAR-100 results. Thanks also for the discussion of knowledge transfer. Taking into account the authors' response and the other reviews I remain convinced that his paper is ready for publication at NeurIPS and vote and argue in favor of acceptance.
> >
> > I think that reviewer mFdX raises some important points (a strictly Bayesian approach is detailed by mFdX, whereas the method in the paper is probably best seen as an approximation of a hierarchical Bayesian model - with, admittedly, a number of approximations and implicit task-inference rather than an explicit belief over tasks; I personally think these approximations are in line with the current state-of-the-art literature, and while more principled modelling is always nice, I think the paper strikes a good balance between practicability and approximations to the full Bayesian solution). I agree with mFdX that this could be discussed better in the paper, and particularly the discussion of related literature pointed out by mFdX adds to the strength of the paper.
> >
> > Regarding the epistemic uncertainty estimation: the results are indeed surprising, thanks for running the experiments. I suspect that these results could be interesting to a larger audience and having them in the appendix might be appreciated by other practitioners. As the authors correctly point out, the estimation of epistemic and aleatoric uncertainty (or rather the decomposition predictive uncertainty into these two additive components) relies on p(y|x) being well-defined outside the data. This could (in theory) be ensured with e.g. a parametric model, like a Gaussian process with a well-defined prior and a certain length-scale, where predictions "far enough outside the data" are guaranteed to fall back to the prior. With neural networks, typically a large enough ensemble of independently randomly initialized networks produces a similar effect, such that the ensemble makes uniform predictions outside the data. However the "length-scale" is not clear in the latter case, and smaller ensembles have been empirically observed to be correlated outside the training data. The story for Bayesian neural networks tends to be even more complex (where weights of the "members of the stochastic ensemble" are stat. correlated in complex ways). I am surprised to see that the epistemic uncertainty leads to worse performance than entropy alone (which should results from a mixture of epistemic and aleatoric uncertainty). The investigation by the authors makes sense, and it seems that the epistemic uncertainty estimates of OOD data break down catastrophically compared to the entropy estimates. Again, this is somewhat unexpected to me, but more detailed investigations (though interesting) are clearly outside the scope of the paper.

---

### Official Review · Reviewer_jqRM · 2021-07-16

**Rating:** 7
**Confidence:** 4

**Summary:**

The authors propose a Bayesian approach for continual learning that models the weights distributions in a task-conditioned fashion, then relying on task inference.
As expected, the main main issue is the task inference phase.
The experimental validation is complete, although it is limited to only a small number of tasks in the same sequence.

**Main Review:**

Main merits:
- The paper is interesting and the proposed method is sound.

Main limitations:
- split-MNIST and split-CIFAR are very simple baselines, and all continual learning tests presented only have a small number of tasks.
- i would definitely not call cifar10 a 'natural image dataset'


**Time Spent Reviewing:**

0.5

---

> ### Author Response · Authors · 2021-08-10
> **Response to reviewer jqRM**
>
> We thank the reviewer for the positive assessment of our work.
>
> To complement the results shown in the original manuscript, we have added results for the **more challenging SplitCIFAR-100 benchmark** (which we no longer call *natural*), consisting of 10 tasks with 10 classes each (see general response to all reviewers).
>
> Regarding **longer task sequences**, we would like to point to SM D.5, where we explored the  scalability of our system to sequences of 100 tasks and observed, in the PermutedMNIST-100 benchmark, that our system successfully prevents forgetting for both deterministic and probabilistic solutions.

---

### Official Review · Reviewer_noon · 2021-07-16

**Rating:** 7
**Confidence:** 3

**Summary:**

The authors propose a new method for (offline) continual learning using bayesian hypernetworks to obtain a distribution over a task-based parameter generating function. This meta perspective enables replay at the level of level of the hypernet parameters (rather than at the input level), while the bayesian approach provides reliable uncertainty estimates which the authors show can be used to perform task inference at test time.

The hypernet setup, (which is really a 3-level hierarchy) comprises a 1) Task-Conditioned network, which maps a task embedding to hypernet parameters $\theta_t$, 2) a hypernet with params $\theta_t$ remaps a noise sample $z$ to a set of weights $w$, 3) which are used as parameters for the downstream classifier.

The authors provide two toy setups to illustrate the validity of their method, before benchmarking their method on Split-Cifar and Split Mnist.

**Ethics Review Area:**

["I don’t know"]

**Limitations And Societal Impact:**

Yes

**Main Review:**

__Originality__:  The proposed method is novel: the bayesian hypernet perspective is a fresh take in the CL field. The related works section is complete and cited adequately.

__Quality__: The submission is technically sound and the claims are well supported via motivation and empirical results. The toy experiments are beautifully designed, especially the second one. The work is well rounded and the authors have made a good use of the place available.

__Clarity__: The method is well explained. It is a very involved method (a 3-stage pipeline) but the authors do a good job to convey the important information across. The only caveat I will add to this is regarding the spectral Stein gradient estimator. Ideally the authors should provide some intuition on how it works, or refer to the appendix for a clearer explanation. To me this part seems quite important, as it key to optimizing implicit hypernets.

__Significance__: The results are significant, especially due to the task inference module. This area is largely unexplored and the authors take a solid step to perform principled inference.

__Other__:
One thing I would like the authors to shed light on is the difficulty of the optimization process. The proposed method is quite complex, and from the looks of it seems nontrivial to optimize. How stable / sensitive to hparams is the method ?
Moreover, it would be best if the authors discuss how the hparam selection was made for the baselines considered. Were all methods given a similar budget ?
Lastly, a good addition to the paper would be an ER baseline, with a buffer size equivalent to the size of the core-set used for `-CS` methods.


**Time Spent Reviewing:**

2.5

---

> ### Author Response · Authors · 2021-08-10
> **Reply to reviewer noon**
>
> We thank the reviewer for the positive evaluation of our method, experiments and manuscript in general.
>
> The question of **ease of optimization** of our method is indeed a crucial one. [1] found that the initialization of the hypernetwork is important for training stability when optimizing via SGD. We therefore use the Adam optimizer throughout, with which the choice of initialization does not seem to be crucial. While we do not suffer noticeably from instabilities (note, that we apply gradient clipping), we do observe that certain hyperparameter choices are crucial for loss minimization (improper choices might cause the optimizer to plateau at chance level performance). For instance, as already noted in [2], the choice of hypernetwork architecture strongly influences how easy it is to find suitable hyperparameter configurations. Interestingly, [2] studied the sensitivity of PR-Dirac to changes in the regularization strength, and found that the accuracies are robust for a wide range of values (Fig. SM A2 in [2]). We now comment on all these aspects in our SM.
>
> Regarding **hyperparameter searches** (see SM E.3 for details), the number of hyperparameters that had to be chosen varies widely across methods. For this reason, a thorough exploration of the hyperparameter space was only feasible for the simplest methods, while more complex methods required spending more search budget for exploring a still limited set of hyperparameter combinations. Note, that our hyperparameter search allowed us to improve upon many of the previously reported baselines (e.g., cf. Table S5).
>
> We agree that **experience replay** results could be an interesting baseline for comparison in our experiments. Although for SplitCIFAR-10 we discuss several papers with such results (last paragraph of SM D.6), either the size of the coresets is not comparable, or the continual learning scenario is different from ours. Therefore, we will add experience replay results with coreset size equivalent to our -CS baseline as suggested (i.e. 100 per task). Preliminary results are shown below (Mean $\pm$
> SEM in \%, $n =10$).
>
> |                             	| Final        	|
> |-----------------------------	|--------------	|
> | SplitMNIST-10 (MLP-400,400) 	| 88.85 ± 0.39 	|
> | SplitMNIST-10 (MLP-100,100) 	| 86.84 ± 0.51 	|
> | SplitCIFAR-10 (Resnet-32)   	| 41.38 ± 2.80 	|
>
> Finally, to provide the reader with some intuition about how **SSGE** works, we expanded the explanation in the main text. Furthermore, a reference to further SSGE details in SM C.4.2, as well as to the alternative implicit method AVB (which we only explore in the SM) are provided in the main text.
>
> *************************
> [1] Chang et al., Principled Weight Initialization for Hypernetworks, ICLR 2020
>
> [2] von Oswald et al. Continual learning with hypernetworks, ICLR 2020.

---

> > ### Comment · Reviewer_noon · 2021-08-18
> > **Thank you for the additional information**
> >
> > The authors have provided in depth explanations to my initial concerns. The supplementary material contained many of the information I was originally seeking.
> >
> > I will raise my score accordingly. Note I will however lower my confidence score, as other reviewers pointed out aspects of the paper which I originally missed.
> >
> > Thank you.

---

### Author Response · Authors · 2021-08-10
**General response to all reviewers**

We thank all reviewers for the time taken to assess our manuscript and for the valuable feedback and suggestions. Based on the feedback, we incorporated some changes into the manuscript that we discuss in the point-by-point answers. Here we briefly summarize major modifications.

Reviewers jqRM and KFBY suggested that the paper would benefit from a more difficult benchmark. Therefore, we will incorporate results on the SplitCIFAR-100 benchmark, which considers the CIFAR-100 dataset split into 10 tasks of 10 classes each. Preliminary results obtained with a Resnet-18 are reported in the table below (Mean $\pm$ SEM in \%, $n =10$, for a description of EWC-Dirac please refer to SM line 1561).

|             	| TGiven-During 	| TGiven-Final 	| Tinfer-Final (Ent) 	| Tinfer-Final (Conf) 	| Tinfer-Final (Agree) 	|
|-------------	|---------------	|--------------	|--------------------	|---------------------	|----------------------	|
| PR-Dirac    	| 85.25±0.34    	| 85.16±0.34   	| 40.35±0.37         	| 39.69±0.35          	| N/A                  	|
| PR-BbB      	| 84.97±0.17    	| 84.78±0.19   	| 42.36±0.26         	| 42.00±0.23          	| 41.78±0.25           	|
| SP-Dirac    	| N/A           	| 89.52±0.05   	| 50.80±0.16         	| 47.79±0.15          	| N/A                  	|
| SP-BbB      	| N/A           	| 82.59 ± 0.09 	| 38.55 ± 0.38       	| 38.18 ± 0.42        	| 38.28 ± 0.42         	|
| EWC-Dirac   	| 66.86±0.25    	| 66.83±0.24   	| 16.96 ± 0.19       	| 17.46±0.18          	| N/A                  	|
| Fine-Tuning 	| 91.10±0.05    	| 24.97±0.47   	| N/A                	| N/A                 	| N/A                  	|

---

### Decision · Program_Chairs · 2021-09-27

**Decision:**

Accept (Poster)

**Comment:**

We thank the authors for their detailed clarifications. Most reviewers agreed that this paper makes interesting contributions to the area of continual learning, which is relevant to the NeurIPS community. There were extended discussions about the Bayesian positioning of the paper and gaps in the related work. The former was settled by the authors' detailed response how online EWC can be obtained from a series of approximations and how posterior meta-replay can be derived by approximations of the posterior. Regarding the latter, I would encourage the authors to not merely cite, but to clearly explain how their work builds on prior work.